# Emergence of ion-channel-mediated electrical oscillations in *Escherichia coli* biofilms

**Emmanuel Akabuogu[1,2], Victor Carneiro da Cunha Martorelli[1,2], Rok Krašovec[3], Ian S Roberts[1]\*, Thomas A Waigh[2]\***

[1]Division of Infection, Lydia Becker Institute of Immunology and Inflammation, School of Biological Sciences, University of Manchester, Manchester, United Kingdom; [2]Biological Physics, Department of Physics and Astronomy, University of Manchester, Manchester, United Kingdom; [3]Division of Evolution, Infection and Genomics, School of Biological Sciences, Faculty of Biology, Medicine and Health University of Manchester, Manchester, United Kingdom

**\*For correspondence:**
i.s.roberts@manchester.ac.uk (ISR);
t.a.waigh@manchester.ac.uk (TAW)

**Competing interest:** The authors declare that no competing interests exist.

## eLife Assessment

This potentially **valuable** study presents claims of evidence for coordinated membrane potential oscillations in *E. coli* biofilms that can be linked to a putative K+ channel and that may serve to enhance photo-protection. The finding of waves of membrane potential would be of interest to a wide audience from molecular biology to microbiology and physical biology. Unfortunately, a major issue is that it is unclear whether the dye used can act as a Nernstian membrane potential dye in *E. coli*. The arguments of the authors, who largely ignore previously published contradictory evidence, are not adequate in that they do not engage with the fact that the dye behaves in their hands differently than in the hands of others. In addition, the lack of proper validation of the experimental method including key control experiments leaves the evidence **incomplete**.

**Abstract** Bacterial biofilms are communities of bacteria usually attached to solid strata and often differentiated into complex structures. Communication across biofilms has been shown to involve chemical signaling and, more recently, electrical signaling in Gram-positive biofilms. We report for the first time, community-level synchronized membrane potential dynamics in three-dimensional *Escherichia coli* biofilms. Two hyperpolarization events are observed in response to light stress. The first requires mechanically sensitive ion channels (MscK, MscL, and MscS) and the second needs the Kch-potassium channel. The channels mediated both local spiking of single *E. coli* biofilms and long-range coordinated electrical signaling in *E. coli* biofilms. The electrical phenomena are explained using Hodgkin-Huxley and 3D fire-diffuse-fire agent-based models. These data demonstrate that electrical wavefronts based on potassium ions are a mechanism by which signaling occurs in Gram-negative biofilms and as such may represent a conserved mechanism for communication across biofilms.

## Introduction

Dense microbial communities attached to surfaces are classified as biofilms (*Kassinger and van Hoek, 2020*). Biofilms account for ~80% of chronic infections and are costly to eradicate in medical applications (*Sharma et al., 2016*). Bacteria in biofilms have recently been found to modulate their membrane potentials similar to excitable eukaryotic cells (*Liu et al., 2017*; *Whitehead et al., 2001*). Potassium

ion-channel-linked electrical signaling was first characterized in Gram-positive *Bacillus subtilis* biofilms (*Prindle et al., 2015*). It enables communication in bacterial biofilms due to the transmission of potassium wavefronts in the local environment of the biofilm. The potassium wavefront occurs in both centripetal and centrifugal varieties (*Blee et al., 2019*) and emerges spontaneously after the biofilm has grown to a critical size (*Coombes, 2001*). Photodynamic therapy using blue light causes a process of stage-dependent cell dispersal in biofilms and membrane hyperpolarization (*Blee et al., 2020*). Such therapy has thus been investigated in detail for the treatment of topical infections (*Kharkwal et al., 2011*; *Cieplik et al., 2018*).

Membrane potential dynamics in prokaryotes are orchestrated using ion-channels. Gating of these channels can be controlled by voltage changes, heat, light stress, metabolic stress, mechanical stress, and chemical agents (*Prindle et al., 2015*; *Bruni et al., 2017*; *Martinac et al., 2008*; *MacKinnon, 2004*; *Yang et al., 2020*). In *B. subtilis* biofilms, only the potassium ion channel, *YugO*, has been directly linked to electrical signaling (*Prindle et al., 2015*).

To date, robust ion-channel-mediated signaling in biofilms of Gram-negative bacteria has not been described. Gram-negative bacterial biofilms have higher resistance to antimicrobial molecules than Gram-positive bacterial biofilms (*Mishra et al., 2020*; *Bruni and Kralj, 2020*). Although voltage-related spiking dynamics have been observed in single *E. coli* cells (*Bruni and Kralj, 2020*; *Kralj et al., 2011)*, only second long stochastic transients have been measured with no distinct coordination between intercellular spikes.

The Kch potassium ion channel in *E. coli* was discovered in *E. coli* by *Milkman, 1994* using comparative genetic techniques. The current consensus is that Kch is a voltage-gated ion channel, although the evidence is slightly indirect (*Kuo et al., 2003*; *Beagle and Lockless, 2021*). The current study provides additional evidence for the voltage-gated nature of the Kch channel in *E. coli* and indicates a physiological role for the ion channel.

We studied three-dimensional ion-channel-mediated signaling in *E. coli* biofilms. We found that under light stress, *E. coli* hyperpolarize twice in response to continued light radiation. We hypothesize that the first peak is when *E. coli* first register the presence of an external stressor in their vicinity and appears to be mediated by mechanosensitive (MS) ion channels. The depolarization and subsequent second peak that occurs in response to continued stimulation corresponds to a habituation phenomenon and is dependent on the Kch potassium-gated channel. On the basis of these data, we devised models (Hodgkin-Huxley and 3D fire-diffuse-fire agent-based models) that explain ion-channel-mediated signaling in *E. coli* biofilms. The work provides a novel outlook on the emergent electrophysiology of bacterial biofilms.

## Results

### Blue light triggers electrical spiking in single *E. coli* cells

We exposed *E. coli* (DH5α) to a blue LED (*Figure 1A*). Single sparse cells are defined as those with no neighboring cells within 10 μm. We monitored the membrane potential dynamics with the cationic fluorescent dye, Thioflavin (ThT; *Biancalana and Koide, 2010*). ThT is a Nernstian voltage indicator *Plásek and Sigler, 1996* which accumulates because bacterial cells have negative potentials (*Blee et al., 2020*; *Stratford et al., 2019*; *Humphries et al., 2017*). We observed a cell-wide rise in the intensity of fluorescence, a period of quiescence followed by a slow increase in intensity which persisted until the end of the 60-min experiments (*Figure 1B*, *Video 1*). Applying the blue light for different time periods and over different timescales yielded no change in the number of peaks (*Appendix 1—figure 1A*). We confirmed that this spike profile existed in other *E. coli* strains (*E. coli* BW25113, *Appendix 1—figure 1B*) and was also detectable when *E. coli* cells were grown in Minimal (M9) media (*Appendix 1—figure 1C*).

We observed similar spiking dynamics when we employed the lipophilic cationic cell permeant membrane potential dye, tetramethyl rhodamine methyl ester (TMRM; *Appendix 1—figure 1D*). We then tested if the observed dynamics is related to the membrane potential or autofluorescence. We used carbonyl cyanide m-chlorophenyl hydrazone (CCCP) which rapidly quenches membrane potential related dynamics in *E. coli* (*Yang et al., 2020*; *Kralj et al., 2011*; *Perry et al., 2011*). When CCCP was added, we observed a fast efflux of ions in all cells and no spiking dynamics (*Appendix 1—figure 1E*), confirming that the observed dynamics is membrane potential related.

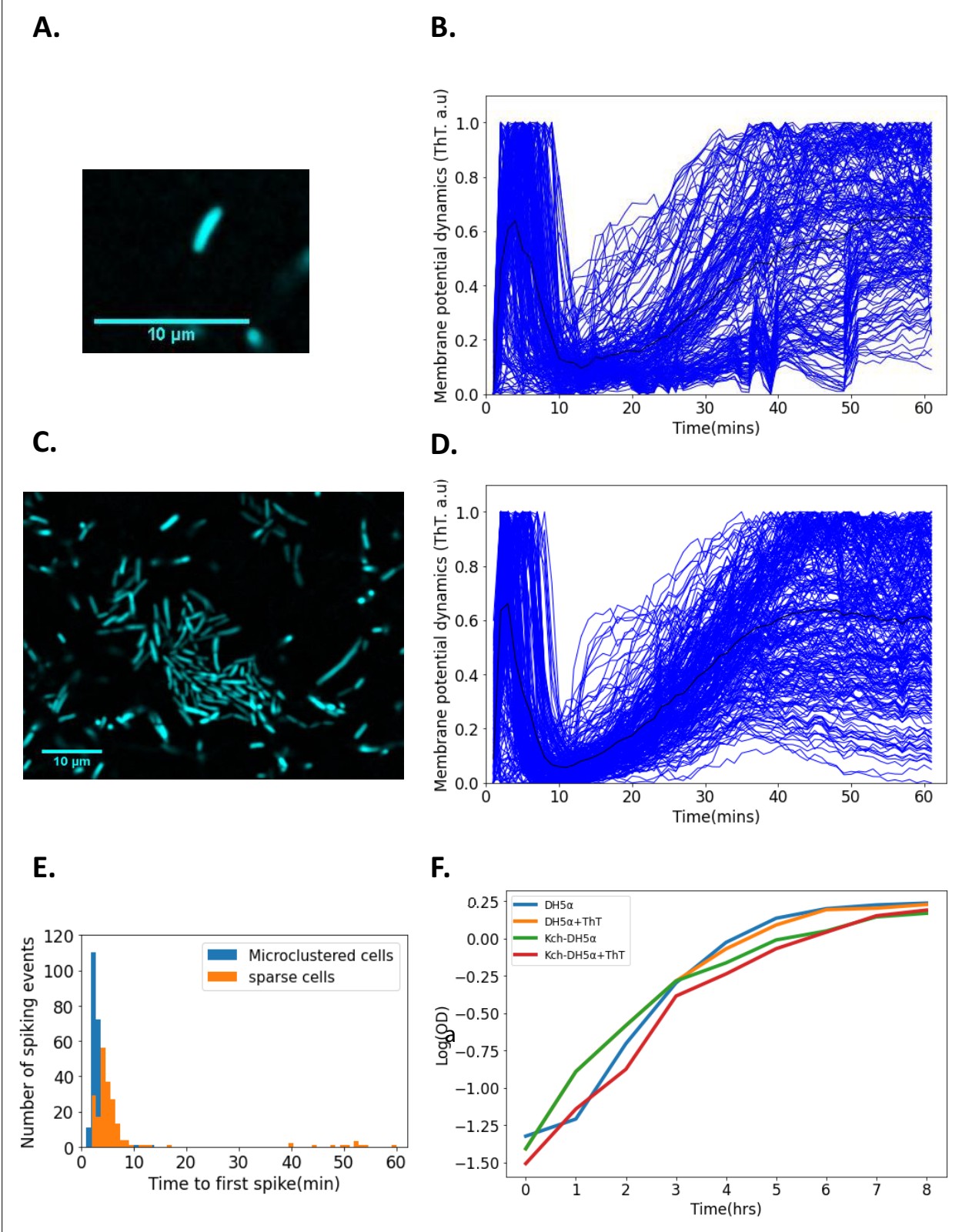

**Figure 1.** Single-cell DH5α *E. coli* exhibit membrane potential dynamics in response to 440 nm blue light stress. (**A**) Image of a sparse single cell containing ThT imaged in the microfluidic device (Scale bar: 10 µm). (**B**) Normalized fluorescence intensities of ion-transients for sparse cells (n=206) as a function of time after stimulation. Each curve describes a single cell. The curve depicting the mean membrane potential dynamics is shown in black. (**C**) Representative image of microclustered cells containing ThT in the microfluidic device. (**D**) Fluorescent intensity of ion-transients for cells in

*Figure 1 continued on next page*

*Figure 1 continued*

microclusters as a function of time after stimulation. Each curve describes a single cell. The curve depicting the mean membrane potential dynamics is shown in black. (**E**) Time to first spike histogram for sparse cells (n=206, Sparse cells in orange) and cells in microclusters (n=272, microclustered cells in blue, cells recovered from 15 clusters). The number of spiking events is shown as a function of time to the first spike. (**F**) Growth curves (in a semi-log coordinates) for *E. coli* (measured via $OD_{600}$) as a function of time in the presence and absence of ThT. All data were from at least three experimental replicates. Light stress was applied for 60 min. The scale bars for all the images are 10 µm.

Two groups have challenged the use of ThT *Mancini et al., 2020* and similar voltage sensitive fluorophores (DiSc3) in bacteria (*Buttress et al., 2022*; Appendix). These authors assume bacteria are unexcitable cells, whereas we provide evidence for excitability, invalidating their claims.

## Membrane potential dynamics depend on the intercellular distance

We hypothesized that the time-to-first peak latency of cells in dense microclusters of *E. coli* could differ from that of sparse single *E. coli* cells. A microcluster is defined as a cell community in which intercellular distances do not exceed two cellular diameters (*Figure 1C*). We applied the same light stimuli as before to *E. coli* DH5α microclusters. We observed a rapid rise in intensity, a decay and subsequently a persisting second peak (*Figure 1D*, *Video 2*). The analysis of time-to-first peak latencies in sparse and microclustered cells showed that the average time-to-first peak was 7.34 ± 10.89 ± 4.44 min (mean ± *SD* ± SE) and 3.24 ± 1.77 ± 0.53 min (mean ± *SD* ± SE; *Figure 1E*), respectively. The membrane potential dynamics of single cells showed more variability in spikes and less synchrony in the phases of the first spikes than those in microclusters (*Figure 1B and D*). This suggests that random electrical signaling in *E. coli* synchronizes as the cells become clustered. We would expect that mutual shielding from the light at higher cell densities should decrease the irradiance that cells experience which should increase the reaction time of the bacteria, however, in our experiments the opposite is observed. We also confirmed that 10 µM of ThT does not affect the growth of the *E. coli* strains used in the experiment (*Figure 1F*).

## Emergence of synchronized global wavefronts in *E. coli* biofilms

A microfluidic chamber (*Appendix 1—figure 2A and B*, Microfluidic section in Methods) was used to explore the growth of *E. coli* from single cells into biofilms (*Figure 2A*, *Appendix 1—figure 3A*; *Prindle et al., 2015*; *Blee et al., 2020*). We exposed our mature biofilm to blue light. We observed a spontaneous rapid rise in spikes within cells in the center of the biofilm while cells at the periphery remained significantly less bright (*Figure 2A and B*, *Video 3*). The ion-channel-mediated wavefronts moved from the center of the biofilm to the edges. At this point, the whole biofilm had an equal level of fluorescence intensity. The wavefronts then rapidly collapsed from the edges to the center of the biofilm. Once the wavefront reached the center, the whole system engaged in a period of quiescence and remained dark even in the presence of

**Video 1.** Fluorescence microscopy images of ThT in sparse single *E. coli* cells irradiated with blue light.
https://elifesciences.org/articles/92525/figures#video1

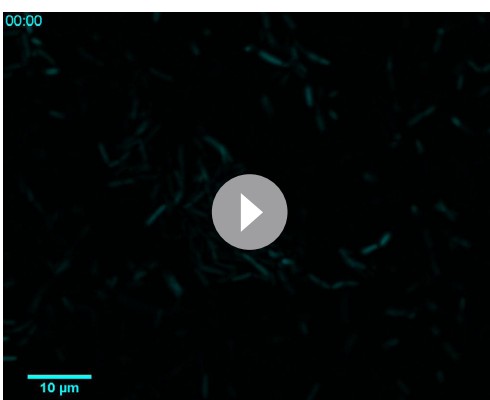

**Video 2.** Fluorescence microscopy images of ThT in *E coli* microclusters irradiated with blue light.
https://elifesciences.org/articles/92525/figures#video2

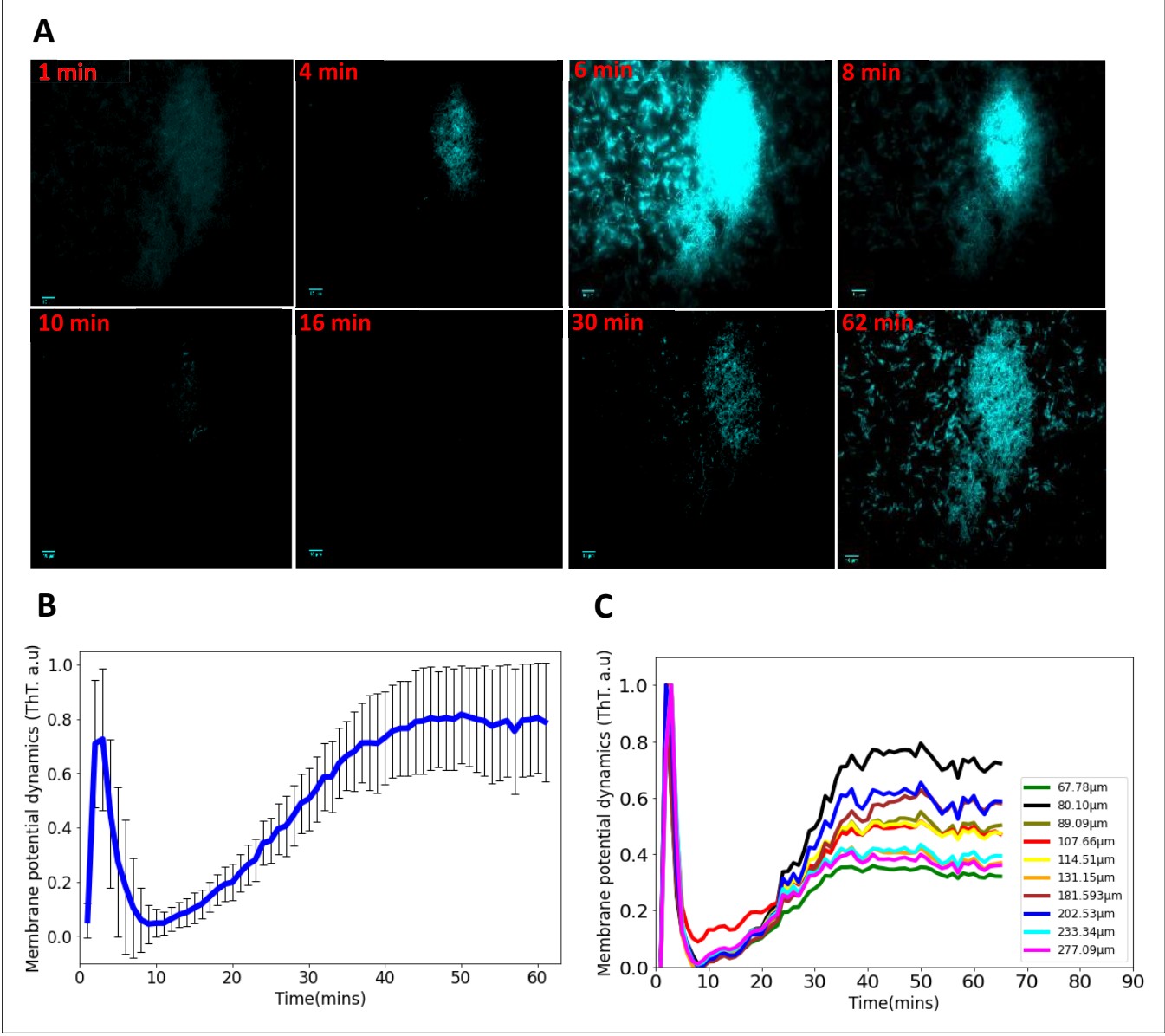

**Figure 2.** Synchronized ion-channel-mediated wavefronts in *E. coli* biofilm. (**A**) Representative fluorescence microscopy image as a function of time (1–62 min). Robust global wavefronts can be seen in an *E. coli* biofilm with ThT. The scale bars for all the images are 10 µm. (**B**) Global averaged intensity wavefront trace obtained from a 2D section of a biofilm as a function of time (mean ± *SD* for 30 biofilms from at least three experiments). (**C**) Globally averaged ion-channel-mediated dynamics in *E. coli* biofilms for different sized biofilms (68–277 µm). ThT intensity is shown as a function of time.

the continued external light stimulation. After a few minutes of inactivity, the wavefront reemerged from the center of the biofilm and slowly reached the periphery of the biofilm. After reaching the edges for the second time, the hyperpolarization persisted, and the entire biofilm remained bright and showed no noticeable change in response to the continued presence of the external stimuli. The latency of the first peak was 2.73 ± 0.85 ± 0.15 min (mean ± *SD* ± SE). This was a much smaller period than that of cells in microclusters and sparse cells (*Figure 1B and D*, *Appendix 1—figure 3B*). Furthermore, the variability of the latency (*SD*) was much lower.

Biofilms of different shapes and sizes were grown and we observed similar intensity profiles for all the biofilms (*Figure 2C*). The peaks of the spiking profiles in all the biofilms (*Figure 2C*) show that the amplitude of the action potentials does not depend on biofilm density or areal coverage of the biofilm. Consequently, we focused our analyses on time-related properties of the wavefront profiles. Action potentials in eukaryotic organisms are stereotyped events; therefore, the time-dependent

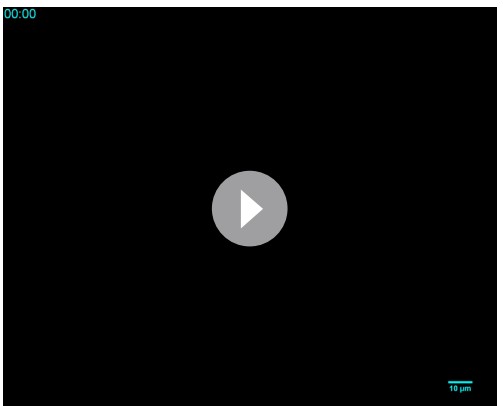

**Video 3.** Fluorescence microscopy images of ThT in an *E. coli* biofilm irradiated with blue light.

https://elifesciences.org/articles/92525/figures#video3

properties of the spikes carry the information about the amplitudes of external stimuli (*Buzsáki, 2009*). The intensity profile of different biofilms across several experiments shows that the wave-front dynamics is robust once the biofilm has an appreciable size (*Figure 2C*). These data provide evidence that coordinated signaling directs ion-channel-mediated wavefronts in *E. coli* biofilms. This data suggests that *E. coli* biofilms use electrical signaling to coordinate long-range responses to light stress.

## Voltage-gated Kch potassium channels mediate ion-channel electrical oscillations in *E. coli*

We hypothesized that the potassium channel, Kch (*Milkman, 1994*), mediates the ion-channel membrane potential dynamics in *E. coli*. This ion channel (*Figure 3A*) helps *E. coli* to survive environmental stress (*Loukin et al., 2005*), but its deletion does not impede the development of *E. coli* from single cells into biofilms (*Loukin et al., 2005*; *Kuo et al., 2005*). Kch had not been previously linked to action potentials and electrical signaling in *E. coli* biofilms (*Kuo et al., 2003*; *Beagle and Lockless, 2021*).

We applied light stimulation to a Δ*kch* mutant of strain BW25113 from the Keio collection *Baba et al., 2006* and saw a fast burst of membrane hyperpolarization identical to the wild-type, but there was a plateau that remained for the whole duration of the experiment (*Appendix 1—figure 3C*, *Video 4*). There was no repolarization or slow rise to the second peak seen in the wildtype (*Figure 1B*

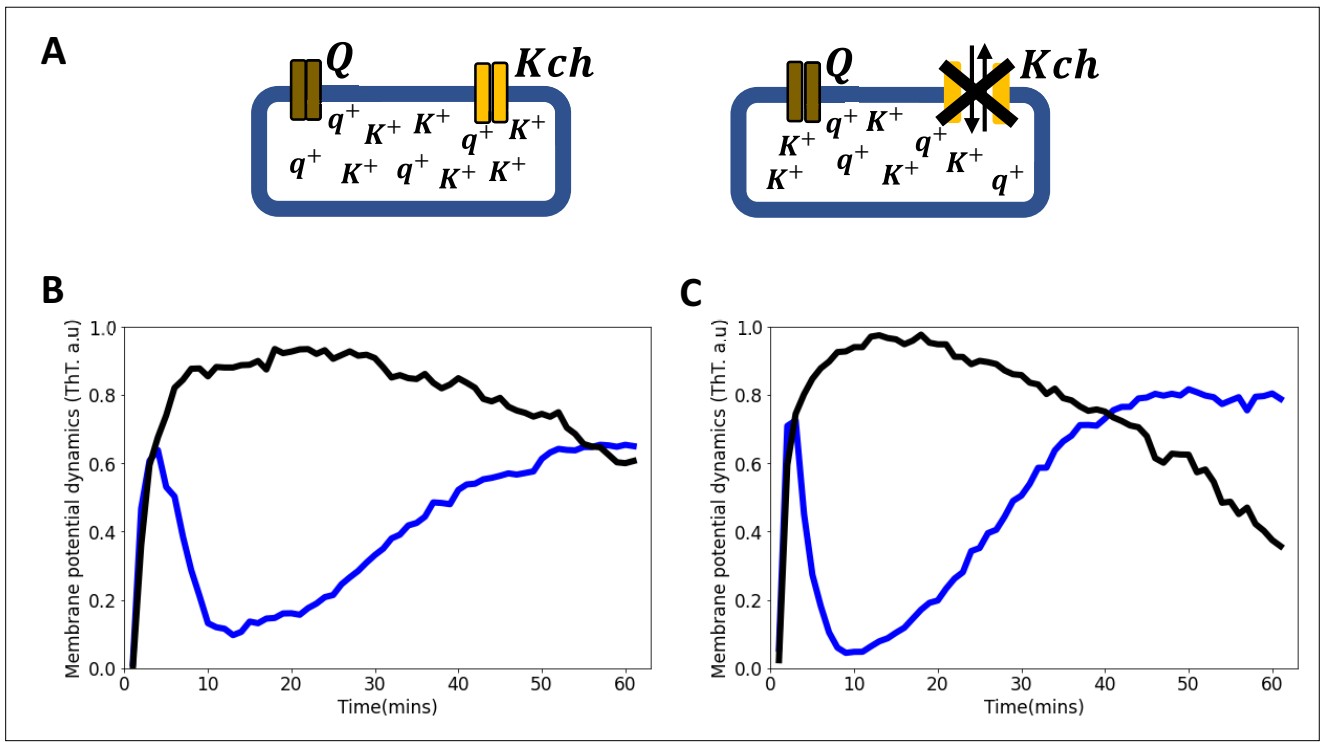

**Figure 3.** Voltage-gated Kch potassium channel mediates ion channel membrane potential dynamics in *E. coli*. (**A**) Schematic diagram showing the deletion of the voltage-gated *Kch* channel in *E. coli*. (**B**) ThT fluorescence shown as a function of time of irradiation. Deletion of *Kch* inactivates the second peak in *single cell E. coli* DH5α. Data is a mean from 52 single cells from three experimental replicates per time point for DH5α Δ*kch* mutant (black) plotted against the wildtype *single cell E. coli* DH5α (blue). (**C**) Deletion of *kch* also inactivates the second peak in *E. coli* biofilms. Data is shown for global membrane potential dynamics for biofilms grown from *E. coli* DH5α Δ*kch* mutant (black) and wildtype DH5α (blue).

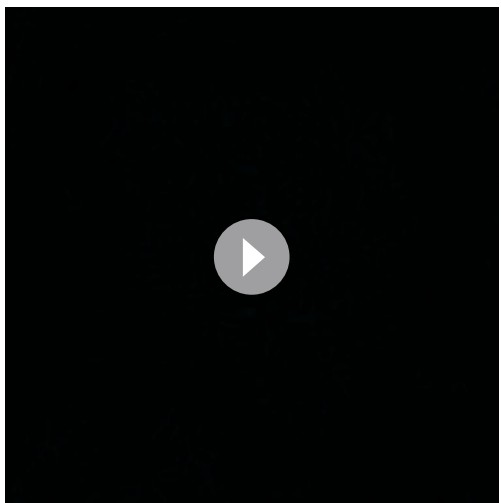

**Video 4.** Fluorescence microscopy images of ThT in knockdown mutants of voltage-gated Kch-potassium channel in *E. coli* irradiated with blue light.
https://elifesciences.org/articles/92525/figures#video4

*and D*). This suggests that the $K^+$ ion channel Kch plays a role in the refractoriness and habituation of the dynamics but does not control the initial hyperpolarization event. Using P1-phage transduction, we moved the *kch* mutation into strain DH5α and confirmed the phenotype with the first peak but no second peak (*Figure 3B and C*, *Video 4*). These data showed that Kch potassium ion channels are important for electrical signaling in *E. coli* in the presence of blue light stress.

To validate the importance of the Kch channel in the membrane potential dynamics of *E. coli*, we complemented the *kch* mutation by introducing a plasmid that carries a cloned functional kch gene into strain Δ*kch*-DH5α. The *kch* complemented strain displayed the same membrane potential dynamics (*Appendix 1—figure 3D*) observed in the wildtype (*Figure 1B and D*). However, the quiescence period in our *kch* complemented strain was reduced compared with the wildtype, presumably due to an increased degree of expression of the *kch* gene on a multi-copy plasmid.

## Blue light influences ion-channel-mediated membrane potential events in *E. coli*

To investigate the effect of irradiance on ion-channel-mediated signaling, biofilms were exposed to blue LED light at different irradiances (*Figure 4A*). For all the irradiances, we observed a first peak in the ThT fluorescence (*Figure 4A*). The time to the first peak decreased as the light irradiance was increased (*Figure 4B*). The fast burst of hyperpolarization and repolarization only occurred above the threshold of 15.99 μW/mm². For irradiances above this threshold, the dynamics exhibited progressively faster hyperpolarization to the second peak with increased light, which was not observed for irradiances below the threshold.

We thereafter examined the extracellular changes in the potassium ion, $K^+$ ions, within regions close to biofilms. We used the yellow-green fluorescent potassium ($K^+$) indicator, ION potassium Green-4 (IPG-4), which can track changes in extracellular concentrations of potassium. We observed a sharp rise, a quiescence period and then a plateau similar to that of the ThT dynamics (*Figure 4C*). This suggests that $K^+$ ions play a vital role in the observed membrane potential dynamics of the biofilm.

We validated our model with a LiveDead assay on the wildtype *E. coli* DH5α and a DH5α Δ*kch* mutant. Viable cells were monitored using propidium iodide (PI; Methods). Damaged cells appear red and we observed that 1.7% of cells stained red after 60 min of light exposure for the DH5α *Figure 4D* (1) and 7.6% for the DH5α Δ*kch* mutant (*Figure 4D*) (2). Student t-test showed that blue light stress damage to the kch-mutant is significant when compared with the DH5α (*Figure 4E*). This data shows that wildtype DH5α cells which engage in full membrane potential dynamics better withstand the light stress and have lower lethal damage.

We used the ROS scavenger, catalase, to accelerate the removal of ROS. After the addition of the catalase, cells only registered the presence of the light (via the first peak), but aborted the process of repolarization, so the second hyperpolarization event does not occur (*Figure 4F*, *Video 5*). This demonstrates that the ion-channel-mediated membrane potential dynamics is a light stress relief process.

## Development of a Hodgkin-Huxley model for the observed membrane potential dynamics

Our data provide evidence that *E. coli* manages light stress through well-controlled modulation of its membrane potential dynamics. The light-induced ion-channel-mediated dynamics present at the single cell level become more coordinated at the biofilm level (*Videos 1 and 3*).

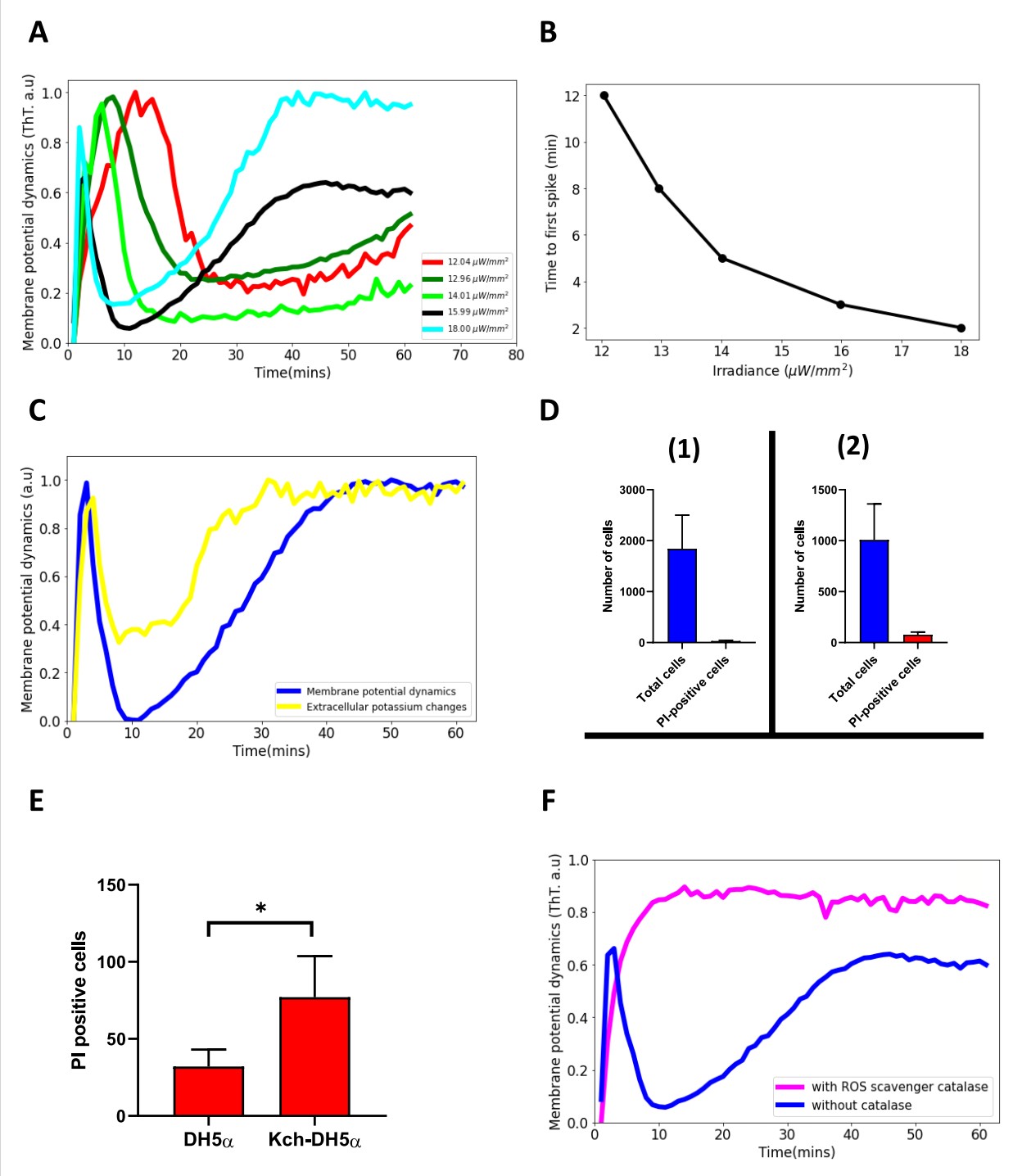

**Figure 4.** Blue light influences ion-channel-mediated membrane potential events in *E. coli*. (**A**) ThT intensity as a function of time when irradiated with different powers of 440 nm light. The time to the second excitation peak is dependent on the power. All subsequent experiments were done at the irradiance value of 15.99 µW/mm². (**B**) Time to first spike plotted as a function of irradiance. Blue-light irradiance affects the time to the first peak in *E. coli* biofilm. (**C**) Measurement of extracellular potassium changes for regions close to biofilms as a function of time using fluorescence microscopy. (**D**) LiveDead Assay using the accumulation of propidium iodide in cells (1) DH5α (n=1842) (2) DH5α Δ*kch* mutant (n=1008). (**E**) Comparison between PI-positive cells for the DH5α and the DH5α Δ*kch* mutant. Statistical significance was calculated using the Student's t-test, p≤0.05. (**F**) ThT fluorescence intensity as a function of time for cells in the presence of a ROS scavenger. *E. coli* cells employ ion-channel-mediated dynamics to manage ROS-induced stress linked to light irradiation. Data was obtained from not less than three experiments.

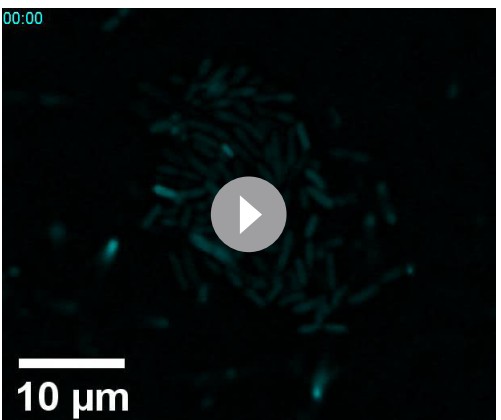

00:00

10 μm

**Video 5.** Fluorescence microscopy images of ThT in an *E. coli* biofilm irradiated with blue light in the presence of ROS scavenger, catalase.

https://elifesciences.org/articles/92525/figures#video5

To understand the biophysical mechanism of the ion channel opening for a single cell, we developed an electrophysiological model. Our Hodgkin-Huxley conductance model predicts that the membrane potential dynamics in *E. coli* biofilms are due to cooperative signaling between two distinct positively charged ion channels (Q and Kch) whose conductivities are voltage gated. We propose that the dynamics causes a process of long-range electrical communication of light stress in the *E. coli* biofilm. We also propose that the source of the photooxidative stress was due to increasing reactive oxygen species (ROS) in the vicinity of cells which gradually builds up as the light stress persists.

We predicted that the ion-channels activate and deactivate differently under the light stress (*Figure 5A, B*). Ion channel Q activates faster than the Kch channel, but deactivates slower than the Kch channel. Hence, while the Q channel activation dynamics is more pronounced for the sharp spike of the first peak, the Kch channel controls its subsequent decay. After the first action potential, the Q channel inactivates and contributes minimally to the dynamics. The Kch ion channel then controls the slow refractoriness and plateau which persists for a longer period in the presence of constant light stress (*Figure 5B*). This prediction supports our results from the deletion of the Kch ion channel from *E. coli* strains (*Figure 3*).

Our two ion-channel electrophysiological model correctly produced the same profile (*Figure 5C*) as the experimental data (*Figure 5D*). This model also predicts that the two spikes perform different roles in *E. coli*. The first spike registers the presence of light stress in the environment, while the second spike modulates the light stress by keeping the cell dynamics robust to the intensity of the external light stress. This mode of signaling is like a specialized type of electrical signaling in neurons called *habituation* (*Figure 6A and B*). Sensory neurons can engage in signal habituation to remain unresponsive to an external unwanted signal in the environment and still engage in control of other stimuli (*Herman, 2013*; *Avery et al., 2021*; *Wu et al., 2020*). The model also predicts that the opening of the ion channels creates an increased concentration level of the extracellular ions which subsequently results in the depolarization of neighboring cells (*Yang et al., 2020*; *Martinez Corral et al., 2019*; *Liu et al., 2017*).

We hypothesized that *E. coli* not only modulates the light-induced stress but also handles the increase of the ROS by adjusting the profile of the membrane potential dynamics. We therefore varied the ROS stress production coefficient at different levels of light in the model. We observed a noticeable change in the membrane potential dynamics. With reduced ROS, the first spike became sharper and the quiescent time lasted longer than previously, with the second peak occurring at much higher intensities of light. With increased ROS, the first spike lasted less than 30 s and the second spike plateau rose to a much higher fluorescence value. This result agrees with our hypothesis and further authenticates the involvement of two channels in the membrane potential dynamics of *E. coli*.

## Mechanosensitive ion channels (MS) are vital for the first hyperpolarization event in *E. coli*

We hypothesized that the first hyperpolarization event is linked to the voltage-gated calcium channels (VGGCs), so we introduced the fluorescent calcium sensor, GCAM6f *Bruni and Kralj, 2020* on a plasmid into the wildtype DH5α strain. When exposed to light stimulation, the spike events observed were consistent with stress-induced signaling of the VGCCs (*Bruni et al., 2017*; *Bruni and Kralj, 2020*). However, the calcium transients imply that the VGGCs (*Appendix 1—figure 3E*) do not play a role in the first peak of the *E. coli* strain under light stress (*Figures 1B, D and 2B*).

The MS ion channels help maintain cell turgidity and are also sensitive to stress-related voltage changes (*Li et al., 2002*; *Booth, 2014*; *Haswell et al., 2011*; *Martinac et al., 2008*; *van den Berg*

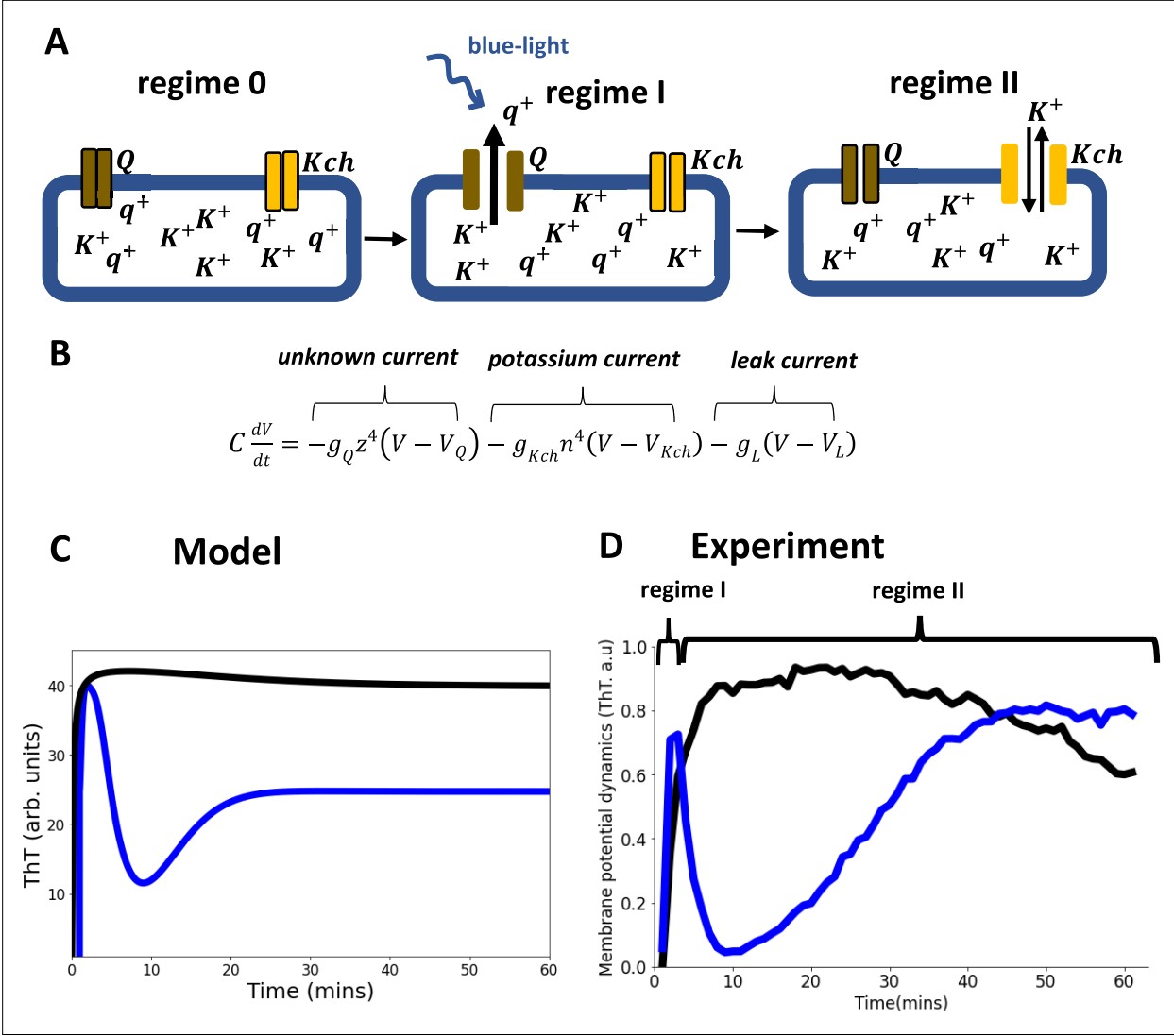

**Figure 5.** Model of ion-channel-mediated membrane potential in *E. coli*, predictions and experimental validation. (**A**) Schematic diagram of the conductance model and its predictions. The model consists of two ion-channel gates. The first channel (bronze, Q) is unknown. The second channel is the potassium channel, Kch (yellow). At the onset regime 0, both ion channels are closed. Exposure to light stress results in a rapid opening of the *Q* channel, which has a faster-opening gating variable than the Kch channel (regime I). The *Q* channel has little contribution to the repolarization event, hence the overlap of regimes I and II. (**B**) In the Hodgkin Huxley type conductance model the current changes are modulated by the two ion channels (*Q* and *Kch*) and the leakage channel (*L*). (**C**) The predicted ThT fluorescence intensity as a function of time for the Hodgkin Huxley model. Our Hodgkin Huxley model correctly reproduces the *E. coli* membrane potential dynamics for the wildtype (blue) and kch-mutants (black). The wildtype has two hyperpolarization events. (**D**) Fluorescence intensity from our microscopy experiments with ThT as a function of time for the wildtype (blue) and *Kch*-mutants (black).

*et al., 2016*). We tested whether the MS ion channels in *E. coli*, MscK, MscL and MscS, play a role in the first spike of the membrane potential dynamics (*Figures 1B, D and 2B*). We exposed the mutant strains, ΔMscK, ΔMscS, and ΔMscL of the wildtype, *E. coli* BW25113 (*Figure 6C*), to light stimulation and observed no spike dynamics typically observed with the wildtype cells (*Figures 1B, D and 2B*). Using P1-phage transduction, we transferred the MS channel mutations into the strain DH5α and confirmed the phenotype (*Figure 6D*).

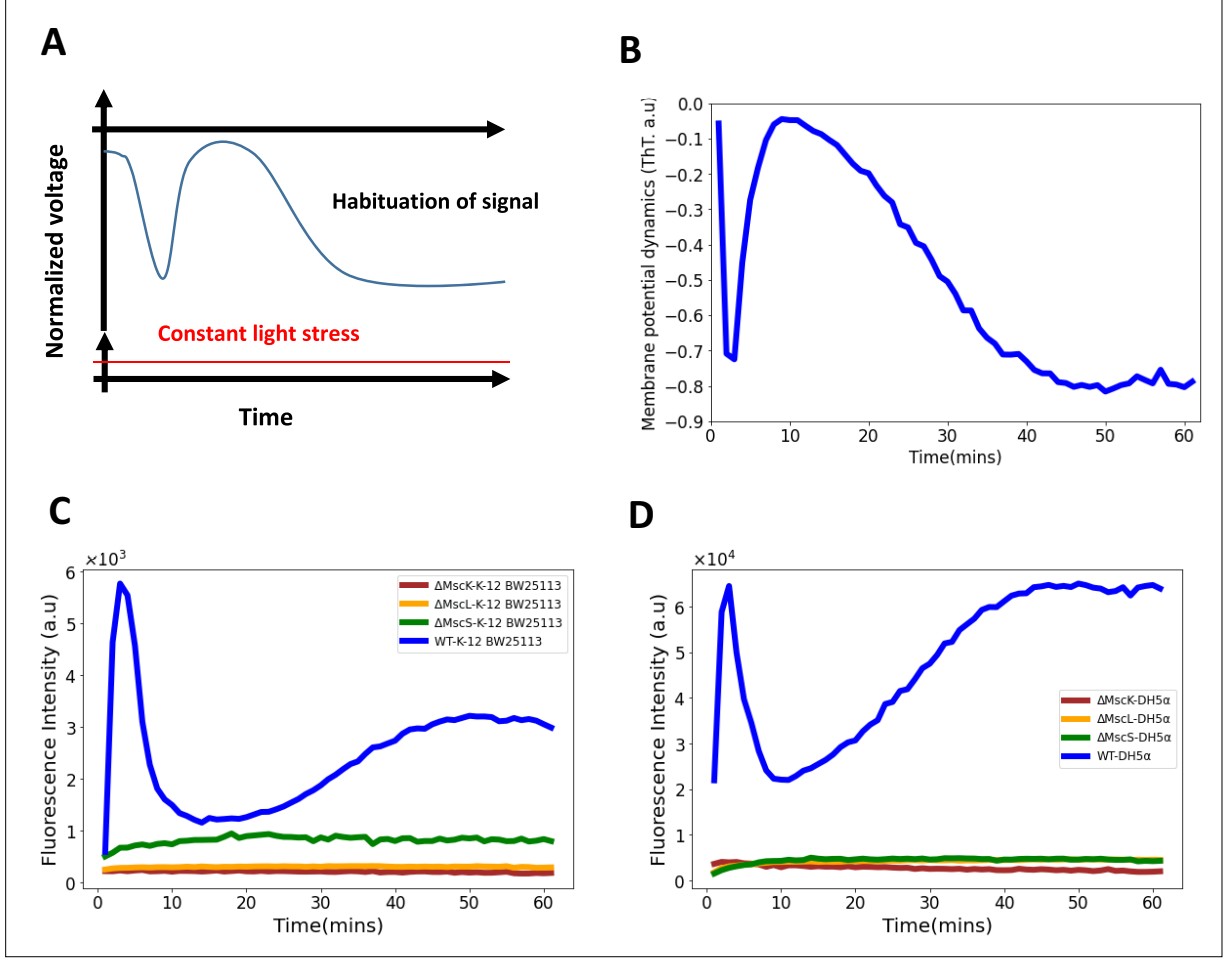

**Figure 6.** Role of mechananosensitive channels in the first hyperpolarization event in *E. coli*. (**A**) A generic diagram for the membrane voltage during neuronal habituation to a constant stimulus e.g light stress (*Avery et al., 2021*; *Wu et al., 2020*). (**B**) An illustrative diagram of membrane potential dynamics of our experiment as a function of time which is a mirror image of the ThT dynamics for comparison with (**A**). (**C**) Membrane potential dynamics for MS mutants of the wildtype, *E. coli* strain BW25113. (**D**) Membrane potential dynamics for MS mutants of the wildtype, *E. coli* DH5α.

## Anomalous ion-channel-mediated wavefronts propagate light stress signals in 3D *E. coli* biofilms

We developed a 3D agent-based fire-diffuse-fire model (ABFDF) using BSim (*Gorochowski et al., 2012*). No analytical solutions are known for the FDF model in 3D, so simulations using agent-based models were needed. In our simulated 3D spherical biofilm (*Figure 7A*), we observed global membrane potential dynamics (*Figure 7B*, *Video 6*) that are like our experimental data (*Figure 2B*).

To understand the nature of the wavefront motion in the two phases of the first peak (outward followed by inward motion), the relationship between the radial distance and the time was determined. *Figure 7C* shows the square radial displacement versus time. Data were fitted with power laws,

$$R\left(t\right)^2 = R_c^2 + bt^\gamma, \tag{1}$$

where $R\left(t\right)^2$ is the square radial distance of the wavefront, $R_c$ is the critical biofilm size for wavefront initiation, $t$ is the time, $b$ is a constant and $\gamma$ is the anomalous exponent. The exponent $\gamma$ describes whether the wave motion is *diffusive* $(\gamma = 1)$, *subdiffusive* $(\gamma < 1)$, *superdiffusive subballistic* $(1 < \gamma < 2)$, *ballistic* $(\gamma = 2)$ or *super-ballistic* $(\gamma > 2)$ (*Alves et al., 2016*; *Woringer et al., 2020*).

All ABFDF simulations produced superdiffusive subballistic behavior for the wavefront from the core to the periphery (*centrifugal wave*) $(\gamma = 1.21 \pm 0.12)$, whereas the periphery to the core (*centripetal wave*) was super-ballistic $(\gamma = 2.26 \pm 0.31)$ (*Figure 7C*).

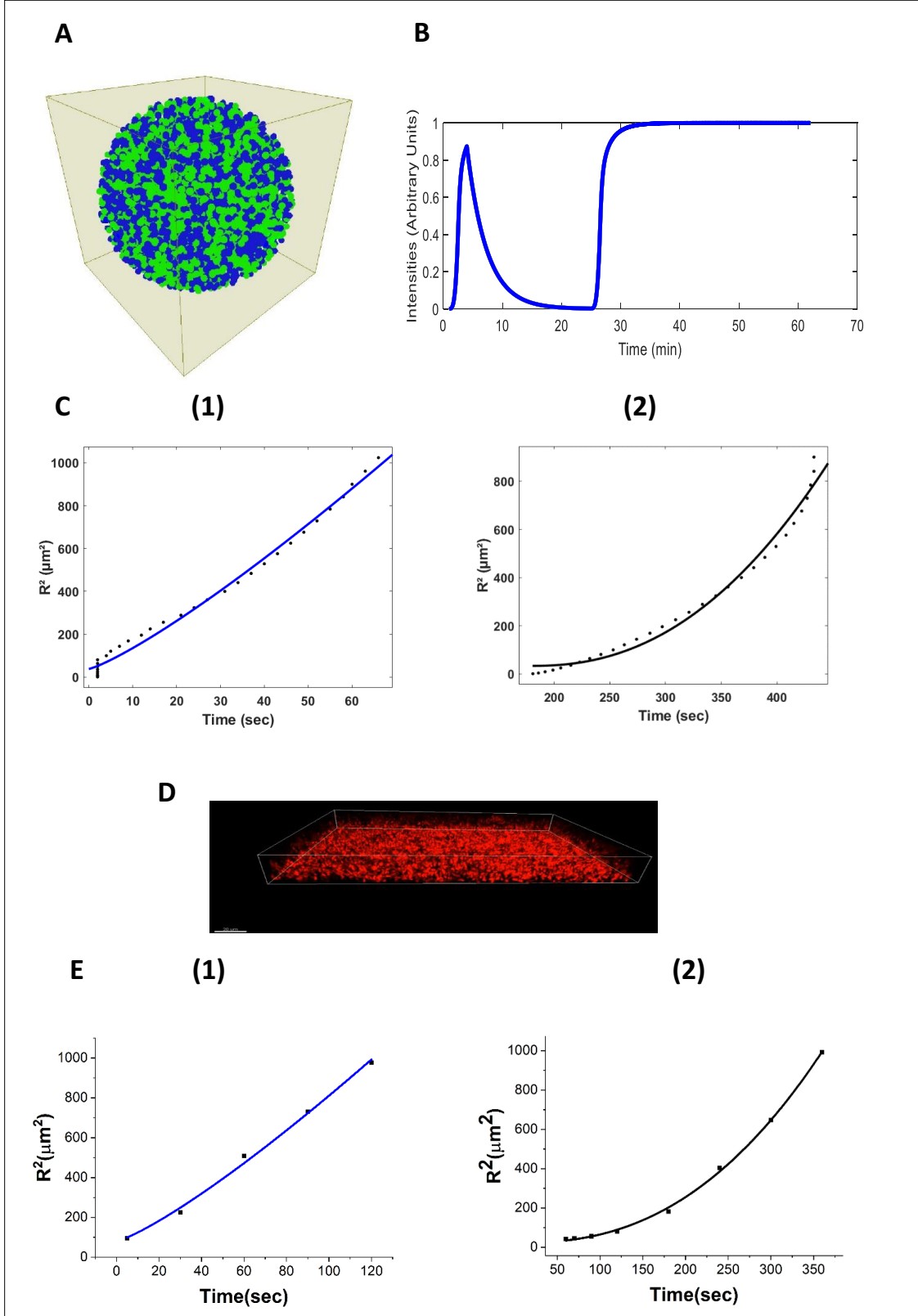

**Figure 7.** Agent-based fire-diffuse-fire model (ABFDF) and experimental validation of anomalous ion-channel-mediated wave propagation in three-dimensional *E. coli* biofilm. (**A**) 3D spherical biofilm in a fluid-filled environment simulated using BSim. (**B**) ABFDF global electrical signaling wavefront profile averaged over a three-dimensional biofilm. The ThT intensity is predicted as a function of time. (**C**) Plot of the square radial distance of the wavefront ($R^2$) against time and fit with a power law, $R\left(t\right)^2 = R_c^2 + bt^\gamma$. For the first peak's (1) centrifugal motion: $\gamma = 1.21 \pm 0.12$ and (2) centripetal

*Figure 7 continued on next page*

*Figure 7 continued*

motion: $\gamma = 2.26 \pm 0.31$ from ABFDF simulation data. (**D**) Representative confocal fluorescence image for a sessile 3D biofilm containing ThT (Scale bar =20 µm). (**E**) Plot of $R^2$ against time fit with a power law, $R\left(t\right)^2 = R_c^2 + bt^\gamma$ for the first peak's (1) centrifugal motion: $\gamma = 1.22 \pm 0.15$ and (2) centripetal motion: $\gamma = 2.43 \pm 0.08$ from the experimental data.

We experimentally tested these simulation findings using confocal microscopy and ThT. We grew a three-dimensional biofilm (126 µm x 172 µm x 31.8 µm) and exposed it to blue light (*Figure 7D*). A timelapse of the sagittal section of the three-dimensional biofilm (*Appendix 1—figure 4A*) reveals membrane potential dynamic akin to the 2D sections through the biofilms (*Figure 2A*). The spatiotemporal membrane potential dynamics of the 3D biofilm (*Appendix 1—figure 4B*, *Video 7*) was similar to our simulation results (*Figure 6B*).

Wavefronts propagating in three-dimensional systems emanating from a point source have a curved geometry (*Keener and Sneyd, 2009*). We tested if the ion-channel wave propagates along the z-axis, adopting the z-plane analysis scheme (*Wussling and Salz, 1996*; *Lipp and Niggli, 1993*; *Wier and Blatter, 1991*). Our experimental data (*Figure 7E*) showed that the centrifugal wave is super-diffusive subballistic ($\gamma = 1.22 \pm 0.15$), while the wave motion for the centripetal wave is super-ballistic ($\gamma = 2.43 \pm 0.08$), in reasonable agreement with simulation. Furthermore, the results confirm that curvature affects the motion of the wavefronts (*Nagy Ungvarai et al., 1992*). Blee and co-workers (*Blee et al., 2019*) previously observed a superdiffusive wave motion for both the centrifugal ($\gamma = 1.42 \pm 0.06$) and centripetal phases ($\gamma = 1.79 \pm 0.03$) of the potassium wavefront in 2D *B. subtilis* biofilm. The centripetal wavefronts appear to travel faster than the centrifugal wavefronts.

Using *eqn*1 we calculated the critical size for wave initiation in 3D *E. coli* biofilms from the experiments to be 4.71±0.98 µm. This is reasonably close to the value predicted from our ABFDF model, 6.17±1.84 µm. 3D *E. coli* biofilms, therefore, need to develop a densely packed biofilm above the critical radius for a robust synchronized ion-channel-mediated wavefront to propagate in the system. This contrasts with 2D *B subtilis* biofilms which need to grow up to 350 µm for a wavefront to emerge in the system (*Prindle et al., 2015*; *Martinez-Corral et al., 2018*; *Liu et al., 2015*). Therefore, our model predicts the transport properties of the wavefront, the patterns of global excitation and the critical radius for wavefront propagation in biofilms (*Table 1*).

As expected, we observed a slow decrease of velocity as the wavefront spread from the core towards the periphery that is predicted by the Eikonal approximation. For the velocity of the wavefront that travels back to the core, we observed a decrease and subsequently a sharp increase at distances close to the core of the biofilm (*Figure 8A and B*). This unexpected behavior may be linked to the heterogeneity of microclusters within the bacterial biofilms. A nonlinear relationship is also observed between the wavefront velocity and curvature for centrifugal and centripetal wavefronts (*Appendix 1—figure 5A, B and C*).

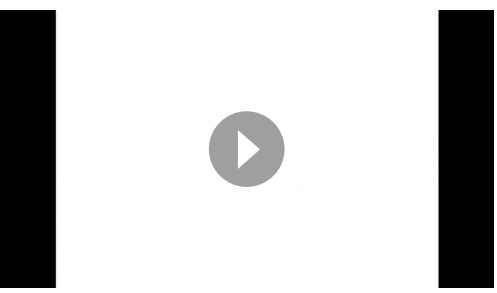

**Video 6.** 3D simulation of electrical signalling in bacterial biofilms via the fire-diffuse-fire agent based model.

https://elifesciences.org/articles/92525/figures#video6

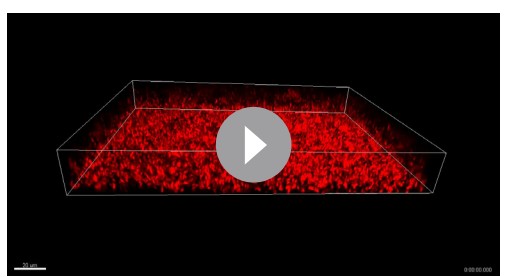

**Video 7.** 3D confocal fluorescence microscopy images of ThT in an E. coli biofilm irradiated with blue light.

https://elifesciences.org/articles/92525/figures#video7

**Table 1.** Fit constants for *Equation 1* to results from the ABFDF and experimental data.

| Constants | Symbol | Centrifugal wavefront Model | Centripetal wavefront Model | Centrifugal wavefront Experiment | Centripetal wavefront Experiment |
|---|---|---|---|---|---|
| The anomalous exponent | γ | 1.21 ± 0.12 | 2.26 ± 0.31 | 1.22 ± 0.15 | 2.43 ± 0.08 |
| The critical biofilm size | $R_c$ | 6.17 ± 1.84 μm | – | 4.71 ± 0.98 μm | – |

## Discussion

*E. coli* biofilms synchronize ion-channel-mediated electrical signaling when under external light stress. The process of communicating the stress becomes faster as the intercellular distance decreases and it results in robust wavefront dynamics in which bacteria take turns to spike in a coordinated manner.

Our experimental data reveals that ion-channel-mediated wavefronts exist in 3D *E. coli* biofilms. 3D wavefronts exhibit anomalous diffusive behavior, which was well described by our 3D ABFDF model. The mode of propagation of the wavefronts was like that described for *B. subtilis* under nutrient stress (*Prindle et al., 2015*). However, noticeable differences are in the frequency of oscillations, the latency, the dimensionality of the system, and the number of spikes (two hyperpolarization events with *E. coli*).

Although Kch was the first potassium channel to experience detailed structural work in *E. coli Milkman, 1994*, it has never been linked to membrane potential dynamics. Our findings establish that the Kch channel plays an important role in *E. coli* membrane potential dynamics. Specifically, the channels control the refractoriness and second peak of the membrane potential. These phases

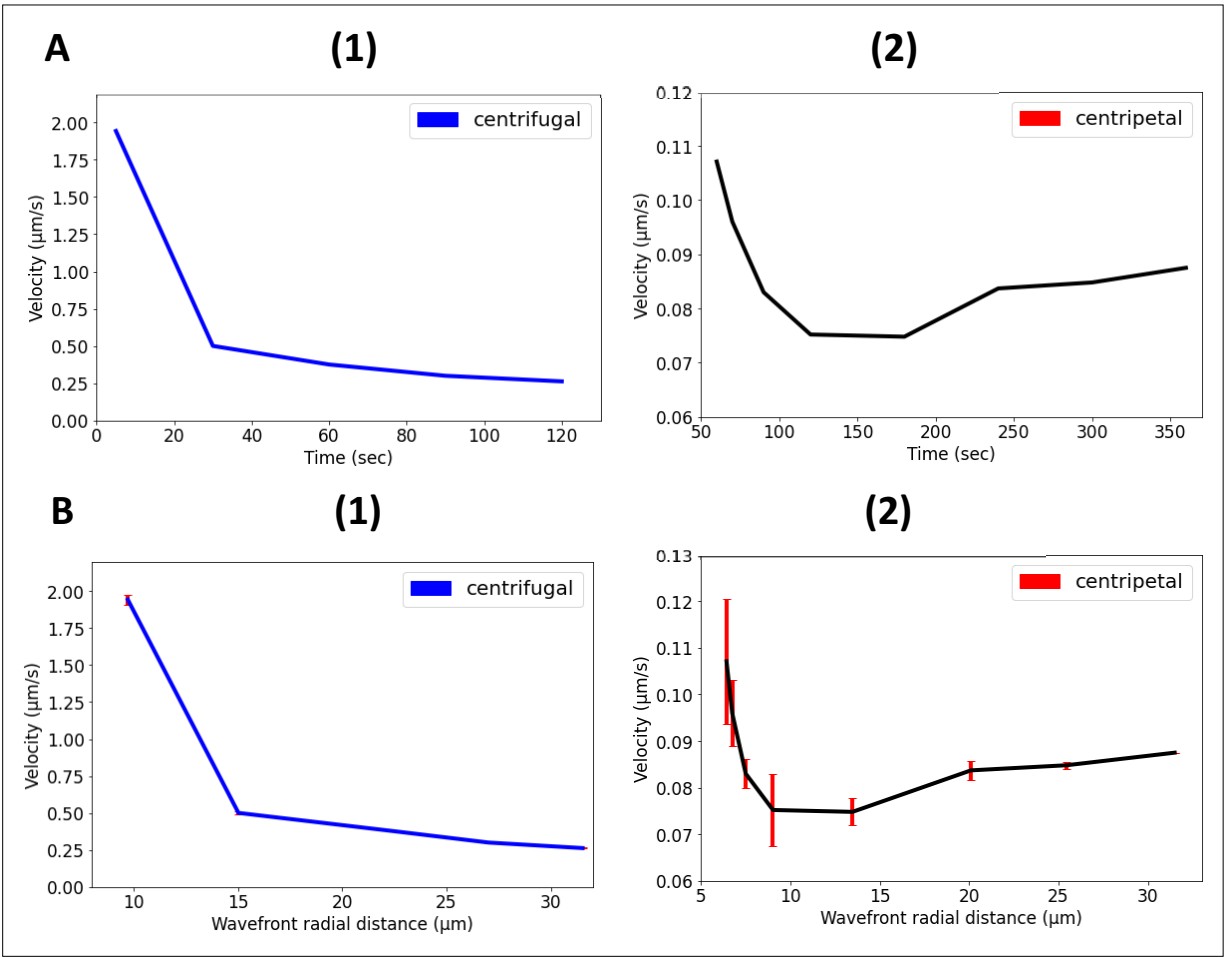

**Figure 8.** Nonlinear propagation of ion-channel-mediated wave in 3D *E. coli* biofilms. (**A**) Nonlinear relationship between propagation velocity of the wavefront and the time for (1) centrifugal wave and (2) centripetal wave of the first peak. (**B**) Nonlinear relationship between propagation velocity of the wavefront and the radial distance for (1) centrifugal wave and (2) centripetal wave of the first peak.

of the dynamics were correlated with light stress modulation in *E. coli* biofilm. We therefore predict that light-based *E. coli* biofilm treatments should be more effective if coupled with Kch targeting modalities.

Both of the two models developed, the Hodgkin-Huxley (HH) model and the fire-diffuse-fire agent-based model, provided new physical insights to understand photodynamic therapy. The HH model satisfactorily describes the globally averaged dynamics of the membrane potential in response to blue light and indicated a minimal model containing at least two ion channels (*Q* and *Kch*) was necessary to describe the double spiking response to blue light irradiation. The fire-diffuse-fire agent-based model could quantify the anomalous dynamics of the wavefront and predicted the critical biofilm radius needed for wavefronts to propagate. Anomalous dynamics of wavefronts are not predicted by classical analytic solutions to reaction-diffusion equations and are thus a substantial advantage of the agent-based modeling approach we followed *Keener and Sneyd, 2009*. Future extensions of the theoretical models will be to predict the form of the wavefronts as they propagate in more complex geometries, for example through mushroom-shaped biofilms and around microchannels which are known to occur with *E. coli* biofilms (*Rooney et al., 2020*). Research on cardiac infarctions have extensively studied the effects of dead non-signaling tissue on propagating electrochemical wavefronts *Keener and Sneyd, 2009* to understand the effects of drug treatments on the pumping of the diseased heart. It would be interesting to consider the analogous system for electrical signaling in biofilms, for example biofilms with defect structures due to inanimate inclusions or mixed species biofilms.

Our data shows that the unknown ion channel (*Q*) is not connected to calcium dynamics, but it is connected with the MS ion channels that work in tandem to propagate the initial spike in *E. coli* under light stress. Minimally the model predicts $Q = MscK \times MscL \times MscS$, but the full dependence is probably more complex. The initial spike is key to registering the presence of the light stress. More work is needed to unravel the detailed molecular mechanism linking the MS channels and light stress gating in *E. coli*, but the data demonstrate a role for MS channels beyond osmo-protection.

When bacterial cells experience light-based stimulation, they experience stress which can be cytotoxic (*Wilson and Patterson, 2008*; *Abana et al., 2017*) due to the production of ROS. The accumulation of ROS leads to deleterious oxidative stress that damages molecules (*Wilson and Patterson, 2008*; *Mittler, 2002*). The propagation of action potentials in the presence of changing irradiance suggests that the stimulus strength is encoded in the response of the biofilms. The membrane potential dynamics also suggest a link between the duration of the stimulus, the oxidative stress due to the strength of the irradiance and the gating of the ion channels that modulate the dynamics.

The unresponsive nature of *E. coli* after the second hyperpolarization event and its marked plateau voltage is reminiscent of the phenomenon of signal habituation in neurons (*Levitan and Kaczmarek, 2002*). Neurons discriminate between external stimuli by observing a sustained decrement in response to a constant external stimulus (*Herman, 2013*; *Avery et al., 2021*; *Wu et al., 2020*). *E. coli* biofilms mostly switch to a viable, but non-culturable phenotype when exposed to either pulsed or constant blue-light therapy (*Abana et al., 2017*; *Kim et al., 2018*; *Marasini et al., 2021*). We suggest that the habituation after light stress registration could be involved.

Our work shows that the ion-channel mediated long-range electrical membrane potential in *E. coli* biofilms help them to withstand light stress. We believe that our findings will provide a good framework for more detailed optogenetic studies of membrane potential signaling in *E. coli* biofilms and help inform photodynamic therapies to combat problematic biofilm infections.

Recent work indicates that waves of hyperpolarization also occur across biofilms of *Neisseria gonorrhoeae Hennes et al., 2023*; another example of electrophysiological phenomena in medically relevant biofilms. The signaling wavefronts were thought to be due to diffusion of ROS rather than the potassium ions observed with *E. coli* biofilms, but genetic experiments with ion channel mutants to clarify this issue were not performed. The *N. gonorrhoeae* study used an analytic reaction-diffusion model to describe the signaling phenomena. We have presented much better quantitative agreement of our model with the propagating wavefronts in *E. coli* biofilms using reaction-diffusion equations combined with agent-based modeling, for example the ABM FDF model is able to predict the anomalous dynamics of the wavefronts and the critical biofilm size for propagation of the wavefront. Such agent-based modeling should be applied to *N. gonorrhoeae* biofilms in the future due to the better handling of geometric constraints in the models.

The electrophysiological properties of bacteria is a rapidly evolving area (*Benarroch and Asally, 2020*). Connecting the membrane potential to photodynamic therapy implies that there will be synergistic effects of externally applied electric fields and blue light on bacterial infections. Thus, conducting dressings could be combined with blue light to improve the efficacy of wound treatments (*Yu et al., 2021*). Furthermore, recent experiments have connected antibiotic activity to fluctuating membrane potentials with *E. coli* (*Jin, 2023*). This in turn implies that synergistic effects will occur between blue light and antibiotic treatment, for example more cationic antibiotics will be absorbed by cells that hyperpolarize in response to blue light and the performance of the antibiotics will be modulated by ion channel activity.

## Methods

### Bacterial strains

The bacterial strains and the media recipes used in this study are listed in the *Appendix 1—table 4*. All experiments were performed with the DH5α and BW25113 strains of *E. coli*. All other strains were derived from these two and are listed in *Table 1*, Methods. When genes were moved by transduction into DH5α, the resulting mutations were sequenced to confirm authenticity and that the strains did not have any additional mutations.

### Dyes and concentrations

ThT was used at a final working concentration of 10 µM for both LB and M9 media. This is the concentration that did not inhibit bacterial cell growth or influence the voltage flux in previous experiments (*Prindle et al., 2015*; *Blee et al., 2020*; *Liu et al., 2017*). Fresh ThT was made up on the day of each experiment and added to the media containing the cells.

To measure extracellular potassium, the IPG-4 AM was converted to its membrane impermeable form. This was achieved by dissolving the dye in 250 µl DMSO and subsequent addition of 0.1 M KOH.

CCCP and TMRM were used at the final concentrations of 100 µM and 100 nM, respectively. Propidium iodide (PI) was used at a final concentration of 1 µg/ml.

### Microfluidics setup and experimental design

All microfluidic experiments were performed with IBIDI uncoated glass bottom µ-Slide VI$^{0.5}$ flow cells (Thistle Scientific, UK) which have dimensions of $17\,mm \times 3.8\,mm \times 0.54\,mm$ for the length, width, and height, respectively (*Appendix 1—figure 1A and B*). The system yielded successful growth for all *E. coli* strains and allowed high-resolution microscopy images to be taken. Single cells and microclusters were also cultured in this system. The microfluidic components and software are listed in *Appendix 1—table 4*.

Growth media (LB or M9) contained in a 20 ml syringe (BD Emerald, UK) were delivered to the microfluidic wells by an Aladdin NE-1002 Programmable Syringe Pump (World Precision Instruments, UK). The media was replenished at intervals throughout the experiment. A C-Flex laboratory tubing with I.D. x O.D. 1/32 in. x 3/32 in (Sigma-Aldrich) and Elbow Luer connector male (Thistle Scientific, UK) completed the microfluidic setup. A 0.22 µm filter was installed at the syringe hub before attaching the syringe needle 0.8 × 40 mm to maintain sterility and reduce the number of air bubbles. Exchangeable components of the microfluidic setup were used only once. Experiments were conducted in more than one channel at a time for data replicates. Prior to the start of the experiments, the flow cell chambers were primed with appropriate media to achieve faster cell attachment. The microscope and all the components of the microfluidic setup were confined within the custom-built Perspex microscope chamber which was maintained at $37^{o}C$ using an Air-THERM ATX (World Precision Instrument Ltd.).

For single cell experiments, cells were left static in the microfluidic chamber for 2 hr for cells to attach to the substrate. The media was then delivered at flow rates of 3 µL/min and subsequently maintained at 5 µL/min to remove unattached cells from the system. After 1 hr of media flow, data were only collected for cells that were attached to the substrate. For biofilms, the system was left static 2–3 hr on the microscope after loading to allow the cells attach. Media flow was initiated at the rate of 5–6 µL/min and maintained for a further $\approx$ 12 hr (see Time-lapse Microscopy section).

### Cell culture and growth conditions

#### Single cell culture

Cells were streaked onto an Agar plate from $-80^{o}C$ glycerol stocks a day prior to the experiment and incubated at $37^{o}C$ overnight. The following day, 10 ml of Luria Broth (LB) in a glass universal was

inoculated with one colony of the required *E. coli* strain. The inoculum was then incubated overnight in a shaking incubator at 200 rpm at $37^oC$. The next day, $10\,\mu l$ of the inoculum was transferred to a fresh 10 ml LB and incubated in a shaking incubator at 200 rpm at $37^oC$ for 4.5 hours or $OD_{600} \approx 0.8$. The optical density of the cells was measured using a spectrometer (JENWAY, Cole-Parmer UK). $10\,\mu M$ ThT was added to the inoculum and left static for 20 min. $200\,\mu l$ of the cell suspensions were then seeded into the required chambers of the microfluidic device. The microfluidic device was mounted on the microscope after attaching the other microfluidic components, such as tubes and tube connectors. The instrument was left static for 2 hr to allow for cell attachment before the media was delivered under flow. To sustain the growth temperature at $37^oC$, a custom-built Perspex microscope chamber heated using an Air-THERM ATX (World Precision Instrument Ltd.) was employed. The salts for the M9 media are listed in *Appendix 1—table 4*. Data were collected for both sparse cells and cells existing within microclusters. The single cell experiments were done in Luria Broth (LB) media and replicated in Minimal media M9; *Appendix 1—figure 1C* to demonstrate independence on the exact media used.

For the LiveDead Assay, the same protocol was followed. However, before seeding the well with 200 µL of the inoculum, 10 µL (1 µg/mL of the stock solution) of PI was added to the 10 ml universal containing cells and ThT.

For the combined CCCP experiments, the same protocol was also followed. 100 µM of CCCP was pipetted into the universal containing inoculum and ThT. This was left for 50 min, then 200 µL of the suspension was transferred into the wells and cells were exposed to blue light stress.

## Biofilm growth

The *E. coli* DH5α strain was chosen based on its ability to adhere to surfaces and to grow into biofilms (*Huang et al., 1994*; *Soleimani et al., 2013*; *Jayaraman et al., 1997*). Biofilms were grown in one of the chambers of the microfluidic devices (triplicates with control experiments). Single cell *E. coli* was cultured as described in section (I). 200 µL of bacteria culture suspensions with ThT were added and then loaded in the flow cell to initiate biofilm formation. The setup was left static for 2 hr within the microscope chamber before the media flow was initiated at a rate of 5–6 µL/min for a further 12 hr. Under sterile and constant media flow (explained in the Microfluidic section) to produce optimal growth conditions, mature sessile biofilms were observed. Our protocol was optimized to obtain mature sessile DH5α biofilm after 15 hr. The growth temperature was maintained at $37^oC$ using a custom-built Perspex microscope chamber heated using an Air-THERM ATX (World Precision Instrument Ltd.). Media replacement was done within the microscope chamber to maintain sterility and avoid air bubbles.

## Time-lapse microscopy and image acquisition

Fluorescence microscopy was performed with an Olympus IX83 inverted microscope (Klaus Decon Vision) using Blue Lumencor LED excitation (illumination), a 60 x (NA 1.42 Plan Apo N) oil immersion objective and the CFP filter set (Chroma [*89000*]). Time-lapse fluorescence images were taken with a Retiga R6 CCD camera [Q-imaging]. ThT fluorescence was measured in the CFP channel using an excitation filter (Ex) 440/20 nm and an emission filter (Em) 482/25. PI fluorescence was measured with the Ex 575/20 nm and Em 641/75 nm. Images were taken every 1 min with an exposure time of 50 ms and camera gain of 3. Prior to setting up the microfluidic apparatus on the microscope, the chamber was maintained at $37^oC$ for at least 3 hr. Image acquisition on the PC was carried out using the Meta-Morph software (Molecular devices). This microscope and the settings were used for all observations on single cells and 2D biofilms. The settings were varied for the irradiance experiment (see section on Irradiance measurement below). Images in *Appendix 1—figure 1A* were obtained at different time scales (every 10 s) for comparison. The pump was turned off before the image acquisition to minimize image drift and vibrations.

Fluorescence confocal 3D image stacks for the 3D biofilm were acquired using a CSU-X1 spinning disc confocal (Yokagowa) on a Zeiss Axio-Observer Z1 microscope with a 63 x/ 1.40 Plan-Apochromat objective, Evolve EMCCD camera (Photometrics) and motorised XYZ stage (ASI). A 445 nm laser line was used for ThT excitation. The 445 nm laser was controlled using an AOTF through the Laserstack (Intelligent Imaging Innovations (3I)) allowing for rapid 'shuttering' of the laser and attenuation of the laser power. Slidebook software (3I) was used to capture images every 1 min with 100 ms exposures. Movies were analysed in Slidebook, ImageJ and Imaris (bitplane) software.

## Image and data analyses

ImageJ (National Institute of Health), MATLAB, BiofilmQ and Imaris (bitplane) software were used for image analysis. Data analyses and plots were done with Python, BiofilmQ and GraphPad (Prism). Python and Scipy package were used for mathematical modeling. Model curve fits were done with Python Scipy package and OriginPro.

### Single cells

Single cell image analysis was conducted using ImageJ (National Institute of Health). Background subtraction was done using the 'ImageJ rolling ball' background plugin with radii of 8–11 µm. This was influenced by experimental conditions, for example media. An ImageJ custom script was used for drift correction. Data were plotted with standard deviations for the error bars. Data were normalized in python for final plotting.

For LiveDead assay experiments we used the imageJ plugin 'cell counter' to identify and count cells. Only cells that are hyperpolarized were counted in the experiment as live and only cells that appeared red after the experimental duration were counted as dead. Time to first peak analysis was done by measuring the individual times for each single cell to experience the first hyperpolarization event.

### Biofilms

To overcome the challenges of diversity in structure and size of the biofilms obtained in our experiments, we conducted image analysis of mature sessile biofilms with BiofilmQ. BiofilmQ is a high-throughput MATLAB-based image processing and analysis software designed for spatiotemporal studies of both 2D and 3D biofilms. A detailed description of BiofilmQ can be found *Hartmann et al., 2021*. It has been used in a number of recent of recent investigations of biofilm (*Wucher et al., 2021*; *Díaz-Pascual et al., 2019*). We were able to accurately obtain membrane potential dynamics of biofilms by tracking ThT traces within the entire biofilm (2D and 3D confocal stacks) using this software.

To ensure data reproducibility, we describe the individual steps in our analysis. The confocal image stacks were first prepared and registered before segmentation. Image preparation included colony separation to identify the preferred microcolonies within the ROI and image alignment was used to correct image drift over time. Image segmentation was then carried out to separate cells from the background. Specifically, image cropping was done to tag the microcolonies, this was followed by denoising of the image using convolutions. We opted not to use the top Hart filter because in our biofilms the cells overlap each other. A filter kernel value of *Bruni and Kralj, 2020*; *Prindle et al., 2015* was employed for the convolution. This value of the filter kernel was used for all our data analysis. It was sufficient to only slightly blur the images and significantly reduce the image noise. To complete the segmentation process, we used the Otsu thresholding method. The sensitivity of the thresholding was set to 1 for all the biofilm analysis. We opted not to dissect the biofilm, since we were interested in the global membrane potential dynamics and ion-channel waves in the biofilms. These steps ensured reproducibility of our analysis across all the biofilms.

To verify the accuracy of this software, we also measured the temporal dynamics of the membrane potential by tracking the global ThT fluorescence of biofilms using ImageJ. To obtain ThT curves in imageJ, we used the 'plot-Z axis' function on the imageJ image analysis toolbox. This has been used in previous research work to successfully track the biofilm ThT fluorescence (*Prindle et al., 2015*; *Blee et al., 2019*). We confirmed that BiofilmQ was efficient for our data analysis.

The optical sections for the velocity measurements were obtained with a combination of both the BiofilmQ and IMARIS software. Radial distances of biofilm volumes from the substrate to the core of the biofilm were made with both BiofilmQ and IMARIS. The choice of the appropriate optical sections perpendicular to the z-slices were also made by the combination of both software.

## 440 nm and 445 nm light stimulation

*E. coli* cells and biofilms were treated with blue light using an Olympus IX83 440 nm-LED and 445 nm laser line. The cells were exposed to the light every 1 min. Experiments were performed with at least three biological replicates.

## Irradiance measurements

Irradiance experiments were conducted with the Olympus IX83 440 nm-LED. Therefore, the irradiance was varied to ascertain the effect of the light stimulation on the membrane potential dynamics of *E. coli*. A Newport power/energy meter (Newport Corporation Irvine US) was used for the irradiance measurements. The power of the 440 nm light was varied by adjusting the percentage light illumination via the CFP light, for example 175 of the 255CFP that corresponds to $2.43\,\mu W$ as measured with the power meter. Uniform illumination of the sample ROI was always maintained via Köhler illumination.

To determine the irradiance, we first measured the power of the LED light at the sample plane in $mW$. The diameter of field of the view (FOV) was then calculated by dividing the objective lens field number (FN) by its magnification. For the Klaus 60 x lens, the FN is $26.5\,mm$. The value of the diameter of the FOV was then used to calculate the area of field of view in $mm^2$. Finally, the irradiance ($I$) is given by

$$I = \frac{Power}{Area}.$$ (2)

The irradiance value of $15.99\,\mu W/mm^2$ was used for all the other experiments except when the irradiance was specifically varied (*Figure 2D*).

## Bacterial strain availability

Strains and further information should also be directed to the lead contact, Thomas Waigh (t.a.waigh@manchester.ac.uk).

## Acknowledgements

EA would like to thank Marie Goldrick for her assistance. EA would like to thank the Knut Drescher group for useful discussions on the use of BiofilmQ EA would like to thank Johanna Blee, Raveen Tank, Emma Layton and Dan Han for useful discussions. The authors would like to thank the Prindle group for providing their code. Special thanks goes to Peter March, Roger Meadows and Steven Marsden for their help with the microscopy. The Bioimaging Facility microscopes used in this study were purchased with grants from the BBSRC, Welcome Trust and the University of Manchester Strategic Fund. RK is supported by the UKRI Future Leaders fellowship (MR/T021225/1). The authors would like to thank TETFund Nigeria and Abia State University Nigeria for EA's PhD scholarship.

## Additional information

### Funding

| Funder | Grant reference number | Author |
|---|---|---|
| TETFund Nigeria | | Emmanuel Akabuogu |
| UK Research and Innovation | Future Leaders Fellowship MR/T021225/1 | Rok Krašovec |

The funders had no role in study design, data collection and interpretation, or the decision to submit the work for publication.

### Author contributions

Emmanuel Akabuogu, Conceptualization, Data curation, Formal analysis, Validation, Investigation, Visualization, Methodology, Writing – original draft, Writing – review and editing; Victor Carneiro da Cunha Martorelli, Software, Formal analysis, Validation, Visualization, Methodology; Rok Krašovec, Conceptualization, Formal analysis, Supervision, Funding acquisition, Methodology, Project administration; Ian S Roberts, Conceptualization, Resources, Formal analysis, Supervision, Visualization, Methodology, Project administration, Writing – review and editing; Thomas A Waigh, Conceptualization, Software, Formal analysis, Supervision, Funding acquisition, Methodology, Writing – original draft, Project administration, Writing – review and editing

## Author ORCIDs

Victor Carneiro da Cunha Martorelli (ID) https://orcid.org/0000-0001-5787-3142
Ian S Roberts (ID) https://orcid.org/0000-0002-0662-0214
Thomas A Waigh (ID) https://orcid.org/0000-0002-7084-559X

Reviewer #1 (Public Review): https://doi.org/10.7554/eLife.92525.3.sa1
Reviewer #2 (Public Review): https://doi.org/10.7554/eLife.92525.3.sa2
Reviewer #3 (Public Review): https://doi.org/10.7554/eLife.92525.3.sa3
Author response https://doi.org/10.7554/eLife.92525.3.sa4

## Additional files

### Supplementary files
MDAR checklist

### Data availability

The python software for the Hodgkin-Huxley simulations can be found at https://github.com/VictorCDCM/HH (copy archived at *Carneiro da Cunha Martorelli, 2025a*). The Java/Eclipse code for the agent based fire-diffuse-fire model can be found at https://github.com/VictorCDCM/FDF (copy archived at *Carneiro da Cunha Martorelli, 2025b*). More data is available in the University of Manchester PhD thesis of Dr. Emmanuel U Akabuogu (*Akabuogu, 2024*). Further, raw and analysed data (7GB) that support the findings of this study are available at https://data.mendeley.com/datasets/6knr855zx5/1.

The following dataset was generated:

| Author(s) | Year | Dataset title | Dataset URL | Database and Identifier |
|---|---|---|---|---|
| Akabuogu E, Martorelli V, Krašovec R, Roberts I, Waigh T | 2025 | Emergence of ion-channel mediated electrical oscillations in *Escherichia coli* biofilms | https://doi.org/10.17632/6knr855zx5.1 | Mendeley Data, 10.17632/6knr855zx5.1 |

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

# Appendix 1

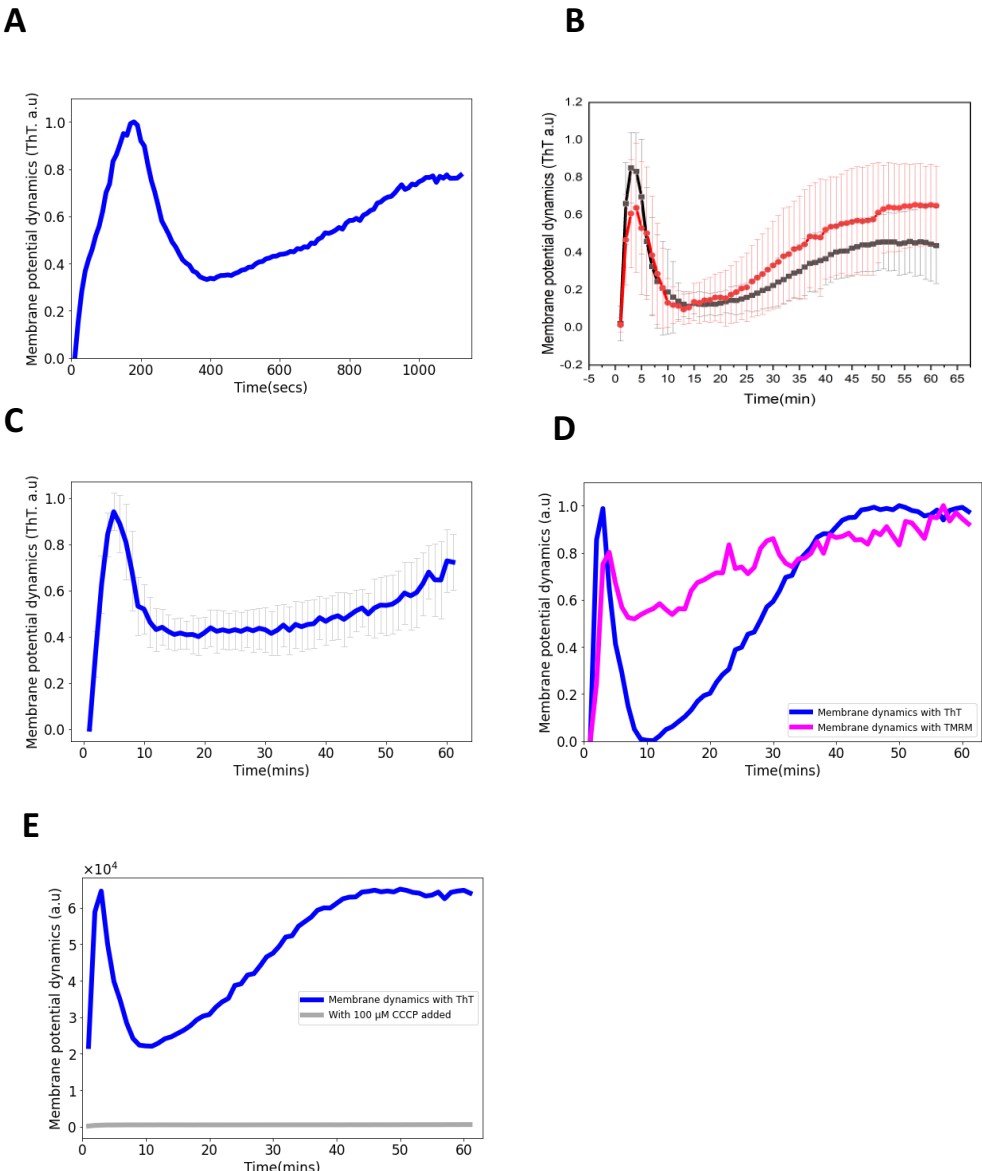

**Appendix 1—figure 1.** Control experiments for ThT fluorescence in *E. coli*. (**A**) ThT fluorescence as a function of time shows the *E. coli* membrane potential dynamics under light stress with images collected every 10 s. (**B**) ThT fluorescence as a function of time shows the membrane potential dynamics for both wildtype *E. coli* strains, DH5α (red) and *E. coli* BW25113 (black) (Data: mean ± SD). (**C**) ThT fluorescence as a function of time shows the membrane potential dynamics of *E. coli* in M9 media. The light stress was applied continuously each minute for 1 hr with images collected every minute. (**D**) TMRM and ThT fluorescence as a function of time with *E. coli*. The cationic dye, TMRM is used to confirm the membrane potential dynamics with ThT. (**E**) ThT fluorescence as a function of time. CCCP quenches membrane potential dynamics. CCCP was added to cell suspension and left for 50 min before exposure to blue light stress.

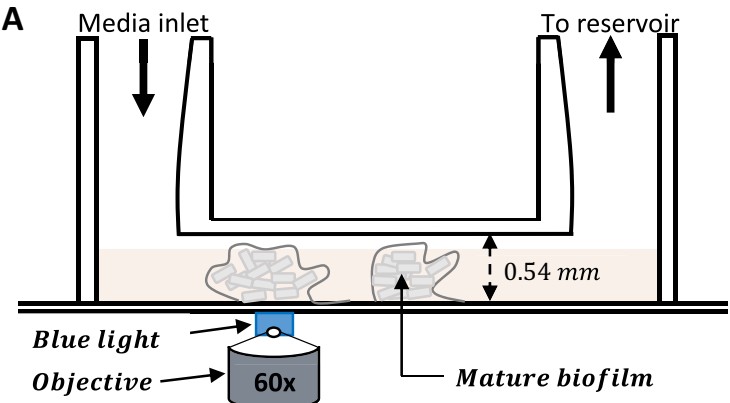

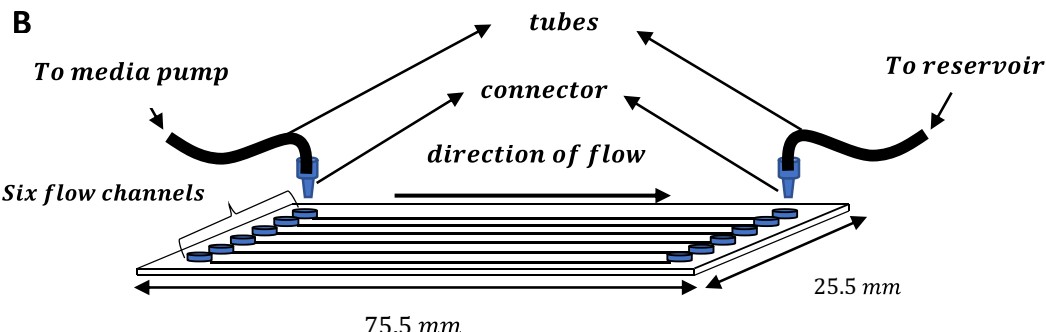

**Appendix 1—figure 2.** Schematic of the IBIDI flow cells used for all the experiments. (**A**) IBIDI uncoated glass bottom μ-Slide VI$^{0.5}$ showing the inlet and outlet openings. The 0.54 mm depth enabled growth of biofilms in 3D. (**B**) IBIDI uncoated glass bottom μ-Slide VI$^{0.5}$ with six identical wells for the growth of *E. coli* biofilms.

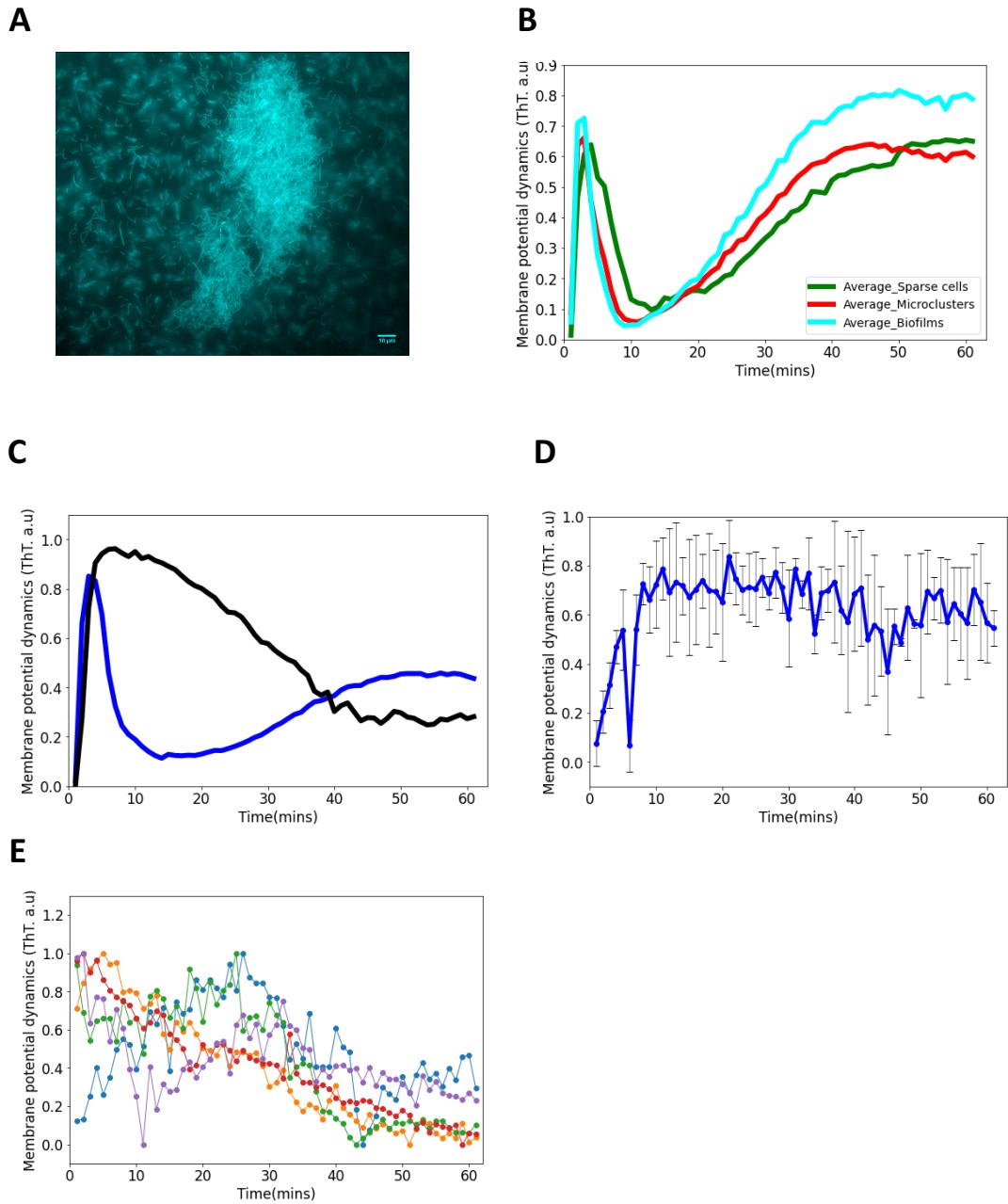

**Appendix 1—figure 3.** ThT fluorescence in *E. coli* biofilms. (**A**) Representative fluorescence image of a mature sessile biofilm labeled with ThT. The scale bar is 10 μm. (**B**) ThT fluorescence as a function of time for sparse cells, clustered cells and biofilms. First peak latency is much shorter in *E. coli* biofilms than in single cells and microclusters. (**C**) ThT fluorescence shown as a function of time of irradiation. Deletion of Kch inactivates the second peak in *single cell E. coli* BW25113 Δ*kch*-mutants. Data is a mean from three experimental replicates per time point for *E. coli* BW25113 Δ*kch*-mutants (black) and BW25113 (blue). (**D**) Membrane potential dynamics of kch-complemented DH5α. (**E**) Calcium flux in single cells DH5α under light stress.

**A**

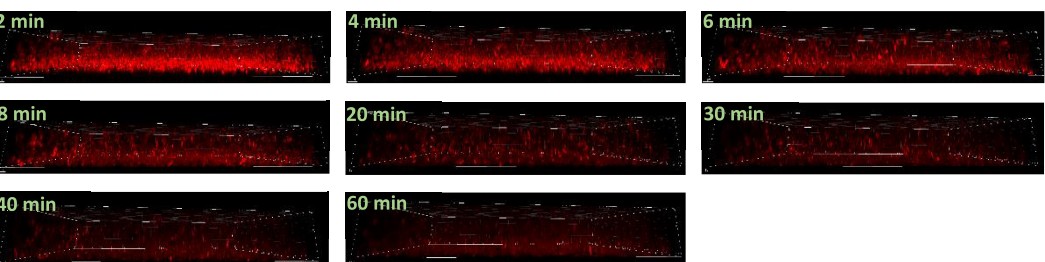

**B**

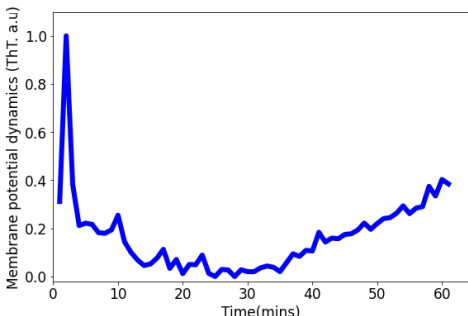

**Appendix 1—figure 4.** Confocal microscopy experiments with ThT in *E. coli* biofilms. (**A**) Timelapse of the sagittal section of the three-dimensional *E. coli* biofilm obtained using the confocal microscope. (**B**) Representative global ThT intensity membrane potential dynamics trace as a function of time obtained from a 3D DH5α *E. coli* biofilm. The scale bars for all the image are 8 µm.

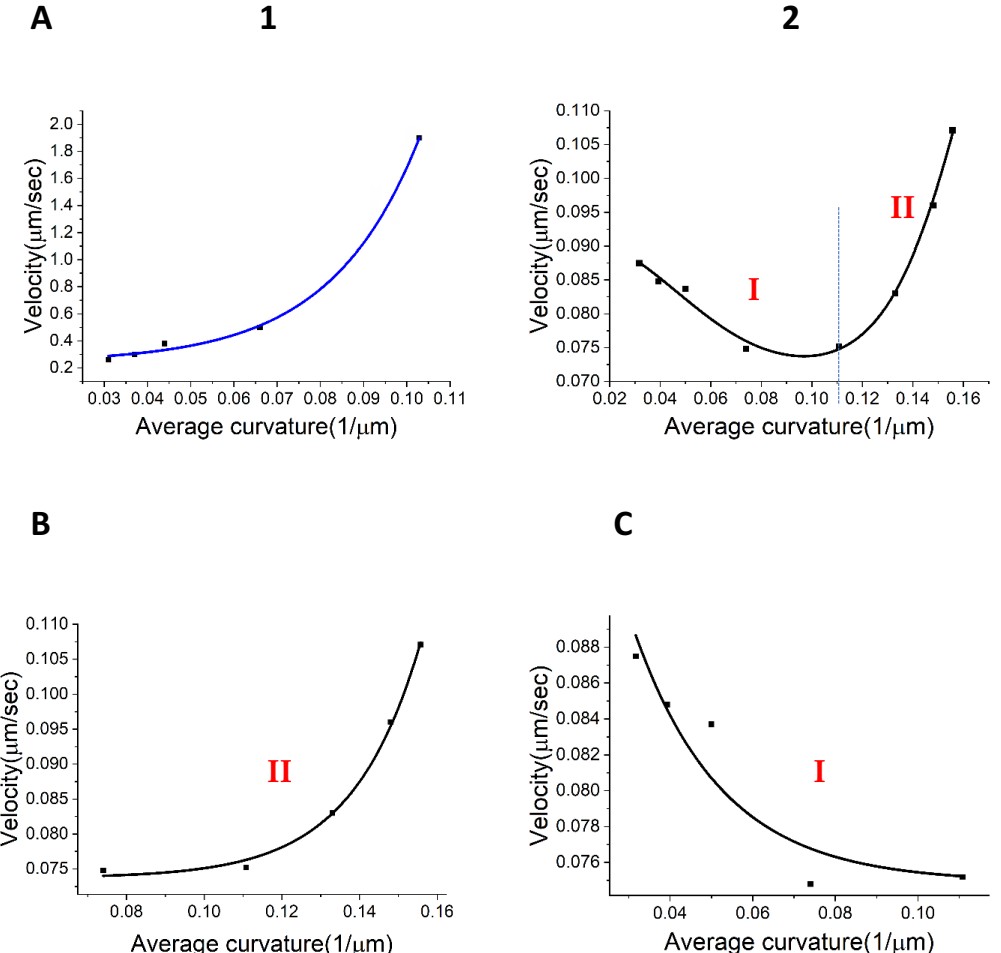

**Appendix 1—figure 5.** Relationships between velocity and curvature for the wavefronts. (**A**) (1) Relation between the wavefront velocity and average curvature from the experimental data for the centrifugal wavefront. (**A**) (2) Nonlinear relationship between the wavefront velocity and average curvature for the centripetal wavefront. (**B**) Exponential fit for stage I of the centripetal wavefront travel as shown in (A 2). (**C**) Exponential fit for stage II of the centripetal wavefront travel shown in (A 2).

## Mathematical model of membrane dynamics in single cell *E. coli*

We formulated a conductance model to explain the membrane potential dynamics in single *E. coli* cells. Our biophysical model is of the type originally used by Hodgkin and Huxley to explain excitability in squid axons (*Hodgkin and Huxeley, 1952*). We extended a version of this model which was previously used to study metabolic stress in *B. subtilis* biofilm *Prindle et al., 2015*, *Blee et al., 2020*,

$$C\frac{dV}{dt} = -g_Q z^4 \left(V - V_Q\right) - g_{Kch} n^4 \left(V - V_{Kch}\right) - g_L \left(V - V_L\right),$$  (A1)

where $g_Q$, $g_{Kch}$, and $g_L$ represent the conductance of the Q$^+$, K$^+$ and the leak channel, respectively. $V_q$, $V_{Kch}$, and $V_L$ represents the Nernst potentials for the Q$^+$, K$^+$ and the leak ions, respectively. $C$ is the membrane capacitance.

We assumed that both of the positively charged ion channels (Q and Kch) have four subunits which are their four activation gates. The variable z represents the gating variable for the $Q^+$ channels. The fraction of time the $Q^+$ channel is open can then be represented by

$$\frac{dz}{dt} = \alpha\left(S\right)\left(1 - z\right) - \beta_z.$$  (A2)

The $Q^+$ channel opening rate $\alpha(S)$ depends on the accumulation of the light induced stress. This stress is related to the levels of ROS. $\beta_z$ represents the rate at which open channels close.

The ion channel gating variable $n$ incorporates the fraction of time the potassium channel $Kch$ is open. It has the relation

$$\frac{dn}{dt} = \alpha(S)(1-n) - \beta_n.$$
(A3)

where $\alpha(S)$ stands for the opening rate for the K$^+$ and depends on the light-induced ROS stress. **Blee et al., 2020**, has already shown that blue-light raises the level of ROS and causes the ion-channels to open.

The stress dependency of $\alpha(S)$ can be further described as

$$\alpha(S) = \frac{\alpha_0 S^r}{S_{th}^r + S^r}$$
(A4)

where $S_{th}$ is the threshold stress value for ion channel opening. $r$ is the cooperativity factor and $\alpha_0$ is the maximal opening rate.

Motivated by the description of neuronal excitability **Hodgkin and Huxeley, 1952** and the works on bacterial signaling dynamics **Prindle et al., 2015**; **Blee et al., 2020**, we assumed that the ROS-induced stress induced by blue-light irradiation is related to the action potential in *E. coli* cells,

$$\frac{ds}{dt} = \frac{\propto_s (V_{th} - V)}{exp\left(\frac{V_{th} - V}{\sigma} - 1\right)} - \gamma_s S \quad .$$
(A5)

The term $\frac{\propto_s (V_{th}-V)}{exp\left(\frac{V_{th}-V}{\sigma}-1\right)}$ is a threshold non-linear function of the *E. coli* cell membrane hyperpolarization, $\alpha_s$ is the stress production constant and $\gamma_s$ is the stress decay rate. We assume that the functions $\alpha_s$ and $\gamma_s$ have a strong link to the gating of the ion channels.

Following **Prindle et al., 2015**, we assumed that the reversal potential increases linearly with the excess extracellular positively charged ion concentration and the relationship between the extracellular ion concentration and the reversal potential can be stated as

$$\frac{dE}{dt} = F_{g_Q} z^4 (V - V_Q) + F_{g_k} n^4 (V - V_K) - \gamma_e E$$
(A6)

where $E$ represent the extracellular ion concentration. The first and second term on the right-hand-side expresses the link between the extracellular q$^+$ and K$^+$ ions, and the membrane potential. $F_{g_Q}$ and $F_{g_k}$ incorporates capacitance of the membrane to each of the ion channels.

Following Prindle, we represent the inverse relationship between the ThT concentration and the efflux of the positively charged ions (hyperpolarization) by a linear function,

$$\frac{dH}{dt} = \alpha_t (V_0 - V) - \gamma_t H \quad .$$
(A7)

The ThT decay rate is represented by $\gamma_t$ and $\alpha_t$ is the ThT uptake rate coefficient. $V$ is the membrane potential at a instant of time while $V_0$ is the membrane potential in the absence of the ROS (the resting potential).

## Agent-based fire-diffuse-fire model

The fire-diffuse-fire (FDF) model was previously developed to study intracellular calcium dynamics in living cells (**Dawson et al., 1999**; **Keener and Sneyd, 2009**). In the model, once an individual cell at a particular site in the biofilm reaches a threshold concentration $c^*$, the cells fires, releasing a given concentration $\sigma$ of the $C^+$ ions. $C^+$ incorporates both unknown ions and potassium ions $(K^+)$ involved in our spike initiation and propagation in *E. coli* biofilms. A $C^+$ wavefront propagates within cells in the biofilm due to progressive firing of neighboring cells due to $C^+$ release close by.

The FDF model is based on a type of reaction-diffusion equation and can be written in three dimensions as

$$\frac{\partial C}{\partial t} = D_C \nabla^2 C + \sigma \sum_i \delta(\hat{r} - r_i)\,\delta(t - t_i),$$ (A8)

where $C$ is the chemical field concentration, $D_C$ is the diffusion coefficient of the ions, $\sigma$ is the fixed amount of chemical $C$, $r_i$ are the positions of the release sites, $t_i$ is the time the chemical attains threshold value $c^*$ at the $i^{th}$ release site and $\delta$ is the Dirac delta.

This original formulation of the FDF model does not include a decay term and results in the dynamics of the ion rising monotonically (**Coombes, 2001**). To correctly reflect findings in our experiment, we include a decay term, $-\vartheta C$, which allows for refractoriness of the system.

$$\frac{\partial C}{\partial t} = D_C \nabla^2 C - \vartheta C + \sigma \sum_i \delta(\hat{r} - r_i)\,\delta(t - t_i) \quad .$$ (A9)

The first term $D_C \nabla^2 C$ on the right-hand-term represent the diffusion term, while the remaining term on the right-hand-side, $-\vartheta C + \sigma \sum_i \partial(\hat{r} - r_i)\,\partial(t - t_i)$ are the reaction terms. $\vartheta$ is the ion decay rate.

The solution of *Equation A9* in one dimension for bacteria placed on a perfect lattice $b$ is

$$C_i(x,t) = \sigma \frac{H(t - t_i)}{\sqrt{4\pi D_C(t - t_i)}} exp\left(-\frac{(x - ib)^2}{4D(t - t_i)} - \vartheta(t - t_i)\right)$$ (A10)

where $H(t - t_i)$ is the Heaviside function so the cells at the $i^{th}$ position only fires at the time $t_i$. $C_i(x,t)$ is the ion concentration for the individual cell (cell $i$).

For all cells along the x-axis, we have

$$C(x,t) = \sum_i \sigma \frac{H(t - t_i)}{\sqrt{4\pi D_C(t - t_i)}} exp\left(-\frac{(x - ib)^2}{4D(t - t_i)} - \vartheta(t - t_i)\right),$$ (A11)

where

$$C(x,t) = \sum_i C_i(x,t).$$ (A12)

Suppose we assume that cells at definite location, say $i = N, N - 1$, have spiked at definite times $t_N > t_N - 1 > \ldots$, it is expected that they could only spike again at a time $t_{N+1}$. This time $t_{N+1}$ is determined by when the field concentration $C$, at time $t_{N+1}$ reaches the threshold value $q^*$. Therefore, we expect a steady wave to propagate only when the neighboring cells exhibit a constant time difference, $\tau = t_i - t_{i-1}$, between their firing. This time constant becomes a solution of the equation

$$\frac{c^* b}{\sigma} = \sum_{n=1}^{\infty} \frac{1}{\sqrt{4\pi n \rho}} exp\left(\frac{-n}{4\rho} - \eta^2 n\right),$$ (A13)

where $\rho = \frac{D_C \tau}{b^2}$ and $\eta^2 = \frac{\vartheta b^2}{D_C}$ are both dimensionless. A wave propagates when $\frac{c^* b}{\sigma} < 1$. The velocity $\nu$ of the propagating wave can then be obtained from $\frac{b}{\tau}$.

While exact solution of this 1D form of the FDF model with regularly placed bacteria can be obtained as shown, we used agent-based modeling to solve this model in three-dimensions with heterogeneously placed bacteria i.e. solve *Equation A9*. We had used the FDF model to successfully model electrical signaling in 2D growing biofilm (**Blee et al., 2019**). We adopted and applied the conditions and rules of this FDF model in our agent-based modeling of three-dimensional biofilm. A key extension was incorporating it in the agent-based modeling tool, BSim, to simulate electrical signaling in three-dimensional biofilms.

BSim is an open-source customizable Java based agent-based modeling software. It is specifically built to incorporate the effects of complex spatial environmental features and community heterogeneity (**Gorochowski et al., 2012**). We modeled the microfluidic device as a three-dimensional fluid in a rectangular environment. We implemented the biologically relevant parameters and field constants

(see *Appendix 1—table 1*) to obtain similar ion-channel mediated waves to those observed in three-dimensional biofilms. The positively charged ions $Q^+$, were described via their diffusion coefficients.

A BSim environment is built as an octree which also controls the spread of the chemical in the $x, y, z$ space. A Bsim environment, 64 × 64 × 64 microns in size with fixed boundaries was created (*Figure 7A*). A spherical biofilm of radius 32 μm containing ~6000 cells was implemented. The signaling ions chemical has a diffusivity and decay rate of 0.02 μm²/s and 7 × 10⁻³ molecules⁻¹, respectively.

The ion diffusion inside the BSim space was calculated according to Fick's law of diffusion,

$$\frac{\partial C}{\partial t} = D\nabla^2 C,$$

(A14)

where $C$ is the ion concentration in the field and $D$ is the diffusion coefficient of the ions. Java constructors were used to initialize the fields of ion concentration and control their diffusivity and decay rate.

**Appendix 1—table 1.** Parameters for the Mathematical modeling of membrane potential dynamics in *E. coli*.

| Parameter | Description | Value | Units |
|---|---|---|---|
| $g_K$ | Potassium channel conductance | 90 | min⁻¹ |
| $g_L$ | Leak channel conductance | 0.2 | min⁻¹ |
| $g_q$ | $q$ channel conductance | 5 | min⁻¹ |
| $V_K$ | Nernst Potential for Potassium | −94 | mV |
| $V_q$ | Nernst potential for $q$ channel | −200 | mV |
| $V_L$ | Nernst potential for leak channel | −156 | mV |
| $S_{th}$ | Stress threshold for opening of ion channels | 0.04 | μM |
| $V_{th}$ | Voltage threshold for stress production | −150 | mV |
|  | Maximum rate of channel opening | 2 | min⁻¹ |
| $b0$ | Channel opening rate decay constant | 1.3 | min⁻¹ |
| $r$ | Cooperativity parameter for ion channels | 1 | – |
| $\sigma$ | Stress threshold sharpness coefficient | 0.2 | mV |
| $y_e$ | Extracellular ion relaxation rate | 10 | min⁻¹ |
| $dl$ | Leak slope coefficient | 8 | mV/mM |
| $d_q$ | $q$ channel slope coefficient | 2 | mV/mM |
| $d_K$ | Potassium channel slope coefficient | 1 | mV/mM |
| $\alpha_s$ | Stress production slope coefficient | 0.001 | μM/ (minmV) |
| $\alpha_t$ | ThT uptake rate coefficient | 0.4 | μM/ (minmV) |
| $\gamma_s$ | Stress decay rate | 0.1 | min⁻¹ |
| $\gamma_t$ | ThT decay rate | 4 | min⁻¹ |
| $F$ | Membrane capacitance | 5.6 | mM/mV |

**Appendix 1—table 2.** Parameters used for the agent-based Fire-diffuse-fire model in the three-dimensional *E. coli* biofilm.

| Field and model parameters | Symbols | Values |
|---|---|---|
| Diffusion coefficient | Dc | 0.1 μm² |
| Simulation time step | Ts | 1 sec |

*Appendix 1—table 2 Continued on next page*

*Appendix 1—table 2 Continued*

| Field and model parameters | Symbols | Values |
|---|---|---|
| Total time of simulation for the 2 peaks | ts | 3660 sec |
| Total time of simulation for the first peak | ts | 720 sec |
| Size of fluid-filled environment | Sb | 64 µm × 64 µm × 64 µm |
| Fixed amount of chemical added | σ | $5 \times 10^9$ µm$^{-3}$ |
| Threshold concentration to fire | c | $10^3$ µm$^{-3}$ |
| Decay rate (for refractoriness) | υ | $7 \times 10^{-3}$ molecules$^{-1}$ |

**Appendix 1—table 3.** Fit constants for the centrifugal and centripetal wavefronts from experimental data.

| Fit constants for the centrifugal wavefront | Values |
|---|---|
| $V_0$ | 0.23 ± 0.04 |
| $A$ | 0.010 ± 0.008 |
| $z$ | 0.020 ± 0.002 |
| **Fit constants for the centripetal wavefront – Stage I** | |
| $V_0$ | 0.074 ± 0.001 |
| $A$ | – |
| $z$ | 0.018 ±0.002 |
| **Fit constants for the centripetal wavefront – Stage II** | |
| $V_0$ | 0.075 ± 0.003 |
| $A$ | – |
| $z$ | −0.02 ± 0.01 |

**Appendix 1—table 4.** Media recipe, bacterial strains, software, and microfluidic components.

| Media | Recipe | |
|---|---|---|
| Luria Broth (LB) | 10 g/l NaCl, 5 g/l yeast extract, 10 g/l Tryptone. Distilled water to 1 L | |
| LB agar | 10 g/l NaCl, 5 g/l yeast extract, 10 g/l Tryptone, 15 g/l agar. | |
| Minimal Media (M9) (1 L) | M9 salts (64 g/l Na$_2$HPO$_4$·7 $_H$20, 15 g/l KH$_2$PO$_4$, 2.5 g/l NaCl, 5.0 g NH$_4$Cl), 1 M CaCl$_2$, 1 M MgSO$_4$, 10% Arginine, 20% of glucose | |
| **Experimental Models: Organisms/Strains** | | |
| *E. coli* DH5α | Ian Robert's lab | |
| *E. coli* DH5α (Δ*Kch*) | This study | |
| *E. coli* DH5α (ΔMscK) | This study | |
| *E. coli* DH5α (ΔMscL) | This study | |
| *E. coli* DH5α (ΔMscS) | This study | |
| *E. coli* BW25113 (JW1242-1) | Keio Collection *Baba et al., 2006* | |
| *E. coli* BW25113 (Δ*Kch*) | Keio Collection *Baba et al., 2006* | |
| *E. coli* BW25113 (ΔMscL) | Keio Collection *Baba et al., 2006* | |
| *E. coli* BW25113 (ΔMscS) | Keio Collection *Baba et al., 2006* | |
| *E. coli* BW25113 (ΔMscK) | Keio Collection *Baba et al., 2006* | |
| **Reagent or Resource** | **Source** | **IdentifierDENTIFIER** |

*Appendix 1—table 4 Continued on next page*

*Appendix 1—table 4 Continued*

| Media | Recipe | |
|---|---|---|
| Propidium Iodide (PI) | Thermo Fisher Scientific | Cat #: P1304MP |
| Thioflavin T (ThT) | Sigma-Aldrich | Cat #: T3516 |
| Carbonyl cyanide 3-chlorophenylhydrazone (CCCP) | Sigma-Aldrich | Cat #: C2759 |
| ION Potassium Green–4 (IPG-4) AM | ION Biosciences | https://ionbiosciences.com/product/ipg-4-am/ |
| Tetramethylrhodamine methyl ester perchlorate (TMRM) | Sigma-Aldrich | Cat No: T5428 |
| pkLL11-GCaMP6f bb100 | A gift from Joel Kralj | Addgene plasmid #158983 |
| **Software and Algorithms** *Hartmann et al., 2021*; *Schindelin et al., 2012* | | |
| ImageJ Fiji | *Schindelin et al., 2012* | https://fiji.sc/ |
| Python (Anaconda) | Python | https://anaconda.org/ContinuumIO |
| MATLAB | MATLAB | https://uk.mathworks.com/products/matlab.html |
| BiofilmQ | *Hartmann et al., 2021* | https://drescherlab.org/data/biofilmQ/ docs/usage/installation.html |
| OriginPro | https://www.originlab.com/origin | https://www.originlab.com/origin |
| IMARIS | https://imaris.oxinst.com/ | https://imaris.oxinst.com/ |
| **Microfluidics** | | |
| IBIDI flow cell | Thistle Scientific | Cat #: 80607 |
| IBIDI Elbow Luer Connector Male | Thistle Scientific | Cat #: 10802 |
| C-Flex laboratory tubing with I.D. x O.D. 1/32 in. x 3/32 in | Sigma-Aldrich | Cat #: T8413-25FT. |

