## [Editor Report · eLife Assessment]

This potentially **valuable** study presents claims of evidence for coordinated membrane potential oscillations in *E. coli* biofilms that can be linked to a putative K+ channel and that may serve to enhance photo-protection. The finding of waves of membrane potential would be of interest to a wide audience from molecular biology to microbiology and physical biology. Unfortunately, a major issue is that it is unclear whether the dye used can act as a Nernstian membrane potential dye in *E. coli*. The arguments of the authors, who largely ignore previously published contradictory evidence, are not adequate in that they do not engage with the fact that the dye behaves in their hands differently than in the hands of others. In addition, the lack of proper validation of the experimental method including key control experiments leaves the evidence **incomplete**.

---

## [Referee Report · Reviewer #1 (Public Review)]

(1) Significance of the findings:

Cell-to-cell communication is essential for higher functions in bacterial biofilms. Electrical signals have proven effective in transmitting signals across biofilms. These signals are then used to coordinate cellular metabolisms or to increase antibiotic tolerance. Here, the authors have reported for the first time coordinated oscillation of membrane potential in *E. coli* biofilms that may have a functional role in photoprotection.

(2) Strengths of the manuscript:

- The authors report original data.

- For the first time, they showed that coordinated oscillations in membrane potential occur in *E. coli* biofilms.

- The authors revealed a complex two-phase dynamic involving distinct molecular response mechanisms.

- The authors developed two rigorous models inspired by (1) Hodgkin-Huxley model for the temporal dynamics of membrane potential and (2) Fire-Diffuse-Fire model for the propagation of the electric signal.

- Since its discovery by comparative genomics, the Kch ion channel has not been associated with any specific phenotype in *E. coli*. Here, the authors proposed a functional role for the putative gated-voltage-gated K+ ion channel (Kch channel) : enhancing survival under photo-toxic conditions.

(3) Weakness:

- Contrarily to what is stated in the abstract, the group of B. Maier has already reported collective electrical oscillations in the Gram-negative bacterium Neisseria gonorrhoeae (Hennes et al., PLoS Biol, 2023).

- The data presented in the manuscript are not sufficient to conclude on the photo-protective role of the Kch channel. The authors should perform the appropriate control experiments related to Fig4D,E, i.e. reproduce these experiments without ThT to rule out possible photo-conversion effects on ThT that would modify its toxicity. In addition, it looks like the data reported on Fig 4E are extracted from Fig 4D. If this is indeed the case, it would be more conclusive to report the percentage of PI-positive cells in the population for each condition. This percentage should be calculated independently for each replicate. The authors should then report the average value and standard deviation of the percentage of dead cells for each condition.

- Although Fig 4A clearly shows that light stimulation has an influence on the dynamics of ThT signal in the biofilm, it is important to rule out possible contributions of other environmental variations that occur when the flow is stopped at the onset of light stimulation. I understand that for technical reasons, the flow of fresh medium must be stopped for the sake of imaging. Therefore, I suggest to perform control experiments consisting in stopping the flow at different time intervals before image acquisition (30min or 1h before). If there is no significant contribution from environmental variations due to medium perfusion arrest, the dynamics of ThT signal must be unchanged regardless of the delay between flow stop and the start of light stimulation.

- To precise the role of K+ in the habituation response, I suggest using the ionophore valinomycin at sub-inhibitory concentrations (5 or 10µM). It should abolish the habituation response. In addition, the Kch complementation experiment exhibits a sharp drop after the first peak but on a single point. It would be more convincing to increase the temporal resolution (1min->10s) to show that there are indeed a first and a second peak. Finally, the high concentration (100µM) of CCCP used in this study completely inhibits cell activity. Therefore, it is not surprising that no ThT dynamics was observed upon light stimulation at such concentration of CCCP.

- Since TMRM signal exhibits a linear increase after the first response peak (Supp Fig1D), I recommend to mitigate the statement at line 78.

- Electrical signal propagation is an important aspect of the manuscript. However, a detailed quantitative analysis of the spatial dynamics within the biofilm is lacking. At minima, I recommend to plot the spatio-temporal diagram of ThT intensity profile averaged along the azimuthal direction in the biofilm. In addition, it is unclear if the electrical signal propagates within the biofilm during the second peak regime, which is mediated by the Kch channel: I have plotted the spatio-temporal diagram for Video S3 and no electrical propagation is evident at the second peak. In addition, the authors should provide technical details of how R^2(t) is measured in the first regime (Fig 7E).

- In the series of images presented in supplementary Figure 4A, no wavefront is apparent. Although the microscopy technics used in this figure differs from other images (like in Fig2), the wavefront should be still present. In addition, there is no second peak in confocal images as well (Supp Fig4B) .

- Many important technical details are missing (e.g. biofilm size, R^2, curvature and 445nm irradiance measurements). The description of how these quantitates are measured should be detailed in the Material & Methods section.

- Fig 5C: The curve in Fig 5D seems to correspond to the biofilm case. Since the model is made for single cells, the curve obtained by the model should be compared with the average curve presented in Fig 1B (i.e. single cell experiments).

- For clarity, I suggest to indicate on the panels if the experiments concern single cell or biofilm experiments. Finally, please provide bright-field images associated to ThT images to locate bacteria.

- In Fig 7B, the plateau is higher in the simulations than in the biofilm experiments. The authors should add a comment in the paper to explain this discrepancy.

---

## [Referee Report · Reviewer #2 (Public Review)]

The authors use ThT dye as a Nernstian potential dye in *E. coli*. Quantitative measurements of membrane potential using any cationic indicator dye are based on the equilibration of the dye across the membrane according to Boltzmann's law.

Ideally, the dye should have high membrane permeability to ensure rapid equilibration. Others have demonstrated that *E. coli* cells in the presence of ThT do not load unless there is blue light present, that the loading profile does not look like it is expected for a cationic Nernstian dye. They also show that the loading profile of the dye is different for *E.coli* cells deleted for the TolC pump. I, therefore, objected to interpreting the signal from the ThT as a Vm signal when used in *E.coli*. Nothing the authors have said has suggested that I should be changing this assessment.

Specifically, the authors responded to my concerns as follows:

(1) 'We are aware of this study, but believe it to be scientifically flawed. We do not cite the article because we do not think it is a particularly useful contribution to the literature.' This seems to go against ethical practices when it comes to scientific literature citations. If the authors identified work that handles the same topic they do, which they believe is scientifically flawed, the discussion to reflect that should be included.

(2)'The Pilizota group invokes some elaborate artefacts to explain the lack of agreement with a simple Nernstian battery model. The model is incorrect not the fluorophore.'

It seems the authors object to the basic principle behind the usage of Nernstian dyes. If the authors wish to use ThT according to some other model, and not as a Nernstian indicator, they need to explain and develop that model. Instead, they state 'ThT is a Nernstian voltage indicator' in their manuscript and expect the dye to behave like a passive voltage indicator throughout it.

(3)'We think the proton effect is a million times weaker than that due to potassium i.e. 0.2 M K+

versus 10-7 M H+. We can comfortably neglect the influx of H+ in our experiments.'

I agree with this statement by the authors. At near-neutral extracellular pH, *E. coli* keeps near-neutral intracellular pH, and the contribution from the chemical concentration gradient to the electrochemical potential of protons is negligible. The main contribution is from the membrane potential. However, this has nothing to do with the criticism to which this is the response of the authors. The criticism is that ThT has been observed not to permeate the cell without blue light. The blue light has been observed to influence the electrochemical potential of protons (and given that at near-neutral intracellular and extracellular pH this is mostly the membrane potential, as authors note themselves, we are talking about Vm effectively). Thus, two things are happening when one is loading the ThT, not just expected equilibration but also lowering of membrane potential. The electrochemical potential of protons is coupled via the membrane potential to all the other electrochemical potentials of ions, including the mentioned K+.

(4) 'The vast majority of cells continue to be viable. We do not think membrane damage is dominating.' In response to the question on how the authors demonstrated TMRM loading and in which conditions (and while reminding them that TMRM loading profile in *E. coli* has been demonstrated in Potassium Phosphate buffer). The request was to demonstrate TMRM loading profile in their condition as well as to show that it does not depend on light. Cells could still be viable, as membrane permeabilisation with light is gradual, but the loading of ThT dye is no longer based on simple electrochemical potential (of the dye) equilibration.

(5) On the comment on the action of CCCP with references included, authors include a comment that consists of phrases like 'our understanding of the literature' with no citations of such literature. Difficult to comment further without references.

(6) 'Shielding would provide the reverse effect, since hyperpolarization begins in the dense centres of the biofilms. For the initial 2 hours the cells receive negligible blue light. Neither of the referee's comments thus seem tenable.'

The authors have misunderstood my comment. I am not advocating shielding (I agree that this is not it) but stating that this is not the only other explanation for what they see (apart from electrical signaling). The other I proposed is that the membrane has changed in composition and/or the effective light power the cells can tolerate. The authors comment only on the light power (not convincingly though, giving the number for that power would be more appropriate), not on the possible changes in the membrane permeability.

(7) 'The work that TolC provides a possible passive pathway for ThT to leave cells seems slightly niche. It just demonstrates another mechanism for the cells to equilibrate the concentrations of ThT in a Nernstian manner i.e. driven by the membrane voltage.' I am not sure what the authors mean by another mechanism. The mechanism of action of a Nernstian dye is passive equilibration according to the electrochemical potential (i.e. until the electrochemical potential of the dye is 0).

(8) 'In the 70 years since Hodgkin and Huxley first presented their model, a huge number of similar models have been proposed to describe cellular electrophysiology. We are not being hyperbolic when we state that the HH models for excitable cells are like the Schrödinger

equation for molecules. We carefully adapted our HH model to reflect the currently understood electrophysiology of *E. coli*.'

I gave a very concrete comment on the fact that in the HH model conductivity and leakage are as they are because this was explicitly measured. The authors state that they have carefully adopted their model based on what is currently understood for *E. coli* electrophysiology. It is not clear how. HH uses gKn^4 based on Figure2 here https://www.ncbi.nlm.nih.gov/pmc/articles/PMC1392413/pdf/jphysiol01442-0106.pdf, i.e. measured rise and fall of potassium conductance on msec time scales. I looked at the citation the authors have given and found a resistance of an entire biofilm of a given strain at 3 applied voltages. So why n^4 based on that? Why does unknown current have gqz^4 form? Sodium conductance in HH is described by m^3hgNa (again based on detailed conductance measurements), so why unknown current in E.coli by gQz^4? Why leakage is in the form that it is, based on what measurement?

Throughout their responses, the authors seem to think that collapsing the electrochemical gradient of protons is all about protons, and this is not the case. At near neutral inside and outside pH, the electrochemical potential of protons is simply membrane voltage. And membrane voltage acts on all ions in the cell.

Authors have started their response to concrete comments on the usage of ThT dye with comments on papers from my group that are not all directly relevant to this publication. I understand that their intention is to discredit a reviewer but given that my role here is to review this manuscript, I will only address their comments to the publications/part of publications that are relevant to this manuscript and mention what is not relevant.

Publications in the order these were commented on.

(1) In a comment on the paper that describes the usage of ThT dye as a Nernstian dye authors seem to talk about a model of an entire active cell.

'Huge oscillations occur in the membrane potentials of *E. coli* that cannot be described by the SNB model.' The two have nothing to do with each other. Nernstian dye equilibrates according to its electrochemical potential. Once that happens it can measure the potential (under the assumption that not too much dye has entered and thus lowered too much the membrane potential under measurement). The time scale of that is important, and the dye can only measure processes that are slower than that equilibration. If one wants to use a dye that acts under a different model, first that needs to be developed, and then coupled to any other active cell model.

(2) The part of this paper that is relevant is simply the usage of TMRM dye. It is used as Nernstian dye, so all the above said applies. The rest is a study of flagellar motor.

(3) The authors seem to not understand that the electrochemical potential of protons is coupled to the electrochemical potentials of all other ions, via the membrane potential. In the manuscript authors talk about, PMF~Vm, as DeltapH~0. Other than that this publication is not relevant to their current manuscript.

(4) The manuscript in fact states precisely that PMF cannot be generated by protons only and some other ions need to be moved out for the purpose. In near neutral environment it stated that these need to be cations (K+ e.g.). The model used in this manuscript is a pump-leak model. Neither is relevant for the usage of ThT dye.

Further comments include, along the lines of:

'The editors stress the main issue raised was a single referee questioning the use of ThT as an indicator of membrane potential. We are well aware of the articles by the Pilizota group and we believe them to be scientifically flawed. The authors assume there are no voltage-gated ion channels in *E. coli* and then attempt to explain motility data based on a simple Nernstian battery model (they assume *E. coli* are unexcitablematter). This in turn leads them to conclude the membrane dye ThT is faulty, when in fact it is a problem with their simple battery model.'

The only assumption made when using a cationic Nernstian dye is that it equilibrates passively across the membrane according to its electrochemical potential. As it does that, it does lower the membrane potential, which is why as little as possible is added so that this is negligible. The equilibration should be as fast as possible, but at the very least it should be known, as no change in membrane potential can be measured that is faster than that.

This behaviour should be orthogonal to what the cell is doing, it is a probe after all. If the cell is excitable, a Nernstian dye can be used, as long as it's still passively equilibrating and doing so faster than any changes in membrane potential due to excitations of the cells. There are absolutely no assumptions made on the active system that is about to be measured by this expected behaviour of a Nernstian dye. And there shouldn't be, it is a probe. If one wants to use a dye that is not purely Nernstian that behaviour needs to be described and a model proposed. As far as I can find, authors do no such thing.

There is a comment on the use of a flagellar motor as a readout of PMF, stating that the motor can be stopped by YcgR citing the work from 2023. Indeed, there is a range of references such as https://doi.org/10.1016/j.molcel.2010.03.001 that demonstrate this (from around 2000-2010 as far as I am aware). The timescale of such slowdown is hours (see here Figure 5 https://www.cell.com/cell/pdf/S0092-8674(10)00019-X.pdf). Needless to say, the flagellar motor when used as a probe, needs to stay that in the conditions used. Thus one should always be on the lookout at any other such proteins that could slow it down and we are not aware of yet or make the speed no longer proportional to the PMF. In the papers my group uses the motor the changes are fast, often reversible, and in the observation window of 30min. They are also the same with DeltaYcgR strain, which we have not included as it seemed given the time scales it's obvious, but certainly can in the future (as well as stay vigilant on any conditions that would render the motor a no longer suitable probe for PMF).

---

## [Referee Report · Reviewer #3 (Public Review)]

This manuscript by Akabuogu et al. investigates membrane potential dynamics in *E. coli*. Membrane potential fluctuations have been observed in bacteria by several research groups in recent years, including in the context of bacterial biofilms where they have been proposed to play a role in cellular communication. Here, these authors investigate membrane potential in *E. coli*, in both single cells and biofilms. I have reviewed the revised manuscript provided by the authors, as well as their responses to the initial reviews; my opinion about the manuscript is largely unchanged. I have focused my public review on those issues that I believe to be most pressing, with additional comments included in the review to authors. Although these authors are working in an exciting research area, the evidence they provide for their claims is inadequate, and several key control experiments are still missing. In some cases, the authors allude to potentially relevant data in their responses to the initial reviews, but unfortunately these data are not shown. Furthermore, I cannot identify any traveling wavefronts in the data included in this manuscript. In addition to the challenges associated with the use of Thioflavin-T (ThT) raised by the second reviewer, these caveats make the work presented in this manuscript difficult to interpret.

First, some of the key experiments presented in the paper lack required controls:

(1) This paper asserts that the observed ThT fluorescence dynamics are induced by blue light. This is a fundamental claim in the paper, since the authors go on to argue that these dynamics are part of a blue light response. This claim must be supported by the appropriate negative control experiment measuring ThT fluorescence dynamics in the absence of blue light- if this idea is correct, these dynamics should not be observed in the absence of blue light exposure. If this experiment cannot be performed with ThT since blue light is used for its excitation, TMRM can be used instead.

In response to this, the authors wrote that ‘the fluorescent baseline is too weak to measure cleanly in this experiment.’ If they observe no ThT signal above noise in their time lapse data in the absence of blue light, this should be reported in the manuscript- this would be a satisfactory negative control. They then wrote that ‘It appears the collective response of all the bacteria hyperpolarization at the same time appears to dominate the signal.’ I am not sure what they mean by this- perhaps that ThT fluorescence changes strongly only in response to blue light? This is a fundamental control for this experiment that ought to be presented to the reader.

(2) The authors claim that a ∆kch mutant is more susceptible to blue light stress, as evidenced by PI staining. The premise that the cells are mounting a protective response to blue light via these channels rests on this claim. However, they do not perform the negative control experiment, conducting PI staining for WT the ∆kch mutant in the absence of blue light. In the absence of this control it is not possible to rule out effects of the ∆kch mutation on overall viability and/or PI uptake. The authors do include a growth curve for comparison, but planktonic growth is a very different context than surface-attached biofilm growth. Additionally, the ∆kch mutation may have impacts on PI permeability specifically that are not addressed by a growth curve. The negative control experiment is of key importance here.

Second, the ideas presented in this manuscript rely entirely on analysis of ThT fluorescence data, specifically a time course of cellular fluorescence following blue light treatment. However, alternate explanations for and potential confounders of the observed dynamics are not sufficiently addressed:

(1) Bacterial cells are autofluorescent, and this fluorescence can change significantly in response to stress (e.g. blue light exposure). To characterize and/or rule out autofluorescence contributions to the measurement, the authors should present time lapse fluorescence traces of unstained cells for comparison, acquired under the same imaging conditions in both wild type and ∆kch mutant cells. In their response to reviewers the authors suggested that they have conducted this experiment and found that the autofluorescence contribution is negligible, which is good, but these data should be included in the manuscript along with a description of how these controls were conducted.

(2) Similarly, in my initial review I raised a concern about the possible contributions of photobleaching to the observed fluorescence dynamics. This is particularly relevant for the interpretation of the experiment in which catalase appears to attenuate the decay of the ThT signal; this attenuation could alternatively be due to catalase decreasing ThT photobleaching. In their response, the authors indicated that photobleaching is negligible, which would be good, but they do not share any evidence to support this claim. Photobleaching can be assessed in this experiment by varying the light dosage (illumination power, frequency, and/or duration) and confirming that the observed fluorescence dynamics are unaffected.

Third, the paper claims in two instances that there are propagating waves of ThT fluorescence that move through biofilms, but I do not observe these waves in any case:

(1) The first wavefront claim relates to small cell clusters, in Fig. 2A and Video S2 and S3 (with Fig. 2A and Video S2 showing the same biofilm.) I simply do not see any evidence of propagation in either case- rather, all cells get brighter and dimmer in tandem. I downloaded and analyzed Video S3 in several ways (plotting intensity profiles for different regions at different distances from the cluster center, drawing a kymograph across the cluster, etc.) and in no case did I see any evidence of a propagating wavefront. (I attempted this same analysis on the biofilm shown in Fig. 2A and Video S2 with similar results, but the images shown in the figure panels and especially the video are still both so saturated that the quantification is difficult to interpret.) If there is evidence for wavefronts, it should be demonstrated explicitly by analysis of several clusters. For example, a figure of time-to-peak vs. position in the cluster demonstrating a propagating wave would satisfy this. Currently, I do not see any wavefronts in this data.

(2) The other wavefront claim relates to biofilms, and the relevant data is presented in Fig. S4 (and I believe also in what is now Video S8, but no supplemental video legends are provided, and this video is not cited in text.) As before, I cannot discern any wavefronts in the image and video provided; Reviewer 1 was also not able to detect wave propagation in this video by kymograph. Some mean squared displacements are shown in Fig. 7. As before, the methods for how these were obtained are not clearly documented either in this manuscript or in the BioRXiv preprint linked in the initial response to reviewers, and since wavefronts are not evident in the video it is hard to understand what is being measured here- radial distance from where? (The methods section mentions radial distance from the substrate, this should mean Z position above the imaging surface, and no wavefronts are evident in Z in the figure panels or movie.) Thus, clear demonstration of these wavefronts is still missing here as well.

Fourth, I have some specific questions about the study of blue light stress and the use of PI as a cell viability indicator:

(1) The logic of this paper includes the premise that blue light exposure is a stressor under the experimental conditions employed in the paper. Although it is of course generally true that blue light can be damaging to bacteria, this is dependent on light power and dosage. The control I recommended above, staining cells with PI in the presence and absence of blue light, will also allow the authors to confirm that this blue light treatment is indeed a stressor- the PI staining would be expected to increase in the presence of blue light if this is so.

(2) The presence of ThT may complicate the study of the blue light stress response, since ThT enhances the photodynamic effects of blue light in *E. coli* (Bondia et al. 2021 Chemical Communications). The authors could investigate ThT toxicity under these conditions by staining cells with PI after exposing them to blue light with or without ThT staining.

(3) In my initial review, I wrote the following: "In Figures 4D - E, the interpretation of this experiment can be confounded by the fact that PI uptake can sometimes be seen in bacterial cells with high membrane potential (Kirchhoff & Cypionka 2017 J Microbial Methods); the interpretation is that high membrane potential can lead to increased PI permeability. Because the membrane potential is largely higher throughout blue light treatment in the ∆kch mutant (Fig. 3[BC]), this complicates the interpretation of this experiment." In their response, the authors suggested that these results are not relevant in this case because "In our experiment methodology, cell death was not forced on the cells by introducing an extra burden or via anoxia." However, the logic of the paper is that the cells are in fact dying due to an imposed external stressor, which presumably also confers an increased burden as the cells try to deal with the stress. Instead, the authors should simply use a parallel method to confirm the results of PI staining. For example, the experiment could be repeated with other stains, or the viability of blue light-treated cells could be addressed more directly by outgrowth or colony-forming unit assays.

The CFU assay suggested above has the additional advantage that it can also be performed on planktonic cells in liquid culture that are exposed to blue light. If, as the paper suggests, a protective response to blue light is being coordinated at the biofilm level by these membrane potential fluctuations, the WT strain might be expected to lose its survival advantage vs. the ∆kch mutant in the absence of a biofilm.

Fifth, in several cases the data are presented in a way that are difficult to interpret, or the paper makes claims that are different to observe in the data:

(1) The authors suggest that the ThT and TMRM traces presented in Fig. S1D have similar shapes, but this is not obvious to me- the TMRM curve has very little decrease after the initial peak and only a modest, gradual rise thereafter. The authors suggest that this is due to increased TMRM photobleaching, but I would expect that photobleaching should exacerbate the signal decrease after the initial peak. Since this figure is used to support the use of ThT as a membrane potential indicator, and since this is the only alternative measurement of membrane potential presented in text, the authors should discuss this discrepancy in more detail.

(2) The comparison of single cells to microcolonies presented in figures 1B and D still needs revision:

First, both reviewer 1 and I commented in our initial reviews that the ThT traces, here and elsewhere, should not be normalized- this will help with the interpretation of some of the claims throughout the manuscript.

Second, the way these figures are shown with all traces overlaid at full opacity makes it very difficult to see what is being compared. Since the point of the comparison is the time to first peak (and the standard deviation thereof), histograms of the distributions of time to first peak in both cases should be plotted as a separate figure panel.

Third, statistical significance tests ought to be used to evaluate the statistical strength of the comparisons between these curves. The authors compare both means and standard deviations of the time to first peak, and there are appropriate statistical tests for both types of comparisons.

(3) The authors claim that the curve shown in Fig. S4B is similar to the simulation result shown in Fig. 7B. I remain unconvinced that this is so, particularly with respect to the kinetics of the second peak- at least it seems to me that the differences should be acknowledged and discussed. In any case, the best thing to do would be to move Fig. S4B to the main text alongside Fig. 7B so that the readers can make the comparison more easily.

(4) As I wrote in my first review, in the discussion of voltage-gated calcium channels, the authors refer to "spiking events", but these are not obvious in Figure S3E. Although the fluorescence intensity changes over time, these fluctuations cannot be distinguished from measurement noise. A no-light control could help clarify this.

(5) In the lower irradiance conditions in Fig. 4A, the ThT dynamics are slower overall, and it looks like the ThT intensity is beginning to rise at the end of the measurement. The authors write that no second peak is observed below an irradiance threshold of 15.99 µW/mm2. However, could a more prominent second peak be observed in these cases if the measurement time was extended? Additionally, the end of these curves looks similar to the curve in Fig. S4B, in which the authors write that the slow rise is evidence of the presence of a second peak, in contrast to their interpretation here.

Additional considerations:

(1) The analysis and interpretation of the first peak, and particularly of the time-to-fire data is challenging throughout the manuscript the time resolution of the data set is quite limited. It seems that a large proportion of cells have already fired after a single acquisition frame. It would be ideal to increase the time resolution on this measurement to improve precision. This could be done by imaging more quickly, but that would perhaps necessitate more blue light exposure; an alternative is to do this experiment under lower blue light irradiance where the first spike time is increased (Figure 4A).

(2) The authors suggest in the manuscript that "*E. coli* biofilms use electrical signalling to coordinate long-range responses to light stress." In addition to the technical caveats discussed above, I am missing a discussion about what these responses might be. What constitutes a long-range response to light stress, and are there known examples of such responses in bacteria?

(3) The presence of long-range blue light responses can also be interrogated experimentally, for example, by repeating the Live/Dead experiment in planktonic culture or the single-cell condition. If the protection from blue light specifically emerges due to coordinated activity of the biofilm, the ∆kch mutant would not be expected to show a change in Live/Dead staining in non-biofilm conditions. The CFU experiment I mentioned above could also implicate coordinated long-range responses specifically, if biofilms and liquid culture experiments can be compared (although I know that recovering cells from biofilms is challenging.)

4. At the end of the results section, the authors suggest a critical biofilm size of only 4 μm for wavefront propagation (not much larger than a single cell!) The authors show responses for various biofilm sizes in Fig. 2C, but these are all substantially larger (and this figure also does not contain wavefront information.) Are there data for cell clusters above and below this size that could support this claim more directly?

(5) In Fig. 4C, the overall trajectories of extracellular potassium are indeed similar, but the kinetics of the second peak of potassium are different than those observed by ThT (it rises minutes earlier)- is this consistent with the idea that Kch is responsible for that peak? Additionally, the potassium dynamics also include the first ThT peak- is this surprising given that the Kch channel has no effect on this peak according to the model?

Detailed comments:

Why are Fig. 2A and Video S2 called a microcluster, whereas Video S3, which is smaller, is called a biofilm?

"We observed a spontaneous rapid rise in spikes within cells in the center of the biofilm" (Line 140): What does "spontaneous" mean here?

"This demonstrates that the ion-channel mediated membrane potential dynamics is a light stress relief process.", "*E. coli* cells employ ion-channel mediated dynamics to manage ROS-induced stress linked to light irradiation." (Line 268 and the second sentence of the Fig. 4F legend): This claim is not well-supported. There are several possible interpretations of the catalase experiment (which should be discussed); this experiment perhaps suggests that ROS impacts membrane potential but does not indicate that these membrane potential fluctuations help the cells respond to blue light stress. The loss of viability in the ∆kch mutant might indicate a link between these membrane potential experiments and viability, but it is hard to interpret without the no light controls I mention above.

"The model also predicts... the external light stress" (Lines 338-341): Please clarify this section. Where does this prediction arise from in the modeling work? Second, I am not sure what is meant by "modulates the light stress" or "keeps the cell dynamics robust to the intensity of external light stress" (especially since the dynamics clearly vary with irradiance, as seen in Figure 4A).

"We hypothesized that *E. coli* not only modulates the light-induced stress but also handles the increase of the ROS by adjusting the profile of the membrane potential dynamics" (Line 347): I am not sure what "handles the ROS by adjusting the profile of the membrane potential dynamics" means. What is meant by "handling" ROS? Is the hypothesis that membrane potential dynamics themselves are protective against ROS, or that they induce a ROS-protective response downstream, or something else? Later the authors write that changes in the response to ROS in the model agree with the hypothesis, but just showing that ROS impacts the membrane potential does not seem to demonstrate that this has a protective effect against ROS.

"Mechanosensitive ion channels (MS) are vital for the first hyperpolarization event in *E. coli*." (Line 391): This is misleading- mechanosensitive ion channels totally ablate membrane potential dynamics, they don't have a specific effect on the first hyperpolarization event. The claim that mechanonsensitive ion channels are specifically involved in the first event also appears in the abstract.

Also, the apparent membrane potential is much lower even at the start of the experiment in these mutants (Fig. 6C-D)- is this expected? This seems to imply that these ion channels also have a blue light-independent effect.

Throughout the paper, there are claims that the initial ThT spike is involved in "registering the presence of the light stress" and similar. What is the evidence for this claim?

"We have presented much better quantitative agreement of our model with the propagating wavefronts in *E. coli* biofilms..." (Line 619): It is not evident to me that the agreement between model and prediction is "much better" in this work than in the cited work (reference 57, Hennes et al. 2023). The model in Figure 4 of ref. 57 seems to capture the key features of their data.

In methods, "Only cells that are hyperpolarized were counted in the experiment as live" (Line 745): what percentage of cells did not hyperpolarize in these experiments?

Some indication of standard deviation (error bars or shading) should be added to all figures where mean traces are plotted.

Video S8 is very confusing- why does the video play first forwards and then backwards? It is easy to misinterpret this as a rise in the intensity at the end of the experiment.

---

## [Author Response]

The issue of a control without blue light illumination was raised. Clearly without the light we will not obtain any signal in the fluorescence microscopy experiments, which would not be very informative. Instead, we changed the level of blue light illumination in the fluorescence microscopy experiments (figure 4A) and the response of the bacteria scales with dosage. It is very hard to find an alternative explanation, beyond that the blue light is stressing the bacteria and modulating their membrane potentials.

One of the referees refuses to see wavefronts in our microscopy data. We struggle to understand whether it is an issue with definitions (Waigh has published a tutorial on the subject in Chapter 5 of his book ‘The physics of bacteria: from cells to biofilms’, T.A.Waigh, CUP, 2024 – figure 5.1 shows a sketch) or something subtler on diffusion in excitable systems. We stand by our claim that we observe wavefronts, similar to those observed by Prindle et al^1^ and Blee et al^2^ for *B. subtilis* biofilms.

The referee is questioning our use of ThT to probe the membrane potential. We believe the Pilizota and Strahl groups are treating the *E. coli* as *unexcitable* cells, leading to their problems. Instead, we believe *E. coli* cells are *excitable* (containing the voltage-gated ion channel Kch) and we now clearly state this in the manuscript. Furthermore, we include a section here discussing some of the issues with ThT.

**Use of ThT as a voltage sensor in cells**

ThT is now used reasonably widely in the microbiology community as a voltage sensor in both bacterial [Prindle et al]1 and fungal cells [Pena et al]12. ThT is a small cationic fluorophore that loads into the cells in proportion to their membrane potential, thus allowing the membrane potential to be measured from fluorescence microscopy measurements.

Previously ThT was widely used to quantify the growth of amyloids in molecular biology experiments (standardized protocols exist and dedicated software has been created)13 and there is a long history of its use14. ThT fluorescence is bright, stable and slow to photobleach.

Author response image 1 shows a schematic diagram of the ThT loading in *E. coli* in our experiments in response to illumination with blue light. Similar results were previously presented by Mancini et al15, but regimes 2 and 3 were mistakenly labelled as artefacts.

**Author response image 1. sa4fig1:** Schematic diagram of ThT loading during an experiment with *E. coli* cells under blue light illumination i.e. ThT fluorescence as a function of time. Three empirical regimes for the fluorescence are shown (1, 2 and 3).

The classic study of Prindle et al on bacterial biofilm electrophysiology established the use of ThT in *B. subtilis* biofilms by showing similar results occurred with DiSc3 which is widely used as a Nernstian voltage sensor in cellular biology1 e.g. with mitochondrial membrane potentials in eukaryotic organisms where there is a large literature. We repeated such a comparative calibration of ThT with DiSc3 in a previous publication with both *B. subtilis* and *P. aeruginosa* cells2. ThT thus functioned well in our previous publications with Gram positive and Gram negative cells.

However, to our knowledge, there are now two groups questioning the use of ThT and DiSc3 as voltage sensors with *E. coli* cells15-16. The first by the Pilizota group claims ThT only works as a voltage sensor in regime 1 of **Author response figure 1** using a method based on the rate of rotation of flagellar motors. Another slightly contradictory study by the Strahl group claims DiSc316 only acts as a voltage sensor with the addition of an ionophore for potassium which allows free movement of potassium through the *E. coli* membranes.

Our resolution to this contradiction is that ThT does indeed work reasonably well with *E. coli*. The Pilizota group’s model for rotating flagellar motors assumes the membrane voltage is not varying due to excitability of the membrane voltage (otherwise a non-linear Hodgkin Huxley type model would be needed to quantify their results) i.e. *E. coli* cells are *unexcitable*. We show clearly in our study that ThT loading in *E. coli* is a function of irradiation with blue light and is a stress response of the *excitable* cells. This is in contradiction to the Pilizota group’s model. The Pilizota group’s model also requires the awkward fiction of why cells decide to unload and then reload ThT in regimes 2 and 3 of **Author response figure 1** due to variable membrane partitioning of the ThT. Our simple explanation is that it is just due to the membrane voltage changing and no membrane permeability switch needs to be invoked. The Strahl group’s16 results with DiSc3 are also explained by a neglect of the excitable nature of *E. coli* cells that are reacting to blue light irradiation. Adding ionophores to the *E. coli* membranes makes the cells unexcitable, reduces their response to blue light and thus leads to simple loading of DiSc3 (the physiological control of K+ in the cells by voltage-gated ion channels has been short circuited by the addition of the ionophore).

Further evidence of our model that ThT functions as a voltage sensor with *E. coli* include:

1. The 3 regimes in **Author response figure 1** from ThT correlate well with measurements of extracellular potassium ion concentration using TMRM i.e. all 3 regimes in **Author response figure 1** are visible with this separate dye (**figure 1d**).

2. We are able to switch regime 3 in **Author response figure 1**, off and then on again by using knock downs of the potassium ion channel Kch in the membranes of the *E. coli* and then reinserting the gene back into the knock downs. This cannot be explained by the Pilizota model.

We conclude that ThT works reasonably well as a sensor of membrane voltage in *E. coli* and the previous contradictory studies15-16 are because they neglect the excitable nature of the membrane voltage of *E. coli* cells in response to the light used to make the ThT fluoresce.

Three further criticisms of the Mancini et al method15 for calibrating membrane voltages include:

1. *E. coli* cells have clutches that are not included in their models. Otherwise the rotation of the flagella would be entirely enslaved to the membrane voltage allowing the bacteria no freedom to modulate their speed of motility.

2. Ripping off the flagella may perturb the integrity of the cell membrane and lead to different loading of the ThT in the *E. coli* cells.

3. Most seriously, the method ignores the activity of many other ion channels (beyond H+) on the membrane voltage that are known to exist with *E. coli* cells e.g. Kch for K+ ions. The Pilizota groups uses a simple Nernstian battery model developed for mitochondria in the 1960s. It is not adequate to explain our results.

An additional criticism of the Winkel et al study17 from the Strahl group is that it indiscriminately switches between discussion of mitochondria and bacteria e.g. on page 8 ‘As a consequence the membrane potential is dominated by H+’. Mitochondria are slightly alkaline intracellular organelles with external ion concentrations in the cytoplasm that are carefully controlled by the eukaryotic cells. *E. coli* are not i.e. they have neutral internal pHs, with widely varying extracellular ionic concentrations and have reinforced outer membranes to resist osmotic shocks (in contrast mitochondria can easily swell in response to moderate changes in osmotic pressure).

A quick calculation of the equilibrium membrane voltage of *E. coli* can be easily done using the Nernst equation dependent on the extracellular ion concentrations defined by the growth media (the intracellular ion concentrations in *E. coli* are 0.2 M K+ and 10-7 M H+ i.e. there is a factor of a million fewer H+ ions). Thus in contradiction to the claims of the groups of Pilizota15 and Strahl17, H+ is a minority determinant to the membrane voltage of *E. coli*. The main determinant is K+. For a textbook version of this point the authors can refer to Chapter 4 of D. White, et al’s ‘The physiology and biochemistry of prokaryotes’, OUP, 2012, 4th edition.

Even in mitochondria the assumption that H+ dominates the membrane potential and the cells are unexcitable can be questioned e.g. people have observed pulsatile depolarization phenomena with mitochondria18-19. A large number of K+ channels are now known to occur in mitochondrial membranes (not to mention Ca2+ channels; mitochondria have extensive stores of Ca2+) and they are implicated in mitochondrial membrane potentials. In this respect the seminal Nobel prize winning research of Peter Mitchell (1961) on mitochondria needs to be amended20. Furthermore, the mitochondrial work is clearly inapplicable to bacteria (the proton motive force, PMF, will instead subtly depend on non-linear Hodgkin-Huxley equations for the excitable membrane potential, similar to those presented in the current article). A much more sophisticated framework has been developed to describe electrophysiology by the mathematical biology community to describe the activity of electrically excitable cells (e.g. with neurons, sensory cells and cardiac cells), beyond Mitchell’s use of the simple stationary equilibrium thermodynamics to define the Proton Motive Force via the electrochemical potential of a proton (the use of the word ‘force’ is unfortunate, since it is a potential). The tools developed in the field of mathematical electrophysiology8 should be more extensively applied to bacteria, fungi, mitochondria and chloroplasts if real progress is to be made.

Related to the previous point, we now cite articles from the Pilizota and Strahl groups in the main text (one from each group). Unfortunately, the space constraints of *eLife* mean we cannot make a more detailed discussion in the main article.

In terms of modelling the ion channels, the Hodgkin-Huxley type model proposes that the Kch ion channel can be modelled as a typical voltage-gated potassium ion channel i.e. with a 𝑛^4^ term in its conductivity. The literature agrees that Kch is a voltage-gated potassium ion channel based on its primary sequence^3^. The protein has the typical 6 transmembrane helix motif for a voltage-gated ion channel. The agent-based model assumes little about the structure of ion channels in *E. coli*, other than they release potassium in response to a threshold potassium concentration in their environment. The agent based model is thus robust to the exact molecular details chosen and predicts the anomalous transport of the potassium wavefronts reasonably well the modelling was extended in a recent *Physical Review E* article(^4^). Such a description of reaction-anomalous diffusion phenomena has not to our knowledge been previously achieved in the literature^5^ and in general could be used to describe other signaling molecules.

1. Prindle, A.; Liu, J.; Asally, M.; Ly, S.; Garcia-Ojalvo, J.; Sudel, G. M., Ion channels enable electrical communication in bacterial communities. *Nature* 2015, *527*, 59.

2. Blee, J. A.; Roberts, I. S.; Waigh, T. A., Membrane potentials, oxidative stress and the dispersal response of bacterial biofilms to 405 nm light. *Physical Biology* 2020, *17*, 036001.

3. Milkman, R., An _E. col_i homologue of eukaryotic potassium channel proteins. *PNAS* 1994, *91*, 3510-3514.

4. Martorelli, V.; Akabuogu, E. U.; Krasovec, R.; Roberts, I. S.; Waigh, T. A., Electrical signaling in three-dimensional bacterial biofilms using an agent-based fire-diffuse-fire model. *Physical Review E* 2024, *109*, 054402.

5. Waigh, T. A.; Korabel, N., Heterogeneous anomalous transport in cellular and molecular biology. *Reports on Progress in Physics* 2023, *86*, 126601.

6. Hodgkin, A. L.; Huxley, A. F., A quantitative description of membrane current and its application to conduction and excitation in nerve. *Journal of Physiology*
**1952,**
*117*, 500.

7. Dawson, S. P.; Keizer, J.; Pearson, J. E., Fire-diffuse-fire model of dynamics of intracellular calcium waves. *PNAS*
**1999,**
*96*, 606.

8. Keener, J.; Sneyd, J., *Mathematical Physiology*. Springer: 2009.

9. Coombes, S., The effect of ion pumps on the speed of travelling waves in the fire-diffuse-fire model of Ca2+ release. *Bulletin of Mathematical Biology*
**2001,**
*63*, 1.

10. Blee, J. A.; Roberts, I. S.; Waigh, T. A., Spatial propagation of electrical signals in circular biofilms. *Physical Review E*
**2019,**
*100*, 052401.

11. Gorochowski, T. E.; Matyjaszkiewicz, A.; Todd, T.; Oak, N.; Kowalska, K., BSim: an agent-based tool for modelling bacterial populations in systems and synthetic biology. *PloS One*
**2012,**
*7*, 1.

12. Pena, A.; Sanchez, N. S.; Padilla-Garfias, F.; Ramiro-Cortes, Y.; Araiza-Villaneuva, M.; Calahorra, M., The use of thioflavin T for the estimation and measurement of the plasma membrane electric potential difference in different yeast strains. *Journal of Fungi*
**2023,**
*9* (9), 948.

13. Xue, C.; Lin, T. Y.; Chang, D.; Guo, Z., Thioflavin T as an amyloid dye: fibril quantification, optimal concentration and effect on aggregation. *Royal Society Open Science*
**2017,**
*4*, 160696.

14. Meisl, G.; Kirkegaard, J. B.; Arosio, P.; Michaels, T. C. T.; Vendruscolo, M.; Dobson, C. M.; Linse, S.; Knowles, T. P. J., Molecular mechanisms of protein aggregation from global fitting of kinetic models. *Nature Protocols*
**2016,**
*11* (2), 252-272.

15. Mancini, L.; Tian, T.; Guillaume, T.; Pu, Y.; Li, Y.; Lo, C. J.; Bai, F.; Pilizota, T., A general workflow for characterization of Nernstian dyes and their effects on bacterial physiology. *Biophysical Journal*
**2020,**
*118* (1), 4-14.

16. Buttress, J. A.; Halte, M.; Winkel, J. D. t.; Erhardt, M.; Popp, P. F.; Strahl, H., A guide for membrane potential measurements in Gram-negative bacteria using voltage-sensitive dyes. *Microbiology*
**2022,**
*168*, 001227.

17. Derk te Winkel, J.; Gray, D. A.; Seistrup, K. H.; Hamoen, L. W.; Strahl, H., Analysis of antimicrobial-triggered membrane depolarization using voltage sensitive dyes. *Frontiers in Cell and Developmental Biology*
**2016,**
*4*, 29.

18. Schawarzlander, M.; Logan, D. C.; Johnston, I. G.; Jones, N. S.; Meyer, A. J.; Fricker, M. D.; Sweetlove, L. J., Pulsing of membrane potential in individual mitochondria. *The Plant Cell*
**2012,**
*24*, 1188-1201.

19. Huser, J.; Blatter, L. A., Fluctuations in mitochondrial membrane potential caused by repetitive gating of the permeability transition pore. *Biochemistry Journal*
**1999,**
*343*, 311-317.

20. Mitchell, P., Coupling of phosphorylation to electron and hydrogen transfer by a chemi-osmotic type of mechanism. *Nature*
**1961,**
*191* (4784), 144-148.

21. Baba, T.; Ara, M.; Hasegawa, Y.; Takai, Y.; Okumura, Y.; Baba, M.; Datsenko, K. A.; Tomita, M.; Wanner, B. L.; Mori, H., Construction of *Escherichia coli* K-12 in-frame, single-gene knockout mutants: the Keio collection. *Molecular Systems Biology*
**2006,**
*2*, 1.

22. Schinedlin, J.; al, e., Fiji: an open-source platform for biological-image analysis. *Nature Methods*
**2012,**
*9*, 676.

23. Hartmann, R.; al, e., Quantitative image analysis of microbial communities with BiofilmQ. *Nature Microbiology*
**2021,**
*6* (2), 151.

The following is the authors’ response to the original reviews.

Critical synopsis of the articles cited by referee 2:(1) ‘Generalized workflow for characterization of Nernstian dyes and their effects on bacterial physiology’, L.Mancini et al, *Biophysical Journal*, 2020, 118, 1, 4-14.This is the central article used by referee 2 to argue that there are issues with the calibration of ThT for the measurement of membrane potentials. The authors use a simple Nernstian battery (SNB) model and unfortunately it is wrong when voltage-gated ion channels occur. Huge oscillations occur in the membrane potentials of *E. coli* that cannot be described by the SNB model. Instead a Hodgkin Huxley model is needed, as shown in our *eLife* manuscript and multiple other studies (see above). Arrhenius kinetics are assumed in the SNB model for pumping with no real evidence and the generalized workflow involves ripping the flagella off the bacteria! The authors construct an elaborate ‘work flow’ to insure their ThT results can be interpreted using their erroneous SNB model over a limited range of parameters.(2) ‘Non-equivalence of membrane voltage and ion-gradient as driving forces for the bacterial flagellar motor at low load’, C.J.Lo, et al, *Biophysical Journal*, 2007, 93, 1, 294.An odd *de novo* chimeric species is developed using an *E. coli* chassis which uses Na+ instead of H+ for the motility of its flagellar motor. It is not clear the relevance to wild type *E. coli*, due to the massive physiological perturbations involved. A SNB model is using to fit the data over a very limited parameter range with all the concomitant errors.(3) Single-cell bacterial electrophysiology reveals mechanisms of stress-induced damage’, E.Krasnopeeva, et al, *Biophysical Journal*, 2019, 116, 2390.The abstract says ‘PMF defines the physiological state of the cell’. This statement is hyperbolic. An extremely wide range of molecules contribute to the physiological state of a cell. PMF does not even define the electrophysiology of the cell e.g. via the membrane potential. There are 0.2 M of K+ compared with 0.0000001 M of H+ in *E. coli*, so K+ is arguably a million times more important for the membrane potential than H+ and thus the electrophysiology!Equation (1) in the manuscript assumes no other ions are exchanged during the experiments other than H+. This is a very bad approximation when voltage-gated potassium ion channels move the majority ion (K+) around!In our model Figure 4A is better explained by depolarisation due to K+ channels closing than direct irreversible photodamage. Why does the THT fluorescence increase again for the second hyperpolarization event if the THT is supposed to be damaged? It does not make sense.(4) ‘The proton motive force determines *E. coli* robustness to extracellular pH’, G.Terradot et al, 2024, preprint.This article expounds the SNB model once more. It still ignores the voltage-gated ion channels. Furthermore, it ignores the effect of the dominant ion in *E. coli*, K+. The manuscript is incorrect as a result and I would not recommend publication.In general, an important problem is being researched i.e. how the membrane potential of *E. coli* is related to motility, but there are serious flaws in the SNB approach and the experimental methodology appears tenuous.Answers to specific questions raised by the referees
**Reviewer #1 (Public Review):**
Summary:Cell-to-cell communication is essential for higher functions in bacterial biofilms. Electrical signals have proven effective in transmitting signals across biofilms. These signals are then used to coordinate cellular metabolisms or to increase antibiotic tolerance. Here, the authors have reported for the first time coordinated oscillation of membrane potential in *E. coli* biofilms that may have a functional role in photoprotection.Strengths:- The authors report original data.- For the first time, they showed that coordinated oscillations in membrane potential occur in *E. coli* biofilms.- The authors revealed a complex two-phase dynamic involving distinct molecular response mechanisms.- The authors developed two rigorous models inspired by (1) Hodgkin-Huxley model for the temporal dynamics of membrane potential and (2) Fire-Diffuse-Fire model for the propagation of the electric signal.- Since its discovery by comparative genomics, the Kch ion channel has not been associated with any specific phenotype in *E. coli*. Here, the authors proposed a functional role for the putative K+ Kch channel : enhancing survival under photo-toxic conditions.

We thank the referee for their positive evaluations and agree with these statements.

Weaknesses:- Since the flow of fresh medium is stopped at the beginning of the acquisition, environmental parameters such as pH and RedOx potential are likely to vary significantly during the experiment. It is therefore important to exclude the contributions of these variations to ensure that the electrical response is only induced by light stimulation. Unfortunately, no control experiments were carried out to address this issue.

The electrical responses occur almost instantaneously when the stimulation with blue light begins i.e. it is too fast to be a build of pH. We are not sure what the referee means by Redox potential since it is an attribute of all chemicals that are able to donate/receive electrons. The electrical response to stress appears to be caused by ROS, since when ROS scavengers are added the electrical response is removed i.e. pH plays a very small minority role if any.

- Furthermore, the control parameter of the experiment (light stimulation) is the same as that used to measure the electrical response, i.e. through fluorescence excitation. The use of the PROPS system could solve this problem.

>>We were enthusiastic at the start of the project to use the PROPs system in *E. coli* as presented by J.M.Krajl et al, ‘Electrical spiking in *E. coli* probed with a fluorescent voltage-indicating protein’, *Science*, 2011, 333, 6040, 345. However, the people we contacted in the microbiology community said that it had some technical issues and there have been no subsequent studies using PROPs in bacteria after the initial promising study. The fluorescent protein system recently presented in PNAS seems more promising, ‘Sensitive bacterial Vm sensors revealed the excitability of bacterial Vm and its role in antibiotic tolerance’, X.Jin et al, *PNAS*, 120, 3, e2208348120.

- Electrical signal propagation is an important aspect of the manuscript. However, a detailed quantitative analysis of the spatial dynamics within the biofilm is lacking. In addition, it is unclear if the electrical signal propagates within the biofilm during the second peak regime, which is mediated by the Kch channel. This is an important question, given that the fire-diffuse-fire model is presented with emphasis on the role of K+ ions.

We have presented a more detailed account of the electrical wavefront modelling work and it is currently under review in a physical journal, ‘Electrical signalling in three dimensional bacterial biofilms using an agent based fire-diffuse-fire model’, V.Martorelli, et al, 2024 https://www.biorxiv.org/content/10.1101/2023.11.17.567515v1

- Since deletion of the kch gene inhibits the long-term electrical response to light stimulation (regime II), the authors concluded that K+ ions play a role in the habituation response. However, Kch is a putative K+ ion channel. The use of specific drugs could help to clarify the role of K+ ions.

Our recent electrical impedance spectroscopy publication provides further evidence that Kch is associated with large changes in conductivity as expected for a voltage-gated ion channel https://pubs.acs.org/doi/10.1021/acs.nanolett.3c04446, 'Electrical impedance spectroscopy with bacterial biofilms: neuronal-like behavior', E.Akabuogu et al, *ACS Nanoletters*, 2024, in print.

- The manuscript as such does not allow us to properly conclude on the photo-protective role of the Kch ion channel.

That Kch has a photoprotective role is our current working hypothesis. The hypothesis fits with the data, but we are not saying we have proven it beyond all possible doubt.

- The link between membrane potential dynamics and mechanosensitivity is not captured in the equation for the Q-channel opening dynamics in the Hodgkin-Huxley model (Supp Eq 2).

Our model is agnostic with respect to the mechanosensitivity of the ion channels, although we deduce that mechanosensitive ion channels contribute to ion channel Q.

- Given the large number of parameters used in the models, it is hard to distinguish between prediction and fitting.

This is always an issue with electrophysiological modelling (compared with most heart and brain modelling studies we are very conservative in the choice of parameters for the bacteria). In terms of predicting the different phenomena observed, we believe the model is very successful.

**Reviewer #2 (Public Review):**
Summary of what the authors were trying to achieve:The authors thought they studied membrane potential dynamics in *E. coli* biofilms. They thought so because they were unaware that the dye they used to report that membrane potential in E.coli, has been previously shown not to report it. Because of this, the interpretation of the authors' results is not accurate.

We believe the Pilizota work is scientifically flawed.

Major strengths and weaknesses of the methods and results:The strength of this work is that all the data is presented clearly, and accurately, as far as I can tell.The major critical weakness of this paper is the use of ThT dye as a membrane potential dye in E.coli. The work is unaware of a publication from 2020 https://www.sciencedirect.com/science/article/pii/S0006349519308793 [sciencedirect.com] that demonstrates that ThT is not a membrane potential dye in *E. coli*. Therefore I think the results of this paper are misinterpreted. The same publication I reference above presents a protocol on how to carefully calibrate any candidate membrane potential dye in any given condition.

We are aware of this study, but believe it to be scientifically flawed. We do not cite the article because we do not think it is a particularly useful contribution to the literature.

I now go over each results section in the manuscript.Result section 1: Blue light triggers electrical spiking in single *E. coli* cellsI do not think the title of the result section is correct for the following reasons. The above-referenced work demonstrates the loading profile one should expect from a Nernstian dye (Figure 1). It also demonstrates that ThT does not show that profile and explains why is this so. ThT only permeates the membrane under light exposure (Figure 5). This finding is consistent with blue light peroxidising the membrane (see also following work Figure 4 https://www.sciencedirect.com/science/article/pii/S0006349519303923 [sciencedirect.com] on light-induced damage to the electrochemical gradient of protons-I am sure there are more references for this).

The Pilizota group invokes some elaborate artefacts to explain the lack of agreement with a simple Nernstian battery model. The model is incorrect not the fluorophore.

Please note that the loading profile (only observed under light) in the current manuscript in Figure 1B as well as in the video S1 is identical to that in Figure 3 from the above-referenced paper (i.e. https://www.sciencedirect.com/science/article/pii/S0006349519308793 [sciencedirect.com]), and corresponding videos S3 and S4. This kind of profile is exactly what one would expect theoretically if the light is simultaneously lowering the membrane potential as the ThT is equilibrating, see Figure S12 of that previous work. There, it is also demonstrated by the means of monitoring the speed of bacterial flagellar motor that the electrochemical gradient of protons is being lowered by the light. The authors state that applying the blue light for different time periods and over different time scales did not change the peak profile. This is expected if the light is lowering the electrochemical gradient of protons. But, in Figure S1, it is clear that it affected the timing of the peak, which is again expected, because the light affects the timing of the decay, and thus of the decay profile of the electrochemical gradient of protons (Figure 4 https://www.sciencedirect.com/science/article/pii/S0006349519303923 [sciencedirect.com]).

We think the proton effect is a million times weaker than that due to potasium i.e. 0.2 M K+ versus 10-7 M H+. We can comfortably neglect the influx of H+ in our experiments.

If find Figure S1D interesting. There authors load TMRM, which is a membrane voltage dye that has been used extensively (as far as I am aware this is the first reference for that and it has not been cited https://www.ncbi.nlm.nih.gov/pmc/articles/PMC1914430 [ncbi.nlm.nih.gov]/). As visible from the last TMRM reference I give, TMRM will only load the cells in Potassium Phosphate buffer with NaCl (and often we used EDTA to permeabilise the membrane). It is not fully clear (to me) whether here TMRM was prepared in rich media (it explicitly says so for ThT in Methods but not for TMRM), but it seems so. If this is the case, it likely also loads because of the damage to the membrane done with light, and therefore I am not surprised that the profiles are similar.

The vast majority of cells continue to be viable. We do not think membrane damage is dominating.

The authors then use CCCP. First, a small correction, as the authors state that it quenches membrane potential. CCCP is a protonophore (https://pubmed.ncbi.nlm.nih.gov/4962086 [pubmed.ncbi.nlm.nih.gov]/), so it collapses electrochemical gradient of protons. This means that it is possible, and this will depend on the type of pumps present in the cell, that CCCP collapses electrochemical gradient of protons, but the membrane potential is equal and opposite in sign to the DeltapH. So using CCCP does not automatically mean membrane potential will collapse (e.g. in some mammalian cells it does not need to be the case, but in *E. coli* it is https://www.biorxiv.org/content/10.1101/2021.11.19.469321v2 [biorxiv.org]). CCCP has also been recently found to be a substrate for TolC (https://journals.asm.org/doi/10.1128/mbio.00676-21 [journals.asm.org]), but at the concentrations the authors are using CCCP (100uM) that should not affect the results. However, the authors then state because they observed, in Figure S1E, a fast efflux of ions in all cells and no spiking dynamics this confirms that observed dynamics are membrane potential related. I do not agree that it does. First, Figure S1E, does not appear to show transients, instead, it is visible that after 50min treatment with 100uM CCCP, ThT dye shows no dynamics. The action of a Nernstian dye is defined. It is not sufficient that a charged molecule is affected in some way by electrical potential, this needs to be in a very specific way to be a Nernstian dye. Part of the profile of ThT loading observed in https://www.sciencedirect.com/science/article/pii/S0006349519308793 [sciencedirect.com] is membrane potential related, but not in a way that is characteristic of Nernstian dye.

Our understanding of the literature is CCCP poisons the whole metabolism of the bacterial cells. The ATP driven K+ channels will stop functioning and this is the dominant contributor to membrane potential.

Result section 2: Membrane potential dynamics depend on the intercellular distanceIn this chapter, the authors report that the time to reach the first intensity peak during ThT loading is different when cells are in microclusters. They interpret this as electrical signalling in clusters because the peak is reached faster in microclusters (as opposed to slower because intuitively in these clusters cells could be shielded from light). However, shielding is one possibility. The other is that the membrane has changed in composition and/or the effective light power the cells can tolerate (with mechanisms to handle light-induced damage, some of which authors mention later in the paper) is lower. Given that these cells were left in a microfluidic chamber for 2h hours to attach in growth media according to Methods, there is sufficient time for that to happen. In Figure S12 C and D of that same paper from my group (https://ars.els-cdn.com/content/image/1-s2.0-S0006349519308793-mmc6.pdf [ars.els-cdn.com]) one can see the effects of peak intensity and timing of the peak on the permeability of the membrane. Therefore I do not think the distance is the explanation for what authors observe.

Shielding would provide the reverse effect, since hyperpolarization begins in the dense centres of the biofilms. For the initial 2 hours the cells receive negligible blue light. Neither of the referee’s comments thus seem tenable.

Result section 3: Emergence of synchronized global wavefronts in *E. coli* biofilmsIn this section, the authors exposed a mature biofilm to blue light. They observe that the intensity peak is reached faster in the cells in the middle. They interpret this as the ion-channel-mediated wavefronts moved from the center of the biofilm. As above, cells in the middle can have different membrane permeability to those at the periphery, and probably even more importantly, there is no light profile shown anywhere in SI/Methods. I could be wrong, but the SI3 A profile is consistent with a potential Gaussian beam profile visible in the field of view. In Methods, I find the light source for the blue light and the type of microscope but no comments on how 'flat' the illumination is across their field of view. This is critical to assess what they are observing in this result section. I do find it interesting that the ThT intensity collapsed from the edges of the biofilms. In the publication I mentioned https://www.sciencedirect.com/science/article/pii/S0006349519308793#app2 [sciencedirect.com], the collapse of fluorescence was not understood (other than it is not membrane potential related). It was observed in Figure 5A, C, and F, that at the point of peak, electrochemical gradient of protons is already collapsed, and that at the point of peak cell expands and cytoplasmic content leaks out. This means that this part of the ThT curve is not membrane potential related. The authors see that after the first peak collapsed there is a period of time where ThT does not stain the cells and then it starts again. If after the first peak the cellular content leaks, as we have observed, then staining that occurs much later could be simply staining of cytoplasmic positively charged content, and the timing of that depends on the dynamics of cytoplasmic content leakage (we observed this to be happening over 2h in individual cells). ThT is also a non-specific amyloid dye, and in starving *E. coli* cells formation of protein clusters has been observed (https://pubmed.ncbi.nlm.nih.gov/30472191 [pubmed.ncbi.nlm.nih.gov]/), so such cytoplasmic staining seems possible.

>>It is very easy to see if the illumination is flat (Köhler illumination) by comparing the intensity of background pixels on the detector. It was flat in our case. Protons have little to do with our work for reasons highlighted before. Differential membrane permittivity is a speculative phenomenon not well supported by any evidence and with no clear molecular mechanism.

Finally, I note that authors observe biofilms of different shapes and sizes and state that they observe similar intensity profiles, which could mean that my comment on 'flatness' of the field of view above is not a concern. However, the scale bar in Figure 2A is not legible, so I can't compare it to the variation of sizes of the biofilms in Figure 2C (67 to 280um). Based on this, I think that the illumination profile is still a concern.

The referee now contradicts themselves and wants a scale bar to be more visible. We have changed the scale bar.

Result section 4: Voltage-gated Kch potassium channels mediate ion-channel electrical oscillations in *E. coli*First I note at this point, given that I disagree that the data presented thus 'suggest that *E. coli* biofilms use electrical signaling to coordinate long-range responses to light stress' as the authors state, it gets harder to comment on the rest of the results.In this result section the authors look at the effect of Kch, a putative voltage-gated potassium channel, on ThT profile in *E. coli* cells. And they see a difference. It is worth noting that in the publication https://www.sciencedirect.com/science/article/pii/S0006349519308793 [sciencedirect.com] it is found that ThT is also likely a substrate for TolC (Figure 4), but that scenario could not be distinguished from the one where TolC mutant has a different membrane permeability (and there is a publication that suggests the latter is happening https://onlinelibrary.wiley.com/doi/10.1111/j.1365-2958.2010.07245.x [onlinelibrary.wiley.com]). Given this, it is also possible that Kch deletion affects the membrane permeability. I do note that in video S4 I seem to see more of, what appear to be, plasmolysed cells. The authors do not see the ThT intensity with this mutant that appears long after the initial peak has disappeared, as they see in WT. It is not clear how long they waited for this, as from Figure S3C it could simply be that the dynamics of this is a lot slower, e.g. Kch deletion changes membrane permeability.

The work that TolC provides a possible passive pathway for ThT to leave cells seems slightly niche. It just demonstrates another mechanism for the cells to equilibriate the concentrations of ThT in a Nernstian manner i.e. driven by the membrane voltage.

The authors themselves state that the evidence for Kch being a voltage-gated channel is indirect (line 54). I do not think there is a need to claim function from a ThT profile of *E. coli* mutants (nor do I believe it's good practice), given how accurate single-channel recordings are currently. To know the exact dependency on the membrane potential, ion channel recordings on this protein are needed first.

We have good evidence form electrical impedance spectroscopy experiments that Kch increases the conductivity of biofilms https://pubs.acs.org/doi/10.1021/acs.nanolett.3c04446, 'Electrical impedance spectroscopy with bacterial biofilms: neuronal-like behavior', E.Akabuogu et al, *ACS Nanoletters*, 2024, in print.

Result section 5: Blue light influences ion-channel mediated membrane potential events in *E. coli*In this chapter the authors vary the light intensity and stain the cells with PI (this dye gets into the cells when the membrane becomes very permeable), and the extracellular environment with K+ dye (I have not yet worked carefully with this dye). They find that different amounts of light influence ThT dynamics. This is in line with previous literature (both papers I have been mentioning: Figure 4 https://www.sciencedirect.com/science/article/pii/S0006349519303923 [sciencedirect.com] and https://ars.els-cdn.com/content/image/1-s2.0-S0006349519308793-mmc6.pdf [ars.els-cdn.com] especially SI12), but does not add anything new. I think the results presented here can be explained with previously published theory and do not indicate that the ion-channel mediated membrane potential dynamics is a light stress relief process.

The simple Nernstian battery model proposed by Pilizota et al is erroneous in our opinion for reasons outlined above. We believe it will prove to be a dead end for bacterial electrophysiology studies.

Result section 6: Development of a Hodgkin-Huxley model for the observed membrane potential dynamicsThis results section starts with the authors stating: 'our data provide evidence that *E. coli* manages light stress through well-controlled modulation of its membrane potential dynamics'. As stated above, I think they are instead observing the process of ThT loading while the light is damaging the membrane and thus simultaneously collapsing the electrochemical gradient of protons. As stated above, this has been modelled before. And then, they observe a ThT staining that is independent from membrane potential.

This is an erroneous niche opinion. Protons have little say in the membrane potential since there are so few of them. The membrane potential is mostly determined by K+.

I will briefly comment on the Hodgkin Huxley (HH) based model. First, I think there is no evidence for two channels with different activation profiles as authors propose. But also, the HH model has been developed for neurons. There, the leakage and the pumping fluxes are both described by a constant representing conductivity, times the difference between the membrane potential and Nernst potential for the given ion. The conductivity in the model is given as gK*n^4 for potassium, gNa*m^3*h sodium, and gL for leakage, where gK, gNa and gL were measured experimentally for neurons. And, n, m, and h are variables that describe the experimentally observed voltage-gated mechanism of neuronal sodium and potassium channels. (Please see Hodgkin AL, Huxley AF. 1952. Currents carried by sodium and potassium ions through the membrane of the giant axon of Loligo. J. Physiol. 116:449-72 and Hodgkin AL, Huxley AF. 1952. A quantitative description of membrane current and its application to conduction and excitation in nerve. J. Physiol. 117:500-44).

In the 70 years since Hodgkin and Huxley first presented their model, a huge number of similar models have been proposed to describe cellular electrophysiology. We are not being hyperbolic when we state that the HH models for excitable cells are like the Schrödinger equation for molecules. We carefully adapted our HH model to reflect the currently understood electrophysiology of *E. coli*.

Thus, in applying the model to describe bacterial electrophysiology one should ensure near equilibrium requirement holds (so that (V-VQ) etc terms in authors' equation Figure 5 B hold), and potassium and other channels in a given bacterium have similar gating properties to those found in neurons. I am not aware of such measurements in any bacteria, and therefore think the pump leak model of the electrophysiology of bacteria needs to start with fluxes that are more general (for example Keener JP, Sneyd J. 2009. Mathematical physiology: I: Cellular physiology. New York: Springer or https://journals.plos.org/plosone/article?id=10.1371/journal.pone.0000144 [journals.plos.org])

The reference is to a slightly more modern version of a simple Nernstian battery model. The model will not oscillate and thus will not help modelling membrane potentials in bacteria. We are unsure where the equilibrium requirement comes from (inadequate modelling of the dynamics?)

Result section 7: Mechanosensitive ion channels (MS) are vital for the first hyperpolarization event in *E. coli*.The results that Mcs channels affect the profile of ThT dye are interesting. It is again possible that the membrane permeability of these mutants has changed and therefore the dynamics have changed, so this needs to be checked first. I also note that our results show that the peak of ThT coincides with cell expansion. For this to be understood a model is needed that also takes into account the link between maintenance of electrochemical gradients of ions in the cell and osmotic pressure.

The evidence for permeability changes in the membranes seems to be tenuous.

A side note is that the authors state that the Msc responds to stress-related voltage changes. I think this is an overstatement. Mscs respond to predominantly membrane tension and are mostly nonspecific (see how their action recovers cellular volume in this publication https://www.pnas.org/doi/full/10.1073/pnas.1522185113 [pnas.org]). Authors cite references 35-39 to support this statement. These publications still state that these channels are predominantly membrane tension-gated. Some of the references state that the presence of external ions is important for tension-related gating but sometimes they gate spontaneously in the presence of certain ions. Other publications cited don't really look at gating with respect to ions (39 is on clustering). This is why I think the statement is somewhat misleading.

We have reworded the discussion of Mscs since the literature appears to be ambiguous. We will try to run some electrical impedance spectroscopy experiments on the Msc mutants in the future to attempt to remove the ambiguity.

Result section 8: Anomalous ion-channel-mediated wavefronts propagate light stress signals in 3D *E. coli* biofilms.I am not commenting on this result section, as it would only be applicable if ThT was membrane potential dye in *E. coli*.

Ok, but we disagree on the use of ThT.

Aims achieved/results support their conclusions:The authors clearly present their data. I am convinced that they have accurately presented everything they observed. However, I think their interpretation of the data and conclusions is inaccurate in line with the discussion I provided above.Likely impact of the work on the field, and the utility of the methods and data to the community:I do not think this publication should be published in its current format. It should be revised in light of the previous literature as discussed in detail above. I believe presenting it in it's current form on eLife pages would create unnecessary confusion.

We believe many of the Pilizota group articles are scientifically flawed and are causing the confusion in the literature.

Any other comments:I note, that while this work studies *E. coli*, it references papers in other bacteria using ThT. For example, in lines 35-36 authors state that bacteria (*Bacillus subtilis* in this case) in biofilms have been recently found to modulate membrane potential citing the relevant literature from 2015. It is worth noting that the most recent paper https://journals.asm.org/doi/10.1128/mbio.02220-23 [journals.asm.org] found that ThT binds to one or more proteins in the spore coat, suggesting that it does not act as a membrane potential in Bacillus spores. It is possible that it still reports membrane potential in Bacillus cells and the recent results are strictly spore-specific, but these should be kept in mind when using ThT with Bacillus.

>>ThT was used successfully in previous studies of normal *B. subtilis* cells (by our own group and A.Prindle, ‘Spatial propagation of electrical signal in circular biofilms’, J.A.Blee et al, *Physical Review E*, 2019, 100, 052401, J.A.Blee et al, ‘Membrane potentials, oxidative stress and the dispersal response of bacterial biofilms to 405 nm light’, *Physical Biology*, 2020, 17, 2, 036001, A.Prindle et al, ‘Ion channels enable electrical communication in bacterial communities’, *Nature*, 2015, 527, 59-63). The connection to low metabolism pore research seems speculative.

**Reviewer #3 (Public Review):**
It has recently been demonstrated that bacteria in biofilms show changes in membrane potential in response to changes in their environment, and that these can propagate signals through the biofilm to coordinate bacterial behavior. Akabuogu et al. contribute to this exciting research area with a study of blue light-induced membrane potential dynamics in *E. coli* biofilms. They demonstrate that Thioflavin-T (ThT) intensity (a proxy for membrane potential) displays multiphasic dynamics in response to blue light treatment. They additionally use genetic manipulations to implicate the potassium channel Kch in the latter part of these dynamics. Mechanosensitive ion channels may also be involved, although these channels seem to have blue light-independent effects on membrane potential as well. In addition, there are challenges to the quantitative interpretation of ThT microscopy data which require consideration. The authors then explore whether these dynamics are involved in signaling at the community level. The authors suggest that cell firing is both more coordinated when cells are clustered and happens in waves in larger, 3D biofilms; however, in both cases evidence for these claims is incomplete. The authors present two simulations to describe the ThT data. The first of these simulations, a Hodgkin-Huxley model, indicates that the data are consistent with the activity of two ion channels with different kinetics; the Kch channel mutant, which ablates a specific portion of the response curve, is consistent with this. The second model is a fire-diffuse-fire model to describe wavefront propagation of membrane potential changes in a 3D biofilm; because the wavefront data are not presented clearly, the results of this model are difficult to interpret. Finally, the authors discuss whether these membrane potential changes could be involved in generating a protective response to blue light exposure; increased death in a Kch ion channel mutant upon blue light exposure suggests that this may be the case, but a no-light control is needed to clarify this.In a few instances, the paper is missing key control experiments that are important to the interpretation of the data. This makes it difficult to judge the meaning of some of the presented experiments.(1) An additional control for the effects of autofluorescence is very important. The authors conduct an experiment where they treat cells with CCCP and see that Thioflavin-T (ThT) dynamics do not change over the course of the experiment. They suggest that this demonstrates that autofluorescence does not impact their measurements. However, cellular autofluorescence depends on the physiological state of the cell, which is impacted by CCCP treatment. A much simpler and more direct experiment would be to repeat the measurement in the absence of ThT or any other stain. This experiment should be performed both in the wild-type strain and in the ∆kch mutant.

ThT is a very bright fluorophore (much brighter than a GFP). It is clear from the images of non-stained samples that autofluorescence provides a negligible contribution to the fluorescence intensity in an image.

(2) The effects of photobleaching should be considered. Of course, the intensity varies a lot over the course of the experiment in a way that photobleaching alone cannot explain. However, photobleaching can still contribute to the kinetics observed. Photobleaching can be assessed by changing the intensity, duration, or frequency of exposure to excitation light during the experiment. Considerations about photobleaching become particularly important when considering the effect of catalase on ThT intensity. The authors find that the decrease in ThT signal after the initial "spike" is attenuated by the addition of catalase; this is what would be predicted by catalase protecting ThT from photobleaching (indeed, catalase can be used to reduce photobleaching in time lapse imaging).

Photobleaching was negligible over the course of the experiments. We employed techniques such as reducing sample exposure time and using the appropriate light intensity to minimize photobleaching.

(3) It would be helpful to have a baseline of membrane potential fluctuations in the absence of the proposed stimulus (in this case, blue light). Including traces of membrane potential recorded without light present would help support the claim that these changes in membrane potential represent a blue light-specific stress response, as the authors suggest. Of course, ThT is blue, so if the excitation light for ThT is problematic for this experiment the alternative dye tetramethylrhodamine methyl ester perchlorate (TMRM) can be used instead.

Unfortunately the fluorescent baseline is too weak to measure cleanly in this experiment. It appears the collective response of all the bacteria hyperpolarization at the same time appears to dominate the signal (measurements in the eLife article and new potentiometry measurements).

(4) The effects of ThT in combination with blue light should be more carefully considered. In mitochondria, a combination of high concentrations of blue light and ThT leads to disruption of the PMF (Skates et al. 2021 BioRXiv), and similarly, ThT treatment enhances the photodynamic effects of blue light in *E. coli* (Bondia et al. 2021 Chemical Communications). If present in this experiment, this effect could confound the interpretation of the PMF dynamics reported in the paper.

We think the PMF plays a minority role in determining the membrane potential in *E. coli*. For reasons outlined before (H+ is a minority ion in *E. coli* compared with K+).

(5) Figures 4D - E indicate that a ∆kch mutant has increased propidium iodide (PI) staining in the presence of blue light; this is interpreted to mean that Kch-mediated membrane potential dynamics help protect cells from blue light. However, Live/Dead staining results in these strains in the absence of blue light are not reported. This means that the possibility that the ∆kch mutant has a general decrease in survival (independent of any effects of blue light) cannot be ruled out.

>>Both strains of bacterial has similar growth curve and also engaged in membrane potential dynamics for the duration of the experiment. We were interested in bacterial cells that observed membrane potential dynamics in the presence of the stress. Bacterial cells need to be alive to engage in membrane potential dynamics (hyperpolarize) under stress conditions. Cells that engaged in membrane potential dynamics and later stained red were only counted after the entire duration. We believe that the wildtype handles the light stress better than the ∆kch mutant as measured with the PI.

(6) Additionally in Figures 4D - E, the interpretation of this experiment can be confounded by the fact that PI uptake can sometimes be seen in bacterial cells with high membrane potential (Kirchhoff & Cypionka 2017 J Microbial Methods); the interpretation is that high membrane potential can lead to increased PI permeability. Because the membrane potential is largely higher throughout blue light treatment in the ∆kch mutant (Fig. 3AB), this complicates the interpretation of this experiment.

Kirchhoff & Cypionka 2017 J Microbial Methods, using fluorescence microscopy, suggested that changes in membrane potential dynamics can introduce experimental bias when propidium iodide is used to confirm the viability of tge bacterial strains, *B subtilis* (DSM-10) and *Dinoroseobacter shibae,* that are starved of oxygen (via N2 gassing) for 2 hours. They attempted to support their findings by using CCCP in stopping the membrane potential dynamics (but never showed any pictoral or plotted data for this confirmatory experiment). In our experiment methodology, cell death was not forced on the cells by introducing an extra burden or via anoxia. We believe that the accumulation of PI in ∆kch mutant is not due to high membrane potential dynamics but is attributed to the PI, unbiasedly showing damaged/dead cells. We think that propidium iodide is good for this experiment. Propidium iodide is a dye that is extensively used in life sciences. PI has also been used in the study of bacterial electrophysiology (https://pubmed.ncbi.nlm.nih.gov/32343961/,) and no membrane potential related bias was reported.

Throughout the paper, many ThT intensity traces are compared, and described as "similar" or "dissimilar", without detailed discussion or a clear standard for comparison. For example, the two membrane potential curves in Fig. S1C are described as "similar" although they have very different shapes, whereas the curves in Fig. 1B and 1D are discussed in terms of their differences although they are evidently much more similar to one another. Without metrics or statistics to compare these curves, it is hard to interpret these claims. These comparative interpretations are additionally challenging because many of the figures in which average trace data are presented do not indicate standard deviation.

Comparison of small changes in the absolute intensities is problematic in such fluorescence experiments. We mean the shape of the traces is similar and they can be modelled using a HH model with similar parameters.

The differences between the TMRM and ThT curves that the authors show in Fig. S1C warrant further consideration. Some of the key features of the response in the ThT curve (on which much of the modeling work in the paper relies) are not very apparent in the TMRM data. It is not obvious to me which of these traces will be more representative of the actual underlying membrane potential dynamics.

In our experiment, TMRM was used to confirm the dynamics observed using ThT. However, ThT appear to be more photostable than TMRM (especially towars the 2nd peak). The most interesting observation is that with both dyes, all phases of the membrane potential dynamics were conspicuous (the first peak, the quiescent period and the second peak). The time periods for these three episodes were also similar.

A key claim in this paper (that dynamics of firing differ depending on whether cells are alone or in a colony) is underpinned by "time-to-first peak" analysis, but there are some challenges in interpreting these results. The authors report an average time-to-first peak of 7.34 min for the data in Figure 1B, but the average curve in Figure 1B peaks earlier than this. In Figure 1E, it appears that there are a handful of outliers in the "sparse cell" condition that likely explain this discrepancy. Either an outlier analysis should be done and the mean recomputed accordingly, or a more outlier-robust method like the median should be used instead. Then, a statistical comparison of these results will indicate whether there is a significant difference between them.

The key point is the comparison of standard errors on the standard deviation.

In two different 3D biofilm experiments, the authors report the propagation of wavefronts of membrane potential; I am unable to discern these wavefronts in the imaging data, and they are not clearly demonstrated by analysis.The first data set is presented in Figures 2A, 2B, and Video S3. The images and video are very difficult to interpret because of how the images have been scaled: the center of the biofilm is highly saturated, and the zero value has also been set too high to consistently observe the single cells surrounding the biofilm. With the images scaled this way, it is very difficult to assess dynamics. The time stamps in Video S3 and on the panels in Figure 2A also do not correspond to one another although the same biofilm is shown (and the time course in 2B is also different from what is indicated in 2B). In either case, it appears that the center of the biofilm is consistently brighter than the edges, and the intensity of all cells in the biofilm increases in tandem; by eye, propagating wavefronts (either directed toward the edge or the center) are not evident to me. Increased brightness at the center of the biofilm could be explained by increased cell thickness there (as is typical in this type of biofilm). From the image legend, it is not clear whether the image presented is a single confocal slice or a projection. Even if this is a single confocal slice, in both Video S3 and Figure 2A there are regions of "haze" from out-of-focus light evident, suggesting that light from other focal planes is nonetheless present. This seems to me to be a simpler explanation for the fluorescence dynamics observed in this experiment: cells are all following the same trajectory that corresponds to that seen for single cells, and the center is brighter because of increased biofilm thickness.

We appreciate the reviewer for this important observation. We have made changes to the figures to address this confusion. The cell cover has no influence on the observed membrane potential dynamics. The entire biofilm was exposed to the same blue light at each time. Therefore all parts of the biofilm received equal amounts of the blue light intensity. The membrane potential dynamics was not influenced by cell density (see Fig 2C).

The second data set is presented in Video S6B; I am similarly unable to see any wave propagation in this video. I observe only a consistent decrease in fluorescence intensity throughout the experiment that is spatially uniform (except for the bright, dynamic cells near the top; these presumably represent cells that are floating in the microfluidic and have newly arrived to the imaging region).

A visual inspection of Video S6B shows a fast rise, a decrease in fluorescence and a second rise (supplementary figure 4B). The data for the fluorescence was carefully obtained using the imaris software. We created a curved geometry on each slice of the confocal stack. We analyzed the surfaces of this curved plane along the z-axis. This was carried out in imaris.

3D imaging data can be difficult to interpret by eye, so it would perhaps be more helpful to demonstrate these propagating wavefronts by analysis; however, such analysis is not presented in a clear way. The legend in Figure 2B mentions a "wavefront trace", but there is no position information included - this trace instead seems to represent the average intensity trace of all cells. To demonstrate the propagation of a wavefront, this analysis should be shown for different subpopulations of cells at different positions from the center of the biofilm. Data is shown in Figure 8 that reflects the velocity of the wavefront as a function of biofilm position; however, because the wavefronts themselves are not evident in the data, it is difficult to interpret this analysis. The methods section additionally does not contain sufficient information about what these velocities represent and how they are calculated. Because of this, it is difficult for me to evaluate the section of the paper pertaining to wave propagation and the predicted biofilm critical size.

The analysis is considered in more detail in a more expansive modelling article, currently under peer review in a physics journal, ‘Electrical signalling in three dimensional bacterial biofilms using an agent based fire-diffuse-fire model’, V.Martorelli, et al, 2024 https://www.biorxiv.org/content/10.1101/2023.11.17.567515v1

There are some instances in the paper where claims are made that do not have data shown or are not evident in the cited data:(1) In the first results section, "When CCCP was added, we observed a fast efflux of ions in all cells"- the data figure pertaining to this experiment is in Fig. S1E, which does not show any ion efflux. The methods section does not mention how ion efflux was measured during CCCP treatment.

We have worded this differently to properly convey our results.

(2) In the discussion of voltage-gated calcium channels, the authors refer to "spiking events", but these are not obvious in Figure S3E. Although the fluorescence intensity changes over time, it's hard to distinguish these fluctuations from measurement noise; a no-light control could help clarify this.

The calcium transients observed were not due to noise or artefacts.

(3) The authors state that the membrane potential dynamics simulated in Figure 7B are similar to those observed in 3D biofilms in Fig. S4B; however, the second peak is not clearly evident in Fig. S4B and it looks very different for the mature biofilm data reported in Fig. 2. I have some additional confusion about this data specifically: in the intensity trace shown in Fig. S4B, the intensity in the second frame is much higher than the first; this is not evident in Video S6B, in which the highest intensity is in the first frame at time 0. Similarly, the graph indicates that the intensity at 60 minutes is higher than the intensity at 4 minutes, but this is not the case in Fig. S4A or Video S6B.

The confusion stated here has now been addressed. Also it should be noted that while Fig 2.1 was obtained with LED light source, Fig S4A was obtained using a laser light source. While obtaining the confocal images (for Fig S4A), the light intensity was controlled to further minimize photobleaching. Most importantly, there is an evidence of slow rise to the 2nd peak in Fig S4B. The first peak, quiescence and slow rise to second peak are evident.

**Recommendations for the authors:**

**Reviewer #1 (Recommendations For The Authors):**
Scientific recommendations:- Although Fig 4A clearly shows that light stimulation has an influence on the dynamics of cell membrane potential in the biofilm, it is important to rule out the contribution of variations in environmental parameters. I understand that for technical reasons, the flow of fresh medium must be stopped during image acquisition. Therefore, I suggest performing control experiments, where the flow is stopped before image acquisition (15min, 30min, 45min, and 1h before). If there is no significant contribution from environmental variations (pH, RedOx), the dynamics of the electrical response should be superimposed whatever the delay between stopping the flow stop and switching on the light.

In this current research study, we were focused on studying how *E. coli* cells and biofilms react to blue light stress via their membrane potential dynamics. This involved growing the cells and biofilms, stopping the media flow and obtaining data immediately. We believe that stopping the flow not only helped us to manage data acquisition, it also helped us reduce the effect of environmental factors. In our future study we will expand the work to include how the membrane potential dynamics evolve in the presence of changing environmental factors for example such induced by stopping the flow at varied times.

- Since TMRM signal exhibits a linear increase after the first response peak (Supplementary Figure 1D), I recommend mitigating the statement at line 78.- To improve the spatial analysis of the electrical response, I suggest plotting kymographs of the intensity profiles across the biofilm. I have plotted this kymograph for Video S3 and it appears that there is no electrical propagation for the second peak. In addition, the authors should provide technical details of how R^2(t) is measured in the first regime (Figure 7E).

See the dedicated simulation article for more details. https://www.biorxiv.org/content/10.1101/2023.11.17.567515v1

- Line 152: To assess the variability of the latency, the authors should consider measuring the variance divided by the mean instead of SD, which may depend on the average value.

We are happy with our current use of standard error on the standard deviation. It shows what we claim to be true.

- Line 154-155: To truly determine whether the amplitude of the "action potential" is independent of biofilm size, the authors should not normalise the signals.

Good point. We qualitatively compared both normalized and unnormalized data. Recent electrical impedance spectroscopy measurements (unpublished) indicate that the electrical activity is an extensive quantity i.e. it scales with the size of the biofilms.

- To precise the role of K+ in the habituation response, I suggest using valinomycin at sub-inhibitory concentrations (10µM). Besides, the high concentration of CCCP used in this study completely inhibits cell activity. Not surprisingly, no electrical response to light stimulation was observed in the presence of CCCP. Finally, the Kch complementation experiment exhibits a "drop after the first peak" on a single point. It would be more convincing to increase the temporal resolution (1min->10s) to show that there is indeed a first and a second peak.

An interesting experiment for the future.

- Line 237-238: There are only two points suggesting that the dynamics of hyperpolarization are faster at higher irradiance(Fig 4A). The authors should consider adding a third intermediate point at 17µW/mm^2 to confirm the statement made in this sentence.

Multiple repeats were performed. We are confident of the robustness of our data.

- Line 249 + Fig 4E: It seems that the data reported on Fig 4E are extracted from Fig 4D. If this is indeed the case, the data should be normalised by the total population size to compare survival probabilities under the two conditions. It would also be great to measure these probabilities (for WT and ∆kch) in the presence of ROS scavengers.- To distinguish between model fitting and model predictions, the authors should clearly state which parameters are taken from the literature and which parameters are adjusted to fit the experimental data.- Supplementary Figure 4A: why can't we see any wavefront in this series of images?

For the experimental data, the wavefront was analyzed by employing the imaris software. We systematically created a ROI with a curved geometry within the confocal stack (the biofilm). The fluorescence of ThT was traced along the surface of the curved geometry was analyzed along the z-axis.

- Fig 7B: Could the authors explain why the plateau is higher in the simulations than in the biofilm experiments? Could they add noise on the firing activities?

See the dedicated Martorelli modelling article. In general we would need to approach stochastic Hodgkin-Huxley modelling and the fluorescence data (and electrical impedance spectroscopy data) presented does not have extensive noise (due to collective averaging over many bacteria cells).

- Supplementary Figure 4B: Why can't we see the second peak in confocal images?

The second peak is present although not as robust as in Fig 2B. The confocal images were obtained with a laser source. Therefore we tried to create a balance between applying sufficient light stress on the bacterial cells and mitigating photobleaching.

Editing recommendations:

The editing recommendations below has been applied where appropriate

- Many important technical details are missing (e.g. R^2, curvature, and 445nm irradiance measurements). Error bars are missing from most graphs. The captions should clearly indicate if these are single-cell or biofilm experiments, strain name, illumination conditions, number of experiments, SD, or SE. Please indicate on all panels of all figures in the main text and in the supplements, which are the conditions: single cell vs. biofilm, strains, medium, centrifugal vs centripetal etc..., where relevant. Please also draw error bars everywhere.

We have now made appropriate changes. We specifically use cells when we were dealing with single cells and biofilms when we worked on biofilms. We decided to describe the strain name either on the panel or the image description.

- Line 47-51: The way the paragraph is written suggests that no coordinated electrical oscillations have been observed in Gram-negative biofilms. However, Hennes et al (referenced as 57 in this manuscript) have shown that a wave of hyperpolarized cells propagates in Neisseria gonorrhoea colony, which is a Gram-negative bacterium.

We are now aware of this work. It was not published when we first submitted our work and the authors claim the waves of activity are due to ROS diffusion NOT propagating waves of ions (coordinated electrical wavefronts).

- Line 59: "stressor" -> "stress" or "perturbation".

The correction has been made.

- Line 153: Please indicate in the Material&Methods how the size of the biofilm is measured.

The biofilm size was obtained using BiofilmQ and the step by step guide for using BiofilmQ were stated..

- Figure 2A: Please provide associated brightfield images to locate bacteria.- Line 186: Please remove "wavefront" from the caption. Fig2B only shows the average signal as a function of time.

This correction has been implemented.

- Fig 3B,C: Please indicate single cell and biofilm on the panels and also WT and ∆kch.- Line 289: I suggest adding "in single cell experiments" to the title of this section.- Fig 5A: blue light is always present at regular time intervals during regime I and II. The presence of blue light only in regime I could be misleading.- Fig 5C: The curve in Fig 5D seems to correspond to the biofilm case. The curve given by the model, should be compared with the average curve presented in Fig 1D.- Fig 6A, B, and C: These figures could be moved to supplements.- Line 392: Replace "turgidity" with "turgor pressure".- Fig 7C,E: Please use a log-log scale to represent these data and indicate the line of slope 1.- Fig 7E: The x-axis has been cropped.- Please provide a supplementary movie for the data presented in Fig 7E.- Line 455: *E. coli* biofilms do not express ThT.- Line 466: "\gamma is the anomalous exponent". Please remove anomalous (\gamma can equal 1 at this stage).- Line 475: Please replace "section" with "projection".- Line 476: Please replace "spatiotemporal" with "temporal". There is no spatial dependency in either figure.- Line 500: Please define Eikonal approximation.- Fig 8 could be moved to supplements.- Line 553: "predicted" -> "predict".- Line 593: Could the authors explain why their model offers much better quantitative agreement?- Line 669: What does "universal" mean in that context?- Line 671: A volume can be pipetted but not a concentration.- Line 676: Are triplicates technical or biological replicates?- Sup Fig1: Please use minutes instead of seconds in panel A.- Model for membrane dynamics: "The fraction of time the Q+ channel is open" -> "The dynamics of Q+ channel activity can be written". Ditto for K+ channel...- Model for membrane dynamics: "the term ... is a threshold-linear". This function is not linear at all. Why is it called linear? Also, please describe what \sigma is.- ABFDF model: "releasing a given concentration" -> "releasing a local concentration" or "a given number" but it's not \sigma anymore. Besides, this \sigma is unlikely related to the previous \sigma used in the model of membrane potential dynamics in single cells. Please consider renaming one or the other. Also, ions are referred to as C+ in the text and C in equation 8. Am I missing something?
**Reviewer #2 (Recommendations For The Authors):**
I have included all my comments as one review. I have done so, despite the fact that some minor comments could have gone into this section, because I decided to review each Result section. I thus felt that not writing it as one review might be harder to follow. I have however highlighted which comments are minor suggestions or where I felt corrections.However, while I am happy with all my comments being public, given their nature I think they should be shown to authors first. Perhaps the authors want to go over them and think about it before deciding if they are happy for their manuscript to be published along with these comments, or not. I will highlight this in an email to the editor. I question whether in this case, given that I am raising major issues, publishing both the manuscript and the comments is the way to go as I think it might just generate confusion among the audience.
**Reviewer #3 (Recommendations For The Authors):**
I was unable to find any legends for any of the supplemental videos in my review materials, and I could not open supplemental video 5.I made some comments in the public review about the analysis and interpretation of the time-to-fire data. One of the other challenges in this data set is that the time resolution is limited- it seems that a large proportion of cells have already fired after a single acquisition frame. It would be ideal to increase the time resolution on this measurement to improve precision. This could be done by imaging more quickly, but that would perhaps necessitate more blue light exposure; an alternative is to do this experiment under lower blue light irradiance where the first spike time is increased (Figure 4A).In the public review, I mentioned the possible impact of high membrane potential on PI permeability. To address this, the experiment could be repeated with other stains, or the viability of blue light-treated cells could be addressed more directly by outgrowth or colony-forming unit assays.In the public review, I mentioned the possible combined toxicity of ThT and blue light. Live/dead experiments after blue light exposure with and without ThT could be used to test for such effects, and/or the growth curve experiment in Figure 1F could be repeated with blue light exposure at a comparable irradiance used in the experiment.Throughout the paper and figure legends, it would help to have more methodological details in the main text, especially those that are critical for the interpretation of the experiment. The experimental details in the methods section are nicely described, but the data analysis section should be expanded significantly.At the end of the results section, the authors suggest a critical biofilm size of only 4 µm for wavefront propagation (not much larger than a single cell!). The authors show responses for various biofilm sizes in Fig. 2C, but these are all substantially larger. Are there data for cell clusters above and below this size that could support this claim more directly?The authors mention image registration as part of their analysis pipeline, but the 3D data sets in Video S6B and Fig. S4A do not appear to be registered- were these registered prior to the velocity analysis reported in Fig. 8?One of the most challenging claims to demonstrate in this paper is that these membrane potential wavefronts are involved in coordinating a large, biofilm-scale response to blue light. One possible way to test this might be to repeat the Live/Dead experiment in planktonic culture or the single-cell condition. If the protection from blue light specifically emerges due to coordinated activity of the biofilm, the Kch mutant would not be expected to show a change in Live/Dead staining in non-biofilm conditions.Line 140: How is "mature biofilm" defined? Also on this same line, what does "spontaneous" mean here?Line 151: "much smaller": Given that the reported time for 3D biofilms is 2.73 {plus minus} 0.85 min and in microclusters is 3.27 {plus minus} 1.77 min, this seems overly strong.Line 155: How is "biofilm density" characterized? Additionally, the data in Figure 2C are presented in distance units (µm), but the text refers to "areal coverage"- please define the meaning of these distance units in the legend and/or here in the text (is this the average radius?).Lines 161-162: These claims seem strong given the data presented before, and the logic is not very explicit. For example, in the second sentence, the idea that this signaling is used to "coordinate long-range responses to light stress" does not seem strongly evidenced at this point in the paper. What is meant by a long-range response to light stress- are there processes to respond to light that occur at long-length scales (rather than on the single-cell scale)? If so, is there evidence that these membrane potential changes could induce these responses? Please clarify the logic behind these conclusions.Lines 235-236: In the lower irradiance conditions, the responses are slower overall, and it looks like the ThT intensity is beginning to rise at the end of the measurement. Could a more prominent second peak be observed in these cases if the measurement time was extended?Line 242-243: The overall trajectories of extracellular potassium are indeed similar, but the kinetics of the second peak of potassium are different than those observed by ThT (it rises some minutes earlier)- is this consistent with the idea that Kch is responsible for that peak? Additionally, the potassium dynamics also reflect the first peak- is this surprising given that the Kch channel has no effect on this peak?Line 255-256: Again, this seems like a very strong claim. There are several possible interpretations of the catalase experiment (which should be discussed); this experiment perhaps suggests that ROS impacts membrane potential, but does not obviously indicate that these membrane potential fluctuations mitigate ROS levels or help the cells respond to ROS stress. The loss of viability in the ∆kch mutant might indicate a link between these membrane potential experiments and viability, but it is hard to interpret without the no-light control I mention in the public review.Lines 313-315: "The model predicts... the external light stress". Please clarify this section. Where this prediction arises from in the modeling work? Second, I am not sure what is meant by "modulates the light stress" or "keeps the cell dynamics robust to the intensity of external light stress" (especially since the dynamics clearly vary with irradiance, as seen in Figure 4A).Line 322: I am not sure what "handles the ROS by adjusting the profile of the membrane potential dynamics" means. What is meant by "handling" ROS? Is the hypothesis that membrane potential dynamics themselves are protective against ROS, or that they induce a ROS-protective response downstream, or something else? Later in lines 327-8 the authors write that changes in the response to ROS in the model agree with the hypothesis, but just showing that ROS impacts the membrane potential does not seem to demonstrate that this has a protective effect against ROS.Line 365-366: This section title seems confusing- mechanosensitive ion channels totally ablate membrane potential dynamics, they don't have a specific effect on the first hyperpolarization event. The claim that mechanonsensitive ion channels are specifically involved in the first event also appears in the abstract.Also, the apparent membrane potential is much lower even at the start of the experiment in these mutants- is this expected? This seems to imply that these ion channels also have a blue light independent effect.Lines 368, 371: Should be VGCCs rather than VGGCs.Line 477: I believe the figure reference here should be to Figure 7B, not 6B.Line 567-568: "The initial spike is key to registering the presence of the light stress." What is the evidence for this claim?Line 592-594: "We have presented much better quantitative agreement..." This is a strong claim; it is not immediately evident to me that the agreement between model and prediction is "much better" in this work than in the cited work. The model in Figure 4 of reference 57 seems to capture the key features of their data. Clarification is needed about this claim.Line 613: "...strains did not have any additional mutations." This seems to imply that whole genome sequencing was performed- is this the case?Line 627: I believe this should refer to Figure S2A-B rather than S1.Line 719: What percentage of cells did not hyperpolarize in these experiments?Lines 751-754: As I mentioned above, significant detail is missing here about how these measurements were made. How is "radius" defined in 3D biofilms like the one shown in Video S6B, which looks very flat? What is meant by the distance from the substrate to the core, since usually in this biofilm geometry, the core is directly on the substrate? Most importantly, this only describes the process of sectioning the data- how were these sections used to compute the velocity of ThT signal propagation?I also have some comments specifically on the figure presentation:Normalization from 0 to 1 has been done in some of the ThT traces in the paper, but not all. The claims in the paper would be easiest to evaluate if the non-normalized data were shown- this is important for the interpretation of some of the claims.Some indication of standard deviation (error bars or shading) should be added to all figures where mean traces are plotted.Throughout the paper, I am a bit confused by the time axis; the data consistently starts at 1 minute. This is not intuitive to me, because it seems that the blue light being applied to the cells is also the excitation laser for ThT- in that case, shouldn't the first imaging frame be at time 0 (when the blue light is first applied)? Or is there an additional exposure of blue light 1 minute before imaging starts? This is consequential because it impacts the measured time to the first spike. (Additionally, all of the video time stamps start at 0).Please increase the size of the scale bars and bar labels throughout, especially in Figure 2A and S4A.In Figure 1B and D, it would help to decrease the opacity on the individual traces so that more of them can be discerned. It would also improve clarity to have data from the different experiments shown with different colored lines, so that variability between experiments can be clearly visualized.Results in Figure 1E would be easier to interpret if the frequency were normalized to total N. It is hard to tell from this graph whether the edges and bin widths are the same between the data sets, but if not, they should be. Also, it would help to reduce the opacity of the sparse cell data set so that the full microcluster data set can be seen as well.Biofilm images are shown in Figures 2A, S3A, and Video S3- these are all of the same biofilm. Why not take the opportunity to show different experimental replicates in these different figures? The same goes for Figure S4A and Video S6B, which again are of the same biofilm.Figure 2C would be much easier to read if the curves were colored in order of their size; the same is true for Figure 4A and irradiance.The complementation data in Figure S3D should be moved to the main text figure 3 alongside the data about the corresponding knockout to make it easier to compare the curves.Fig.ure S3E: Is the Y-axis in this graph mislabeled? It is labeled as ThT fluorescence, but it seems that it is reporting fluorescence from the calcium indicator?Video S6B is very confusing - why does the video play first forwards and then backwards? Unless I am looking very carefully at the time stamps it is easy to misinterpret this as a rise in the intensity at the end of the experiment. Without a video legend, it's hard to understand this, but I think it would be much more straightforward to interpret if it only played forward. (Also, why is this video labeled 6B when there is no video 6A?)